# Adaptive laboratory evolution of a genome-reduced *Escherichia coli*

Donghui Choe[1,2], Jun Hyoung Lee[1,2,3], Minseob Yoo[1,2], Soonkyu Hwang[1,2], Bong Hyun Sung[3,4], Suhyung Cho[1,2], Bernhard Palsson [5,6], Sun Chang Kim[1,2,3] & Byung-Kwan Cho [1,2,3]

Synthetic biology aims to design and construct bacterial genomes harboring the minimum number of genes required for self-replicable life. However, the genome-reduced bacteria often show impaired growth under laboratory conditions that cannot be understood based on the removed genes. The unexpected phenotypes highlight our limited understanding of bacterial genomes. Here, we deploy adaptive laboratory evolution (ALE) to re-optimize growth performance of a genome-reduced strain. The basis for suboptimal growth is the imbalanced metabolism that is rewired during ALE. The metabolic rewiring is globally orchestrated by mutations in *rpoD* altering promoter binding of RNA polymerase. Lastly, the evolved strain has no translational buffering capacity, enabling effective translation of abundant mRNAs. Multi-omic analysis of the evolved strain reveals transcriptome- and translatome-wide remodeling that orchestrate metabolism and growth. These results reveal that failure of prediction may not be associated with understanding individual genes, but rather from insufficient understanding of the strain's systems biology.

[1] Department of Biological Sciences, Korea Advanced Institute of Science and Technology, Daejeon 34141, Republic of Korea. [2] KI for the BioCentury, Korea Advanced Institute of Science and Technology, Daejeon 34141, Republic of Korea. [3] Intelligent Synthetic Biology Center, Daejeon 34141, Republic of Korea. [4] Korea Research Institute of Bioscience and Biotechnology, Daejeon 34141, Republic of Korea. [5] Department of Bioengineering, University of California San Diego, La Jolla, CA 92093, USA. [6] Department of Pediatrics, University of California San Diego, La Jolla, CA 92093, USA. These authors contributed equally: Donghui Choe, Jun Hyoung Lee. Correspondence and requests for materials should be addressed to S.C.K. (email: sunkim@kaist.ac.kr) or to B.-K.C. (email: bcho@kaist.ac.kr)

Minimal genomes, containing only the necessary genes to maintain self-replicable life, have been constructed[1–3]. For example, a native 1.08-Mbp *Mycoplasma mycoides* genome and its redesigned version (JCVI-syn3.0) was generated by de novo genome synthesis. Both genomes created viable organisms through genome transplantation. Specifically, the genome of JCVI-syn3.0 was designed based upon essential genes identified using transposon mutagenesis of *M. mycoides*[3]. However, while these fascinating minimal genomes often show unpredictable phenotypes such as growth retardation, a set of genes necessary for survival remains intact. The unexpected phenotypes highlight our limited understanding of bacterial genomes. For instance, all essential genes of *M. mycoides* were contained in the initial design; however, a viable genome could only be constructed after quasi-essential genes, which are not strictly essential but were required for robust growth, were included in the minimal genome. In contrast to this bottom-up approach to genome design, several *Escherichia coli* strains harboring reduced genomes have been constructed by sequential genome reduction mostly without growth retardation in rich media[1,2,4–7]. However, when genome-reduced strains are grown in minimal medium, their growth rate is often reduced. The decreased growth rate has been attributed to our limited understanding of some bacterial genome processes, such as synthetic lethality and interactions between interconnected cellular components, making it difficult to construct minimal genomes with a top-down approach.

To compensate for incomplete knowledge of bacterial genomes, we implement adaptive laboratory evolution (ALE) to allow self-optimization of the unknown processes encoded on a genome. It has been widely reported that ALE rapidly generates desired phenotypes such as tolerance against stresses[8,9], fast growth rates under given media[10], and utilization of non-natural substrates[11]. Those phenotypes are acquired by a number of intriguing mechanisms during adaptation such as mutations on metabolic enzymes[12], rewired serendipitous pathways[11], and transcriptomic re-organization[13,14]. Mutations on metabolic enzymes provide different substrate specificity and kinetic properties. As a global response, transcription machinery is often mutated, which have been reported to remodel cells' catabolic efficiency[15,16]. Moreover, ALE provides valuable insights into the genotype–phenotype relationship by investigating a time series of genomic changes. Thus, we exploit this robust method to recover the innate potential for rapid growth on a given medium and report a growth-recovered genome containing a reduced number of genes enabling rapid growth. Here, we apply ALE to a genome-reduced strain, named MS56, derived from the standard *E. coli* K-12 MG1655 strain, which yields growth retardation in minimal medium. We generated the evolved strain, named eMS57, which exhibits a growth rate comparable to MG1655. This is followed by multiple omics measurements revealing that remodeling of the transcriptome and translatome in eMS57 results in metabolic re-optimization and growth recovery. This comprehensive data provides valuable insights for cellular design principles for synthetic biology.

## Results

**ALE of a genome-reduced *E. coli*.** The *E. coli* MS56 was used as a starting strain for ALE[4]. MS56 was created from the systematic deletion of 55 genomic regions of the wild-type *E. coli* MG1655. The 55 regions had a combined length of approximately 1.1Mbp. No essential genes or genes expressed at a significant level were removed from MG1655. Although MS56 exhibited a comparable growth rate to *E. coli* MG1655 in rich medium (Fig. 1a), it showed severe growth reduction in M9 minimal medium (Fig. 1b). To reveal a molecular basis for the growth reduction, we determined whether MS56 could adaptively evolve to recover the growth rate of its MG1655 parental strain. Due to the low growth rate of MS56 in M9 minimal medium, we initially supplemented 0.1% of LB medium (v v$^{-1}$), which restored the growth rate of MS56 to 2/3 that of MG1655 (Supplementary Figure 1). Then, LB supplementation was reduced in a stepwise fashion to reach supplement-free growth (Fig. 1c). After 807 generations of ALE, we isolated a clone from the evolved population, eMS57, which restored final cell density and growth rate to levels comparable to MG1655 in M9 minimal medium without any nutrient supplementation (Fig. 1d).

We then sought to evaluate the phenotypic differences between eMS57, MS56, and MG1655. In terms of morphology, eMS57 was similar in cell size and length to MG1655, whereas the unevolved MS56 had a longer cell length than the other two strains (Fig. 1e). Similar morphological changes were observed in a previously constructed genome-reduced strain Δ16 that exhibited severe growth retardation[7]. eMS57 showed a narrower spectrum of nutrient utilization for carbon, nitrogen, phosphorus, and sulfur sources than MG1655 due to the deletion of genes responsible for respective nutrient utilization (Fig. 1f, Supplementary Figure 2, and Supplementary Data 1). For example, eMS57 did not show respiration capability on glycolate and glyoxylate as the sole carbon source because the genes responsible for glycolate utilization were removed by MD10 deletion[4]. There was no significant change in phosphorus and sulfur source utilization; however, MG1655 and eMS57 exhibited different nitrogen utilization preferences. The respiration rate of MG1655 in cytidine was much higher than that of eMS57, whereas eMS57 preferentially utilized uric acid as the sole nitrogen source; this may have originated from the deletion of nitrate respiration genes. Interestingly, a batch culture of eMS57 excreted approximately 9-fold higher extracellular pyruvate than MG1655 (Fig. 1g), which was notable since the genes related to pyruvate metabolism were intact in the MS56 genome. As pyruvate transporter-deficient *E. coli* has shown high levels of pyruvate in the medium[17], we tested whether pyruvate uptake is functional in eMS57 using toxic halogenated pyruvate analog[17,18]. eMS57 grows well when fed with pyruvate as a sole carbon source; additionally, its growth was inhibited by 3-flouropyruvate (FP), similar to MG1655, indicating that eMS57's pyruvate uptake function is intact (Fig. 1h). FP interferes with the function of pyruvate dehydrogenase (encoded by *aceE*). Thus, the increased level of pyruvate was not caused by impaired uptake function. Rather, it is regarded as a consequence of metabolic changes in eMS57. Lastly, we obtained the biomass yield of eMS57 equivalent to MG1655 from fed-batch fermentation (Supplementary Table 1). This result illustrates that eMS57 is comparable to its ancestor with regard to industrial-scale applications.

**Causal links between mutations and phenotypic change**. We next sought to elucidate the genetic basis linked with growth improvements and the phenotypic changes using genome resequencing of the evolved and parental strains. We determined the genome sequences of evolving populations at least every 5 days (Supplementary Data 2). Notably, a large genomic region spanning 21 kb in length (genomic coordinates from 2,038,496 to 2,059,460 bp) was deleted spontaneously during ALE (Fig. 2a). The region contains 21 genes including *rpoS* (a sigma factor 38 for transcription initiation of genes associated with stationary phase and stress conditions)[19] and *mutS* (a component of the MutHLS DNA repair system). The deletion mainly occurred after 352 generations and was confirmed by PCR (Fig. 2b). A single knockout of *rpoS* or deletion of the 21-kb region from MS56

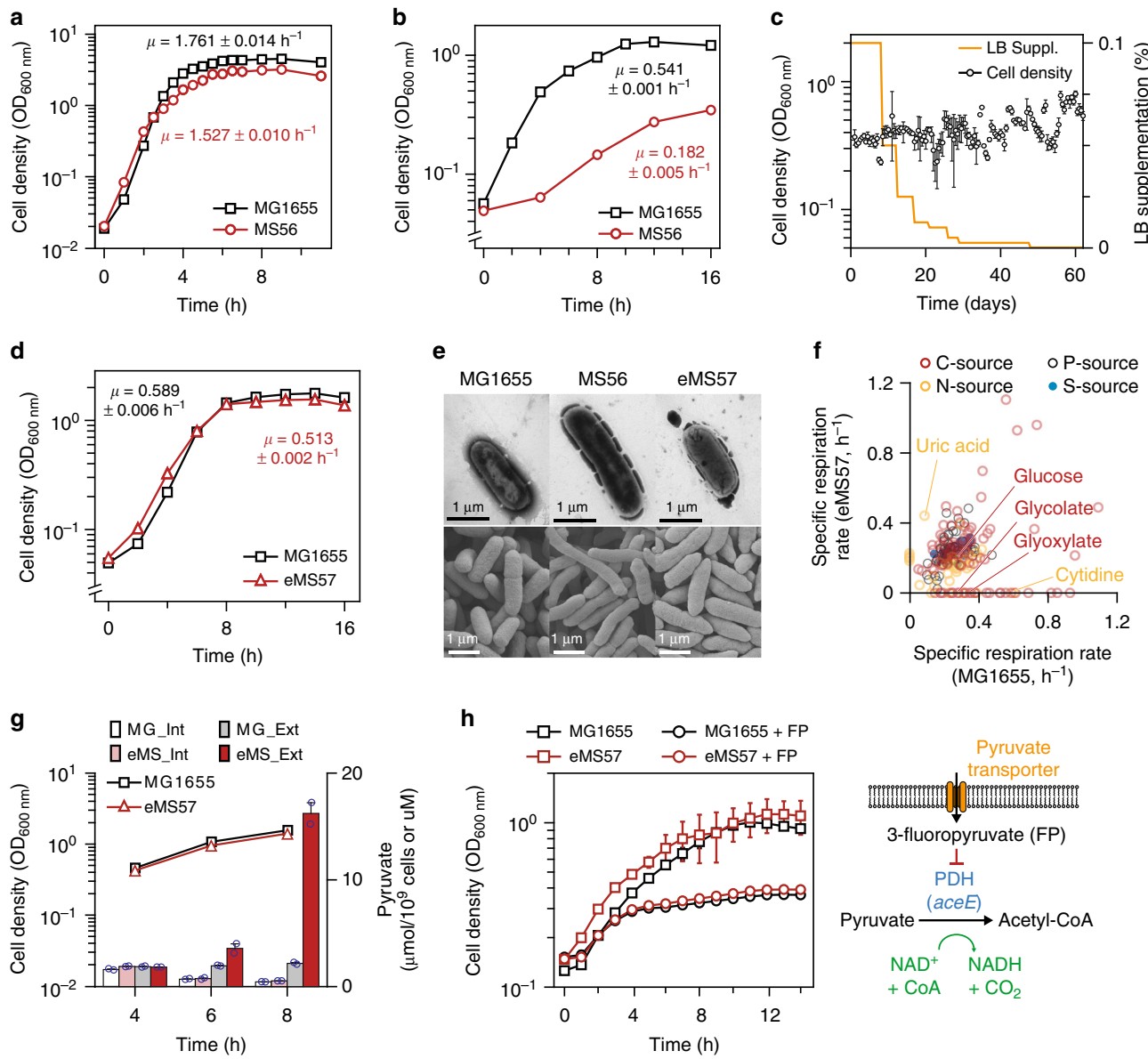

**Fig. 1** Adaptive laboratory evolution (ALE) of a genome-reduced strain (MS56). **a** Growth profiles of genome-reduced strain MS56 (red line) and wild-type *E. coli* MG1655 (black line) in LB medium. $\mu$ indicates specific growth rate. Error bars indicate standard deviation (s.d.) of two biological replicates. **b** Growth profiles of genome-reduced strain MS56 (red line) and wild-type *E. coli* MG1655 (black line) in M9 minimal medium. Error bars indicate s.d. of three biological replicates. **c** Cell growth trajectory showing changes in fitness during the ALE of MS56 in M9 minimal medium with supplementation of LB medium. Cell density was measured after 12 h of three individual batch cultivation (circles) and error bars indicate the s.d. of three individual cultures. LB supplementation was stepwise reduced from 0.1 to 0% over time (orange line). At the end of the ALE experiment, the evolved population exhibited restored growth rate in M9 minimal medium without any nutrient supplementation. **d** Growth profiles of a clone eMS57 (red line) isolated from the ALE population and wild-type *E. coli* MG1655 (black line) in M9 minimal medium. Error bars indicate s.d. of three biological replicates. **e** Morphological changes between MG1655, MS56, and eMS57. Upper panel, TEM images. Lower panel, SEM images. **f** Phenotype microarray characterization of MG1655 and eMS57 showing different nutrient utilization capability. Various carbon sources (red circle), phosphorus sources (black circles), nitrogen sources (yellow circles), and sulfur sources (blue circles). **g** Intracellular and extracellular pyruvate concentrations for MG1655 and eMG57 at 4, 6, and 8 h after inoculation. Int: intracellular pyruvate concentration. Ext: pyruvate concentration in the medium. Black (MG1655) and red (eMS57) lines show cell density at 4, 6, and 8 h after inoculation. Intracellular pyruvate level is presented as mole pyruvate per $10^9$ cells and extracellular pyruvate level was measured in molar concentration. Individual data points are shown as blue circles and error bars indicate the s.d. of two biological replicates. **h** Pyruvate uptake function in MG1655 and eMS57 was examined by growth inhibition induced by a toxic pyruvate analog (3-fluoropyruvate, FP). Error bars indicate s.d. of three biological replicates. Source data are provided as a Source Data file

recovered its growth rate to 80% of that of eMS57; however, the deletion did not fully recover to the growth rate of eMS57 (Fig. 2c). Thus, while it was a dominant factor, this deletion was not the sole cause of the growth rate recovery exhibited by eMS57.

Additionally, a total of 117 mutations including single-nucleotide variations (SNVs), multi-nucleotide variations (MNVs), and insertions and deletions of bases (indels) were observed during ALE (Fig. 2d and Supplementary Data 2). To remove sequencing artifacts, variants with allele frequency <0.1 were discarded and allele frequencies were confirmed by digital PCR (dPCR) (Supplementary Figure 3). Time-course tracking revealed that many sequence variants were linked. For example,

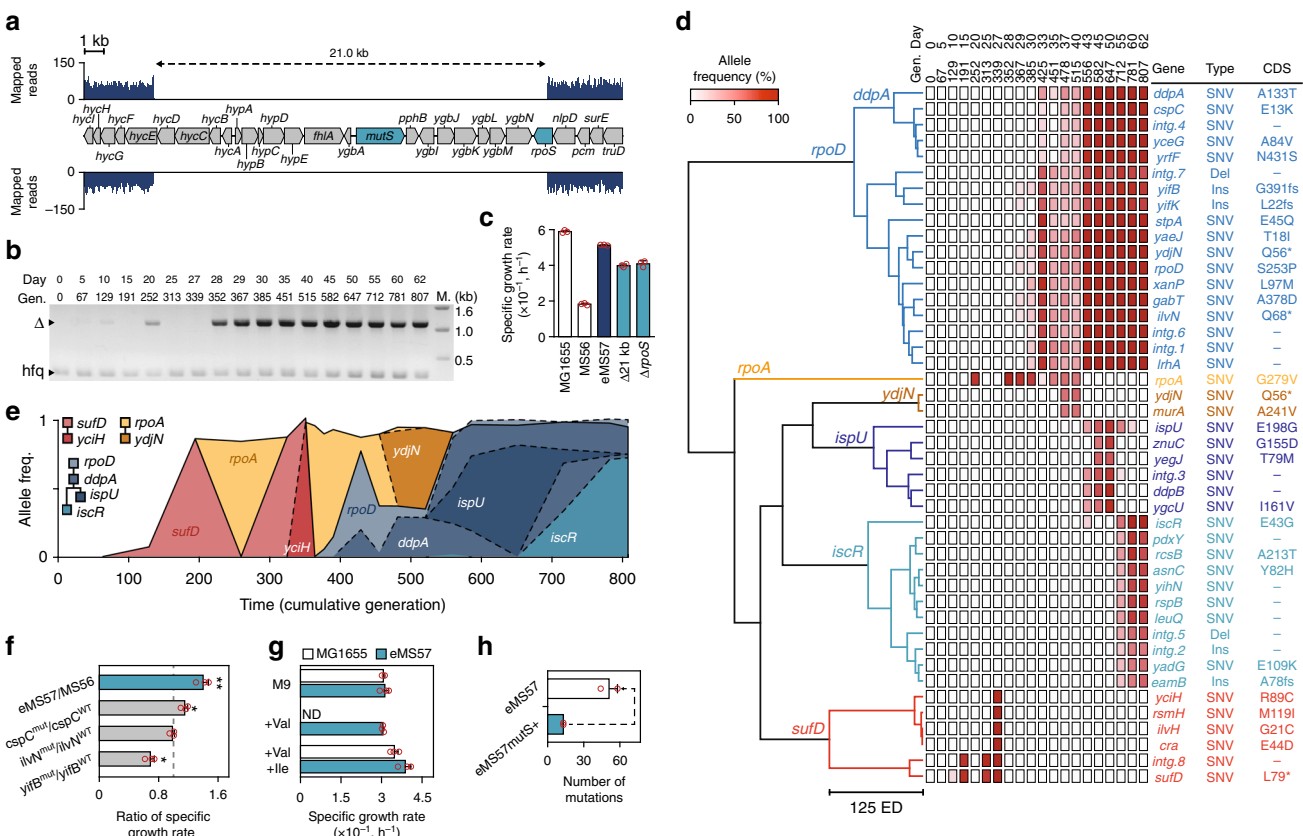

**Fig. 2** Whole-genome resequencing of adaptive laboratory evolution (ALE) experiment. **a** Spontaneous large deletion in eMS57 spanning 21 kb including 21 genes. No sequencing read was mapped onto this region. **b** The occurrence of large deletions was tracked by PCR. Δ indicates PCR amplicon only amplified from the deletion. *hfq* was used as a positive control of PCR. Gen and M indicate cumulative generations and a DNA marker, respectively. **c** Growth rates of MS56 with *rpoS* deletion or a large deletion compared to MG1655 and eMS57. Error bars indicate the s.d. of three individual cultures shown in red circles. **d** Heatmap indicates frequencies of mutations in a given population. Shown are 31 mutations with allele frequency >0.5 at least once during ALE. Dendrogram shows clonal lineages determined by hierarchical clustering. Gen: cumulative generations, SNV: single-nucleotide variation, *: stop codon, fs: frame shift, ED: Euclidean distance. **e** Three lineages and their sub-lineages were inferred from hierarchical clustering. *yciH* sub-lineage emerged within the *sufD* lineage. *ddpA*, *ispU*, and *iscR* are sub-lineages of the *rpoD* lineage. *ydjN* is a sub-lineage of the *rpoA* lineage. Sub-lineages are presented as dotted lines. **f** Construction of the *cspC* point mutation on MS56 had a beneficial effect on fitness, whereas *ilvN* mutation showed no effect. The *yifB* mutation on MS56 decreased the growth rate. Error bars indicate s.d. of three biological replicates shown as red circles. *P value <0.05, **P value <0.01 (two-sided Welch's *t* test, a difference between growth rate of wild-type and mutant were tested, $n = 3$). **g** Growth rate of MG1655 and eMS57 in response to valine supplementation which inhibits cell growth (white bar: MG1655, blue bar: eMS57, error bars indicate s.d. of three biological replicates shown as red circles). eMS57 was completely resistant to valine toxicity. Addition of isoleucine compensated for the inhibitory effect of valine in MG1655. **h** Additional 300 ALE generations of eMS57 and eMS57mutS⁺ revealed that inactivation of *mutS* increased the mutation rate of eMS57. Error bars indicate s.d. of two independent ALE populations (shown as red circles). Source data are provided as a Source Data file

allele frequencies of variations in *yaeJ*, *ydjN*, and *rpoD* were similar to each other and changed in a coherent manner. It is unlikely that many of the mutations emerged simultaneously and independently with similar allele frequency. It is instead more plausible that some variations accumulated in a clone were fixed with a beneficial mutation driving clonal expansion. Thus, we analyzed clonal lineages appearing during ALE using hierarchical clustering of 48 variants, which identified three clonal lineages with five sub-lineages (Fig. 2e). According to allelic frequencies representing clonal abundance, mutations in *sufD* and the intergenic region (between *fepA* and *fes*) emerged after 129 generations (Fig. 2e, *sufD* lineage). The lineage rapidly flourished, so that the majority of cells in the population carried the mutations (frequency = 86%) after 191 generations. Afterward, a lineage having one missense mutation on *rpoA* (RNA polymerase α-subunit) arose (Fig. 2e, *rpoA* lineage). The two lineages competed from generation 252 to 339 and the *sufD* lineage finally disappeared after 352 generations. The *rpoA* lineage appeared to be fixed as its density increased to 94.5% of the entire population;

however, another lineage emerged, carrying 13 mutations, including *rpoD* (housekeeping sigma factor; σ⁷⁰) (Fig. 2e, *rpoD* lineage). Mutation of transcription machinery represents the most prominent means for *E. coli* to optimize its metabolism against environmental perturbation[20]. However, it is unlikely that all 13 mutations were beneficial for cell fitness and emerged together. Instead, it is more likely that the low-frequency variants accumulated during 367 generations expanded along with one or two trigger mutations.

To test the effect of individual mutations in the *rpoD* lineage, we randomly selected three mutations and inserted them into the MS56 genome. The individual mutations showed different effects on growth rate. First, a mutation in the cold shock stress-related gene *cspC* ($E^{13} \rightarrow K$) increased the growth rate (Fig. 2f). It has been reported that laboratory strains of *E. coli* frequently gain loss-of-function mutations in *cspC*[21], such that inactivation of CspC is preferred in genome-reduced strains as well. Mutation of *ilvN* encoding regulatory subunit of acetohydroxy acid synthase I (AHAS 1) did not recover the growth rate of MS56, indicating

that the mutation might be a hitchhiker (Fig. 2f). It is understood that non-functional isoleucine synthetic AHAS II induces recurrent isoleucine starvation in the K-12 strain[22]. The K-12 strain cannot synthesize sufficient isoleucine because of feedback inhibition of the remaining AHAS III and I by another product, valine. In a previous report, a mutation on *ilvN* makes the AHAS I resistant to feedback inhibition by valine[23]. Thus, we hypothesized that mutant IlvN in eMS57 may have similar resistance. As expected, only eMS57 was resistant to valine supplementation that inhibits cell growth by blocking isoleucine biosynthesis (Fig. 2g). The valine toxicity in MG1655 and MS56 was completely diminished by isoleucine co-supplementation, indicating that isoleucine starvation was a cause of the growth arrest. Thus, we concluded that mutations on *ilvN* have selective advantages by rewiring isoleucine metabolism, although it did not recover the apparent growth rate of MS56. In addition, this indicates that an adaptive evolution would not solely depend on cellular growth rate. Notably, the growth rate of MS56 carrying *yifB* (predicted protease) frameshift insertion was 69% of the parental strain (Fig. 2f). It is unclear how a deleterious *yifB* mutation became enriched during adaptation; however, we suspect that the combination of multiple mutations may have an opposite effect of individual mutations as previously reported[24].

We next conducted ALE of MG1655 in M9 glucose medium with LB supplementation. After 800 generations, we identified 101 mutations from three individual populations (Supplementary Figure 4 and Supplementary Data 3). None of the advantageous mutations in the eMS57 genome, such as *rpoS/mutS* inactivation and *rpoD* mutation, were found from the ALE of MG1655. Instead, the ALE populations have mutations on *rpoC* (RNA polymerase β′-subunit) and *rpoB* (RNA polymerase β-subunit; Supplementary Data 3). These two genes are the most frequently mutated genes during the ALE of *E. coli*[15,25–27], inducing large-scale transcriptional reprogramming[20,28]. Since the *rpoC* and *rpoB* mutations are common in the adaptively evolved *E. coli*, the *rpoD* mutation on eMS57 is quite a unique feature of genome-reduced bacteria. Additionally, four mutations (*nfrA*, *glpA*, *yfaL*, and *yifB*) occurred during the ALE of both MG1655 and MS56. NfrA is an outer membrane bacteriophage N4 receptor and YfaL is a putative autotransporter adhesin. The culture condition in this study is not related to functions of the two genes, however, considering they are membrane proteins, there should be a selective advantage by mutating them. Interestingly, deleterious mutation *yifB*, encoding a putative ATP-dependent protease, whose function remains unknown, was also observed during the ALE of MG1655. This is a clear indication of epistasis of mutations and evolutionary convergence between MG1655 and eMS57.

Taken together, clonal lineage analysis and the respective point mutations in MS56 demonstrated how subtle genetic variations orchestrate rapid adaptation of genome-reduced *E. coli* to growth conditions. It is notable that the genome-reduced MS56 followed a similar adaptation trajectory, such as mutating RNA polymerase subunits and inactivating unnecessary proteins, with the limited repertoire of genes. However, molecular players involved in the functional changes in eMS57 seem to be different due to the fundamental difference in gene composition.

**Spontaneous inactivation of a DNA repair system.** MS56 appears to have a very high mutation rate (one mutation per 32.0 and 4.3 generations before and after *mutS* deletion, respectively). To understand the basis of such a high mutation rate, we hypothesized that the deletion of *mutS* and subsequent inactivation of the MutHLS system increased the mutation rate. In fact, the DNA

repair system is occasionally inactivated during ALE, resulting in hyper-mutator strains[24,29]. To test whether such characteristics occur in the genome-reduced strain, we adaptively evolved a *mutS*-restored eMS57 strain (eMS57mutS+, Supplementary Figure 5) for an additional 300 generations. The eMS57 in two replicated cultures accumulated a total of 102 new mutations, whereas the eMS57mutS+ strain obtained only 26 mutations (Fig. 2h and Supplementary Data 4). Thus, *mutS* deletion was sufficient to increase the mutation rate, causing rapid adaptation toward given conditions. The parental MS56 strain exhibited a low genome evolution rate because mobile DNAs, such as IS elements, were removed[6]. The decreased evolvability of MS56 was compensated by inactivation of the MutHLS system as a hyper-mutator of laboratory evolution[15,24]. Thus, we observed that the genomic mutation rate can be accelerated by a spontaneous inactivation of the DNA repair system, whose restoration is required for constructing a stable production host to avoid further mutation of target proteins.

**Transcriptome reprogramming by the mutant transcription machinery.** Immediately after *mutS* deletion, a mutation in *rpoD* (S[253] → P) emerged. As σ[70] has a housekeeping role in providing specificity to the RNA polymerase core enzyme on the majority of promoters, the mutant σ[70] was expected to reprogram the transcriptome to adapt to the growth conditions[10,20,30]. Additionally, 63 transcription factors were deleted in MS56. Although none of the deletions induced severe growth defects (Supplementary Figure 6), they may have caused transcriptional interference in MS56. Thus, we examined genome-scale binding of mutant σ[70] and resulting transcriptomic changes by chromatin immunoprecipitation coupled with sequencing (ChIP-Seq) and whole transcriptome sequencing (RNA-Seq). A total of 421 and 418 binding sites in wild-type and mutant σ[70] were determined, respectively (Supplementary Figures 7, 8, and Supplementary Data 5). All σ[70]-binding sites were cross-validated with the RegulonDB database[30], indicating that the two σ[70] had 320 shared binding sites (Fig. 3a). Excluding 45 binding sites in the deletion regions, wild-type and mutant σ[70] had 56 and 98 unique binding sites, respectively. Although promoter structures in 320, 56, and 98 binding regions showed no dramatic difference, promoters with a thiamine or adenine at the fifth nucleotide of −35 element were preferentially used by the mutant σ[70] (Supplementary Figure 9). To examine the effect of Ser253Pro mutation on the specificity of σ[70], we compared native and mutant σ[70] in MG1655. Under control of the Trc promoter, native or mutant σ[70] tagged with the c-Myc epitope was expressed in MG1655 and bound DNA fragments were immunoprecipitated and quantified by quantitative PCR (qPCR). Mutant σ[70] showed high specificity to E promoters (Supplementary Figure 9), while the specificity to M promoters remained unchanged. Thus, the mutant σ[70] bound an additional set of E promoters, while the loss of ability to bind M promoters was not caused by the mutation.

RNA-Seq revealed that the specific binding of σ[70] resulted in transcriptomic changes. The expression levels of the unique genes transcribed by mutant σ[70] were significantly increased (Fig. 3b and Supplementary Figure 10, *P*-value < 0.001, two-sided Wilcoxon's rank-sum test). For example, an operon *deoCABD* encoding enzymes for deoxynucleotide degradation was transcribed differently between MG1655 and eMS57. The *deoCABD* operon is regulated by two promoters: one in an upstream gene, *yjjI*, expressing the entire operon, the other transcribing just *deoB* and *deoD* upstream of *deoB*. Unlike the wild-type σ[70], the mutant σ[70] bound strongly to promoters that regulate expression of the entire operon, resulting in the *deoCABD* operon being highly expressed

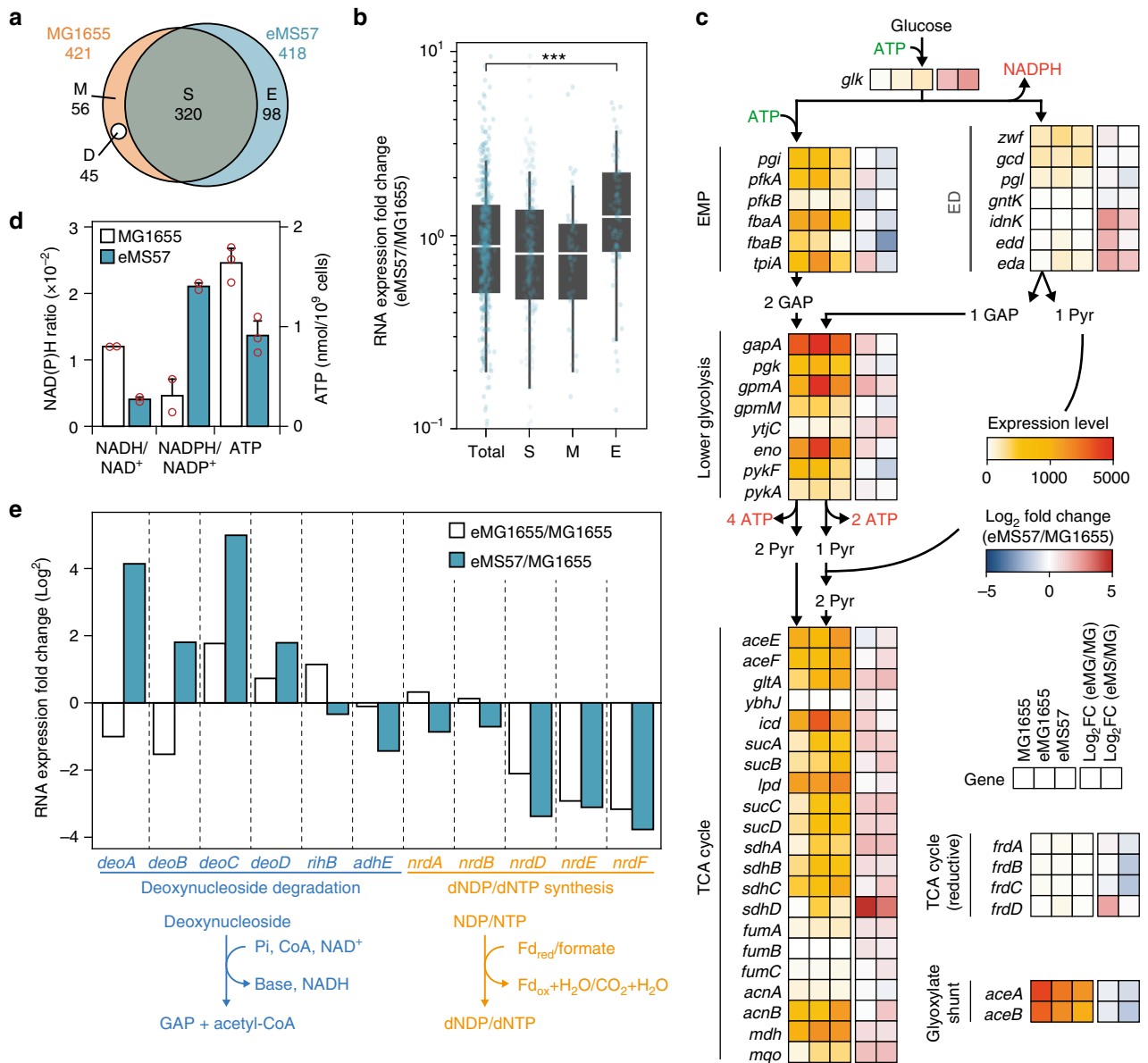

**Fig. 3** Transcriptome analysis of eMS57. **a** A total of 421 and 418 binding sites of σ70 (MG1655) and mutant σ70 (eMS57), respectively, were determined by chromatin immunoprecipitation sequencing (ChIP-Seq); 320 sites are shared (S). Except for eMS57 deleted regions (D), wild-type (M) and mutant σ70 (E) specifically binds to 56 and 98 promoters, respectively. **b** Box and whisker plots show changes of gene expression between MG1655 and eMS57 according to differential binding of σ70. T: total promoters examined, S: shared promoters, M: MG1655-specific promoters, E: eMS57-specific promoters. ***P value < 0.001 (two-sided Wilcoxon's rank-sum test). Box limits, whiskers, center lines indicate 1st and 3rd quartiles, 10 and 90 percentiles, and median of the distribution, respectively. Dots are individual genes. **c** Glycolysis and TCA cycle expression levels are shown with indication of the required cofactors. EMP: Embden–Meyerhof–Parnas pathway, ED: Entner–Doudoroff pathway, GAP: glycerol-3-phosphate, Pyr: pyruvate. **d** Intracellular NADH/NAD+ ratio was decreased and NADPH/NADP+ ratio was increased in eMS57. ATP intracellular level was decreased in eMS57. Red circles indicate three independent assays from biological replicates. Error bars indicate the s.d. of three biological replicates. **e** Relative gene expression for deoxynucleoside degradation and synthesis between the evolved strains (eMS57 and eMG1655) and the wild-type *E. coli* K-12 MG1655. Source data are provided as a Source Data file

only in eMS57 (Supplementary Figure 11). With this example, we assumed that the mutation in σ70 induces transcriptome remodeling, enabling eMS57 to adapt in M9 minimal medium (Supplementary Data 6).

**Metabolic perturbation by transcriptomic reprogramming.** Of 3457 genes in eMS57, we identified 356 differentially expressed genes (DEGs, P-value <0.01, two-sided Welch's t test). DEGs were first analyzed by functional enrichment analysis (Supplementary Figure 12). Among those differentially enriched cellular functions, we first focused on two different glycolytic pathways,

Embden–Meyerhof–Parnas (EMP) and Entner–Doudoroff (ED). The EMP pathway, which is a primary pathway of glycolysis in many species, is composed of 10 enzymatic cascades to convert one molecule of glucose into two molecules of pyruvate, generating two molecules of ATP and NADH. Meanwhile, one ATP, one NADH, and one NADPH are obtained from one molecule of glucose metabolized by the ED pathway, which is composed of a smaller number of enzymes than the EMP pathway. Expression levels of EMP genes were 7.20- and 7.25-fold higher than ED pathway genes in MG1655 and eMG1655 (a clone isolated from the evolved MG1655 population) (Fig. 3c and Supplementary Figure 13). In eMS57, EMP-specific enzyme expression levels

were only 3.61-fold higher than ED enzyme levels (Fig. 3c and Supplementary Figure 13), indicating that eMS57 utilizes glucose via the ED pathway in part, losing one ATP but gaining NADPH from NADH. Relative utilization of ED to EMP pathway involves several tradeoffs, including differential proteome costs (the ED requiring less than EMP pathway)[31], differential ATP, NADH, and NADPH generation[31], and global regulation; ED pathway bypasses the catabolite repression induced by glucose metabolism. To support this, we measured cellular $NAD^+$/NADH, $NADP^+$/NADPH ratios, and ATP levels to gain further understanding of the shifts in cofactor generation (Fig. 3d). In eMS57, the NADH level was 3.0 times lower and the NADPH level was 4.5 times higher than MG1655. Moreover, the ATP level in eMS57 was maintained at a lower level as expected from the transcriptomic change. The distinct glycolytic strategy of eMS57 enables the strain to use glucose effectively, potentially contributing to growth recovery of MS56.

The 1.1 Mb genome reduction comes with lowered demand for deoxynucleotide triphosphates (dNTPs), leading to a predicted shift in metabolic pathway usage. Transcriptomic analysis showed that the expression levels of deoxyribonucleotide biosynthesis were lower in both eMG1655 and eMS57 compared to MG1655, and conversely deoxynucleoside degradation was higher in eMS57 than eMG1655 and MG1655 (Fig. 3e and Supplementary Figure 14). Salvage of dNTP surplus is considered to be a distinct metabolic feature of eMS57, whereas synthesis is inhibited in both evolved strains. As a result of the nucleotide salvage pathway, large amounts of degradation product enter lower glycolysis as glyceraldehyde-3-phosphates. All genes, except *acnA*, responsible for forward flux (catabolic) from lower glycolysis to the TCA cycle were upregulated in eMS57 (Fig. 3c). Increased influx of metabolites into glycolysis may contribute to increased pyruvate levels (Fig. 1h). Overall, increased levels of central carbon metabolism and a reduction of metabolic burden from the reduced genome synthesis appear to be key factors for eMS57 growth restoration. Thus, the unknown processes that lead to growth reduction are not associated with the function of individual gene products but are systemic in nature. These results highlight that insufficient understanding of *E. coli*'s systems biology is an underlying cause of failure to predict the consequences of genome reduction. Thus, ALE-driven understanding of the strain's systems biology lowers the probability of unknown host factors adversely impacting production strain design.

**Reprogrammed protein expression by post-transcriptional regulation**. In addition to transcriptome remodeling, we sought to determine the changes in the translation landscape of the evolved strain. Translation processes in microbes have been systematically addressed by capturing ribosome-protected messenger RNA (mRNA) fragments (RPFs)[32]. Transcription of a gene does not directly represent intracellular levels of encoded protein because of post-transcriptional regulation[33,34]. In *Streptomyces*, the translation level of genes related to antibiotics production did not correspond with the increased transcription level[33]. A similar regulation has also been reported in yeast[34]. This suggests the presence of complex regulation at the post-transcriptional level, functional dynamics of transcription–translation (i.e., ribosome hibernation), or spatiotemporal resource limitation for translation, although the underlying mechanism remains largely unknown.

Thus, we measured the translational level of MG1655 and eMS57 by ribosome profiling to examine whether the transcriptional changes were consistent with changes in the translational levels (Supplementary Figure 15 and Supplementary Data 7). As a result, in MG1655, the translational efficiency (TE), calculated by dividing RPF levels by the corresponding mRNA transcript levels for each gene, decreases as transcription level increases (Supplementary Figure 16), showing that MG1655 had similar translational buffering as in *Streptomyces* and yeast[32,33]. Surprisingly, eMS57 exhibited no translational buffering, as TE values remained relatively constant regardless of expression level (Supplementary Figure 16). For example, translation of *gapA* (encoding G3P dehydrogenase), one of the most highly transcribed genes (98.8% and 98.7% percentile in MG1655 and eMS57, respectively), was not concomitantly high in MG1655 compared to eMS57 (Supplementary Figure 16). We further investigated whether TE in eMS57 is higher than that in MG1655 using a monomeric mutant of coral red fluorescence protein (mRFP1)[35]. Indeed, eMS57 showed 3.1-fold higher fluorescence intensity than MG1655 (Supplementary Figure 16).

We assume that the high correlation of transcription and translation in eMS57 indicates the increased TE of highly transcribed genes. Meanwhile, we could not resolve the significant change in TE of transcription and translational machinery of eMS57. Although transcription of 54 ribosomal proteins was increased with a mean of 1.6-fold, translation of ribosome remained unchanged (Fig. 4a). Along with ribosomal genes, transcription and translation of genes responsible for maintaining DNA polymerase and RNA polymerase levels remained unchanged as well (<2-fold; Fig. 4b). Thus, reduced translational buffering is unlikely to be induced by an abundance of transcription or translation machinery. Initiation and termination are major rate-limiting and energy consuming steps in translation. These two steps are likely to be different between MG1655 and eMS57, if the translational buffering originated from the kinetics of translation. According to a meta-analysis of ribosome density, MG1655 and eMS57 showed no difference near proximity of the start and stop codon (Supplementary Figure 17). To examine whether the translational buffering was an artifact of high ribosome density at the start and stop codon, we recalculated the RPF level of each coding sequence (CDS) excluding 30 bp from both ends (Supplementary Figure 17). The translational buffering remained unchanged in the recalculated RPF levels, indicating that it was not originated from translational kinetics.

Next, we examined sequence level difference on the 5′-untranslated region (5′-UTR). We sought a common sequence motif in 5′-UTR of 91 coding sequences (CDSs) that are translationally buffered in MG1655 (TE < 0.8) and unbuffered in eMS57 (0.91 < TE < 1.1). There was no particular sequence motif other than the well conserved Shine–Dalgarno sequence (AAGGAG) (Supplementary Figure 18). Because structure and interaction of the 5′-UTR with ribosomes play a critical role in translation, we computationally predicted translation initiation rate (TIR), which is calculated collectively from multiple factors such as RNA structure and interaction with ribosomes[36]. TIR of the 91 CDSs with low TE and random CDSs showed no correlation with TE (Supplementary Figure 18). Conclusively, there was no difference in translation mechanism between the two strains, the specific sequence motif, and the RNA structure. Despite the same ribosome profile and sequence motif, the genes in eMS57 showed low variance in TE distribution (Supplementary Figure 16). Thus, a reduced number of genes provides an increased level of available ribosomes and establishes the unbuffered translation in the reduced genome *E. coli*.

Meanwhile, the correlation between transcription and translation was even higher in central carbon metabolism and nucleic acid metabolism (Fig. 4b). Unlike central carbon and nucleic acid metabolism, translation of branched-chain amino acid (BCAA) synthetic genes behaved differently from their transcription (Fig. 4b). Specifically, translation of an entry point of BCAA

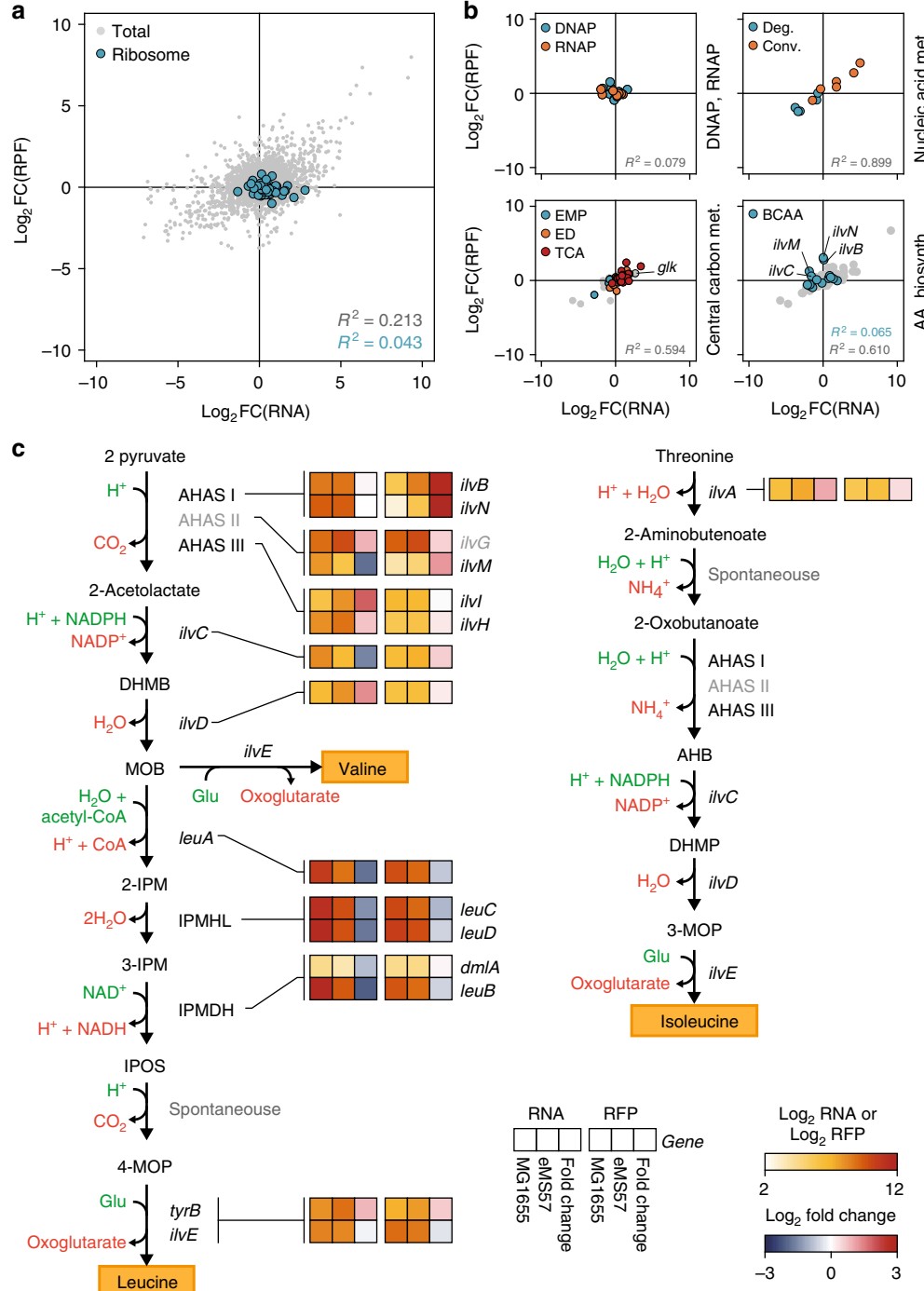

**Fig. 4** Post-transcriptional changes in eMS57 analyzed by ribosome profiling. **a** Correlation between translation and transcription changes in eMS57 compared to that in MG1655. Translational changes in eMS57 generally showed a weak correlation with transcriptional change (Total, Pearson's $r^2$ of 0.213). Transcription of ribosomal proteins was upregulated, whereas translation remained relatively unchanged. Each dot indicates a gene. Pearson's correlation constants ($r^2$) between transcription and translational changes in total genes (gray dots) and ribosome (blue dots) are presented. FC: fold change (eMS57/MG1655). **b** Correlation between transcription and translation changes in DNA synthesis machinery (DNAP), transcription machinery (RNA polymerase), nucleic acid metabolism, central carbon metabolism, and amino acid biosynthetic pathway. No changes were observed in the transcriptional and translational levels of DNA synthesis and transcription machinery. Translation of genes responsible for nucleic acid and central carbon metabolism correlated linearly to transcriptional change. BCAA biosynthetic genes were translated differently than transcription in eMS57. Each dot indicates a gene and dots are colored by their function or pathway. Correlation between transcription and translational changes were examined by Pearson's correlation ($r^2$). FC: fold change (eMS57/MG1655). **c** Transcription and translation level of BCAA biosynthesis pathway. Translation of valine-resistant acetohydroxy acid synthase I (AHAS I) were markedly upregulated in eMS57. DHMB: 2,3-dihydroxy-3-methylbutanoate, MOB: 3-methyl-2-oxobutanoate, 2-IPM: 2-isopropylmalate, IPOS: 2-isopropyl-3-oxosuccinate, 4-MOP: 4-methyl-2-oxopentanoate, AHB: 2-aceto-2-hydroxybutanoate, DHMP: 2,3-dihydroxy-3-methylpentanoate, 3-MOP: 3-methyl-2-oxopentanoate, IPMHL: isopropylmalate hydrolyase, IPMDH: isopropylmalate dehydrogenase. Metabolites colored green and red are consumed and produced by given enzymatic reaction, respectively

synthesis from pyruvate, mediated by AHAS I, was increased (Fig. 4b and c). In *E. coli*, enzymes for amino acid biosynthesis are controlled by complex transcriptional regulations[37,38]. However, expression of AHAS I was increased without a transcriptional change. As described earlier, AHAS I in eMS57 is resistant to valine. It seems that the strain rewired BCAA biosynthesis by selectively translating AHAS I, bypassing the regulatory framework for fully functional AHAS I, II, and III. This example of post-transcriptional change shows that both transcription and translation were remodeled during ALE.

## Discussion

This study demonstrates a way to improve growth phenotypes of a genome-reduced strain in laboratory conditions. ALE provided an efficient way to restore the fitness of genome-reduced *E. coli* without additional genome engineering. Considering the cost and time for de novo genome synthesis, integration of ALE with rational genome reduction can reduce the practical challenges in the top-down approach to minimal genome construction. ALE as a learning tool reveals a lack of understanding of the reduced strain's systems biology.

Systematic development of an understanding of eMS57 provides specific lessons for genome-wide engineering of bacteria. The clearest overall lesson is that growth retardation in genome-reduced strains results from a metabolic imbalance; a systemic function. The major metabolic pathways in eMS57 are reconfigured during ALE to produce a new mix of three key cofactors (ATP, NADH, and NADPH), altering pyruvate metabolism and reducing deoxynucleotide synthesis. We found that the suboptimal metabolic state of MG56 was fixed during ALE through transcriptome reprogramming by RNA polymerase mutation. In previous reports of ALE, RNA polymerase was the most frequently mutated protein complex. Mutations in subunits constituting the holoenzyme, additional factor *nusA*, and *rho* have been reported[27]. During the ALE of MS56, we detected mutations in *rpoA* and *rpoD*. In RpoA, four mutants (R33H, two R317C, and R317H) were previously observed in ALE under heat stress. In our study, the mutation was located in the α-C-terminal domain, which is involved in the interaction with CRP and upstream elements of the promoter[39]. The mutation in α-CTD may confer a selective advantage on glucose medium through different CRP regulation or transcriptional changes. However, the mutation in *rpoA* was eliminated by another mutant carrying a *rpoD* mutation. *rpoD*, in contrast, was the fifth most mutated gene among more than 100 cell lines produced by Tenaillon et al.[27]. Harden et al.[40] also reported mutations on *rpoD* during acid adaptation[40]. Considering that most mutations occurred in auto-inhibitory region 1.1 (sequesters the DNA-binding region of free RpoD) when heat-adapted[27], while acid-adapted *E. coli* and eMS57 accumulated mutations in the non-essential flexible linker region[40], there may be a functional context specifying the subunits and domain of RNA polymerase to be mutated depending on the selective stress. Although the functional consequence of *rpoD* mutation was not examined previously, we determined its impact on a specific set of promoters and thus on the transcriptome and concomitant metabolic flux re-optimization that led to optimal growth of eMS57. In addition, we would like to acknowledge that 63 transcription factors, composed of 28 families, were deleted in MS56. Except for the *dicA* single knockout strain (which can only be deleted with *dicB*[41]), we have confirmed no significant growth defect of *E. coli* Keio strains with one of the 62 TFs deleted (Supplementary

Figure 6). Although the growth was unchanged by transcription factor deletions, there must be transcriptional perturbation that sporadically interferes with new transcriptomic balance in eMS57.

We also learned that the translatome has been remodeled through diminished translational buffering. Recent reports suggest that spatial proximity of genes induces buffering at the protein level[42]. A mutation on *yaeJ*, known to rescue stalled ribosomes by hydrolyzing peptides and recruiting release factors[43], was the only mutation related to translation; however, it seems unrelated to translational buffering because *yaeJ* was silent in both MG1655 and eMS57. In addition, expression levels of ribosomal proteins and auxiliary factors (such as those for translation initiation, elongation, termination, and modulation) were unchanged. Thus, it remains unclear whether diminished translational buffering occurs because of reduced numbers of genes transcribed in a cell or the conservation of resources for translation otherwise used to produce unnecessary proteins and metabolites. With reduced translational buffering, translation in eMS57 correlates with transcriptional changes. Interestingly, translational regulation may differ depending on gene functions. Translation of metabolic pathways showed high correlation with transcriptional changes as those genes were occasionally exposed to environmental perturbation. In contrast, the correlation of DNA polymerases, transcription machinery, and ribosomes did not change during adaptation, indicating that *E. coli* may sustain the most essential cellular functions. Frequent mutations on RNA polymerase subunits observed from adaptation experiments may be a way to remodel transcriptome. With the selective translation between two AHAS, we found that some genes can be fine-tuned at a post-translational level. The truncated mutant of regulatory subunit IlvN of AHAS I induced by premature stop codon mutation (Fig. 2d and Supplementary Data 2) may provide advantageous translational preference.

Finally, we learned that genome reduction of unnecessary cellular processes for condition-specific strain design and re-optimization of the systemic function of the remaining gene products by adaptation leads to a comparable descendent strain. The eMS57 strain can be further engineered for applications of genome-reduced *E. coli* that is a synthetic cell chassis upon which new cellular functions can be designed. Revising reduced fitness of genome-reduced organisms is challenging due to limited understanding of their genomes. ALE of the genome-reduced *E. coli* rapidly re-optimized cellular fitness without a deep understanding of them. By systematically analyzing biological changes during adaptation, metabolic imbalance, transcriptomic, and translatomic perturbation could be inferred. Ribosome profiling of adaptively evolved *E. coli* elucidates translational evolution resulting in unbuffered translation and metabolic rewiring, in addition to transcriptional remodeling via RNA polymerase mutation. This understanding provides valuable design principles for genome minimization and even for bottom-up genome synthesis.

## Methods

**Bacterial strains and primers**. Cells were grown in M9 glucose medium (47.75 mM of $Na_2HPO_4$, 22.04 mM of $KH_2PO_4$, 8.56 mM of NaCl, 18.70 mM of $NH_4Cl$, 2 mM of $MgSO_4$, 0.1 mM of $CaCl_2$, and 2 g $l^{-1}$ of glucose), unless stated otherwise. Isoleucine or valine was supplemented with a final concentration of 0.15 mM, when appropriate. Primers and probes used in this study are listed in Supplementary Table 2. Uncropped and unprocessed gel images are provided in Source Data. Bacterial strains and plasmids used in this study are summarized in Supplementary Table 3.

**ALE and strain isolation.** *Escherichia coli* MS56 was grown in 50 ml of M9 glucose medium in a 250-ml Erlenmeyer flat-bottom flask at 37 °C with agitation. To support the growth of MS56 in M9 glucose medium, 0.1% LB medium was supplemented initially. The supplementation was reduced in a stepwise manner to eventually achieve supplement-free growth. Batch cultures were manually transferred to fresh medium every 12 h at an initial optical density at 600 nm ($OD_{600 \, nm}$) of approximately 0.005. Number of cell divisions during ALE was calculated from final and initial cell densities according to the Eq. (1):

$$\text{Number of generation} = \log_2 \frac{\text{Final cell density}}{\text{Initial cell density}}. \quad (1)$$

After ALE, single clones were isolated on M9 glucose agar medium. Because the clones showed equivalent growth rate, one of the clones was selected for further analyses and experiments (named eMS57 from ALE of MS56 and eMG1655 from ALE of MG1655).

**Electron microscopy.** For scanning electron microscopy (SEM), 1 ml of exponential phase culture was prefixed in 2.5% paraformaldehyde–glutaraldehyde mixture buffered with 0.1 M phosphate buffer (pH 7.2) at 4 °C for 2 h. Next, the prefixed sample was treated with 1% osmium tetroxide solution buffered with 0.1 M phosphate buffer (pH 7.2) for 1 h at room temperature (25 °C). The fixed sample was dehydrated in graded ethanol, substituted by isoamyl acetate, and critical point-dried in liquid $CO_2$. The sample was finally sputter-coated with gold in a Sputter Coater SC502 (Polaron, Quorum Technologies, East Sussex, UK) to 20 nm thickness and SEM images were obtained using the FEI Quanta 250 FEG SEM (FEI, Hillsboro, OR, USA) installed at the Korea Research Institute of Bioscience and Biotechnology at a 10-kV acceleration voltage. For transmission electron microscopy, a sample fixed using the same method as used for SEM imaging was dehydrated in graded ethanol, substituted with propylene oxide, and embedded in Epon-812 resin for 36 h at 60 °C. The embedded sample was ultra-sectioned with an Ultracut E Ultramicrotome (Leica, Wetzlar, Germany) and double-stained with uranyl acetate and lead citrate. The sample was examined under a CM20 transmission electron microscope (Philips, Amsterdam, Netherlands) installed at the Korea Research Institute of Bioscience and Biotechnology at a 100-kV acceleration voltage.

**Phenotype microarray.** Cells were streaked on Biolog Universal Growth Agar (Biolog) plates and grown overnight at 37 °C. Then, cells were resuspended and diluted with Inoculating Fluid A (80% IF-0a GN/GP Base in sterile water; Biolog) to 42% transmittance (T) measured using a Turbidimeter (Biolog). A 42% T cell resuspension was diluted with Inoculating Fluid B (83.33% IF-0a GN/GP Base and 1.2% Biolog Redox Dye mix A in sterile water) to generate 85% T cell resuspension. For PM plate 3B and 4A, 19.8 mM of sodium succinate and 1.98 nM of ferric citrate were added to Inoculating Fluid B as carbon sources. Finally, 100 µl of the 85% T cell resuspension was inoculated onto PM plates and cellular respiration was measured using an Omnilog instrument (Biolog).

**Fed-batch fermentation.** Fed-batch fermentation was conducted in a 2-L stirred-tank reactor containing 1 L of LB or M9 glucose medium at 37 °C. The culture was aerated with 1 bar compressed air with a rate of 200 ml min$^{-1}$ and agitated by pitched-blade impellers with speed controlled from 1000 to 1800rpm so as $pO_2$ was not to drop below 90% saturation. Feeding solution (50% glucose (wv$^{-1}$), 23.65 mM $MgSO_4$, and 8.16 mM $CaCl_2$) was added at a rate of 20 ml hr$^{-1}$ to support exponential growth. Antifoam 204 (Sigma) and 2 M NaOH were added to remove excess foam and maintain pH of the medium.

**Whole-genome resequencing.** Genomic DNA of *E. coli* was isolated using a Wizard Genomic DNA Purification Kit (Promega) according to the manufacturer's instruction. Briefly, cells were collected by centrifugation and lysed by incubating at 80 °C for 5 min with Nuclei Lysis Solution. The cell lysate was incubated at 37 °C for 15 min with 1.2 µg of RNase A (Qiagen). Protein was precipitated with Protein Precipitation Solution and genomic DNA in the supernatant was purified by isopropanol/ethanol precipitation. Finally, DNA was rehydrated by distilled water. The resequencing library was constructed from the isolated genomic DNA using the TruSeq DNA LT Sample Prep Kit (Illumina) according to the manufacturer's instruction. Then, the library was sequenced using a MiSeq Reagent Kit v2 (Illumina) in a 50 cycle single-ended reaction on the MiSeq instrument (Illumina).

**Determination of clonal lineages in the ALE population.** Sequence variants were mathematically clustered according to their time-course allelic frequencies. Sequence variants with allelic frequency >0.8 once or >0.5 at more than two time points were subjected to hierarchical clustering. Clustering was performed using the SciPy clustering package[44]. Complete and Euclidean methods were used for linkage calculation. Variants within a Euclidean distance of 125 or less were regarded as being in a cluster. Clusters that cannot withstand together in a population (sum of allelic frequencies exceed 100%) were regarded as sub-clusters.

**Pyruvate transporter assay.** The pyruvate uptake function of a cell was determined with a method used in previous studies with slight modifications[17,18]. *Escherichia coli* cells were grown in LB medium at 37 °C for 8 h. The cells were washed twice with M9 pyruvate medium (M9 minimal medium supplemented with 2 g l$^{-1}$ of pyruvate). Then, the washed cells were grown overnight in M9 pyruvate medium at 37 °C with an initial $OD_{600 \, nm}$ of 0.05. Overnight cultures were washed twice with M9 sorbitol medium (M9 minimal medium supplemented with 2 g l$^{-1}$ of sorbitol) and inoculated in 30 ml of M9 sorbitol medium with initial $OD_{600 \, nm}$ of 0.05. The culture was grown for one doubling ($OD_{600 \, nm}$ of 0.1) at 37 °C. Then, each 400 µl of the grown culture was transferred into a 48-well microplate and 3-FP was added to 1 mM final concentration. The plate was incubated at 37 °C with constant double orbital shaking (5 mm of amplitude) and $OD_{600 \, nm}$ was recorded using a Synergy H1 microplate reader (BioTek).

**Intracellular and extracellular pyruvate assay.** Cells were grown in M9 glucose medium at 37 °C. Then, 1 ml of culture was centrifuged briefly and the supernatant was collected for extracellular pyruvate. Intracellular pyruvate sample was prepared as described in the previous study[45]. Briefly, 5 ml of culture was collected and immediately quenched with 5 ml of quenching solution (40% ethanol (vv$^{-1}$) 0.8% NaCl (wv$^{-1}$)) pre-chilled to −35 °C. Quenched cells were collected when the temperature reached −5 °C by centrifugation at −11 °C, 3400 × g for 10 min. Then, the cell pellet was resuspended with 500 µl of methanol pre-chilled to −80 °C. The resuspension was flash frozen and thawed three times with liquid nitrogen to lyse cells. Lysed cells were centrifuged at 4 °C, 10,000 × g for 2 min and the supernatant was collected. Methanol extraction was repeated for the remaining pellet to extract pyruvate completely. The two methanol extracts were combined. Pyruvate concentration was measured using an EnzyChrom™ Pyruvate Assay Kit (Bioassay Systems) according to the manufacturer's protocol.

**Construction of MS56 Δ21kb and ΔrpoS strain.** Target regions were knocked out using lambda recombination of MS56[46]. A kanamycin resistance cassette was PCR amplified from pKD13[46] using primers with homology to the target region. The DNA cassette (1 µg) was introduced to electrocompetent MS56 harboring pKD46 via electroporation on MicroPulser (Bio-Rad) with 1.8 kV in a 0.1-cm gap cuvette and a recipient was selected by incubating at 30 °C on LB agar medium containing 50 µg ml$^{-1}$ of kanamycin. The electrocompetent MS56 was prepared by washing MS56 (carrying pKD46) grown to $OD_{600 \, nm}$ of 0.4 at 30 °C in 50 ml of LB medium containing 10 mM of arabinose three times with ice-cold 10% glycerol. The knockout was confirmed by Sanger sequencing. To remove the kanamycin resistance cassette, pCP20 was introduced to the strain by electroporation. A clone carrying pCP20 was selected by incubating 30 °C on LB agar medium containing 34 µg ml$^{-1}$ of chloramphenicol. A clone carrying pCP20 was then inoculated in 3 ml of liquid LB medium and incubated at 42 °C for overnight with agitation. The overnight grown culture was streaked on LB agar medium and few colonies were subjected to replica plating on LB agar plates containing ampicillin or kanamycin. A colony sensitive to both kanamycin and ampicillin is free of the plasmids and kanamycin resistance cassette and thus propagated further.

**Western blot analysis.** Cell lysate containing 40 µg of protein was separated by sodium dodecyl sulfate (SDS)-polyacrylamide gel electrophoresis. Separated protein was transferred to polyvinylidene difluoride membrane using a Trans-blot Turbo Transfer System (Bio-Rad). Membranes were incubated for 2 h in blocking buffer (5% skimmed milk in Tris-buffered saline with 0.1% Tween-20; TBST) with gentle shaking at room temperature. Blocked membranes were washed three times with TBST, each for 10 min of gentle shaking at room temperature. Appropriate primary antibody (anti-RpoD mouse IgG; NeoClone; Cat. #663202; Lot. #B193929 or anti-MutS rabbit polyclonal; GeneCheck; Cat. #GC-M001; Lot. #219) diluted to 20 µg ml$^{-1}$ of final concentration in dilution buffer (1% skimmed milk in TBST) was added on washed membranes. Membranes were incubated at 4 °C overnight with gentle shaking. Then, membranes were washed three times with TBST and an appropriate secondary antibody (horseradish peroxidase (HRP)-conjugated goat anti-mouse IgG antibody (Thermo), Cat. #31430; or HRP-conjugated goat anti-rabbit IgG (Thermo), Cat. #31460) with a final concentration of 80 ng ml$^{-1}$ was applied for 2 h at room temperature with gentle shaking. Bands were detected with Pierce ECL Plus Western Blotting Substrate (Thermo, Cat. #32132) using a ChemiDoc Imaging System (Bio-Rad).

**Construction of the eMS57mutS+ strain.** The native *mutS* locus was PCR amplified from MG1655 genomic DNA using MutS_Tn_F and MutS_neo_R primers. The amplified *mutS* DNA fragment was connected with a kanamycin resistance DNA cassette of pKD13 (PCR amplified with primers MutS_neo_F and Neo_Tn_R) using overlap extension PCR (OE-PCR) as follows: 5 ng each of MutS cassette and kanamycin cassette were first ligated by 15 cycles of the following PCR reaction: 96 °C for 30 s, 56 °C for 30 s, and 72 °C for 3 min. Ligated product was amplified in the same tube by adding two outermost primers (MutS_Tn_F and Neo_Tn_R) and 20 cycles of PCR reaction: 96 °C for 30 s, 56 °C for 30 s, and 72 °C for 4 min. Ligated product was cloned into pMOD3 with EcoRI and XbaI restriction ligation, generating pMOD3-MutS plasmid in *E. coli* BW25113 strain. Transposon DNA containing mosaic end (ME) sequence was PCR amplified from

pMOD3-MutS using ME_plus_3′ and ME_plus_5′ primers. Then, 50 ng of transposon DNA and 1 U of EZ-Tn5 transposase (Epicentre) were mixed and incubated at room temperature for 30 min to construct a transposome complex, which was introduced into eMS57 and a transposed clone was selected by a kanamycin-selective medium. Expression of MutS was confirmed by quantitative reverse transcription-PCR (qRT-PCR) and western blotting (Supplementary Figure 5). A clone with the most similar expression level of MutS, when compared to MG1655, was chosen and termed as eMS57mutS+. The location of transposon insertion was determined from semi-random PCR using Tn_confirm_F and Random_R1 primers. The eMS57mutS+ had the *mutS* expression cassette inserted in putrescine symporter, PuuP (Supplementary Figure 5).

**Introduction of point mutation on MS56 genome**. Homologous lambda recombination was used to introduce point mutations in the MS56 genome[46]. The OE-PCR product of the kanamycin resistance gene cassette (amplified from pKD13[46]) and target gene (amplified from MS56 or eMS56) was introduced into MS56 harboring pKD46[46] using electroporation as described above. Colonies grown on a kanamycin-selective medium were isolated and confirmed by Sanger sequencing. Constructed strains possessing a wild-type or mutant gene with the kanamycin cassette were compared.

**TaqMan assay coupled with dPCR**. The concentration of genomic DNA was measured by NanoDrop 2000 Spectrophotometer (Thermo) and diluted to a concentration of $20,000-30,000$ copies ml$^{-1}$. Target genes were amplified with QuantStudio 3D Digital PCR System (Thermo) as follows. Each PCR reaction was set up total 20 μl reaction volume, comprised of 10 μl of 2× QuantStudio 3D Digital PCR Master Mix (Thermo), 0.5 μl of 40× TaqMan Probe/Primer Mix (8 μM each of reporters and 36 μM each of primers, Supplementary Table 2), and 1 μl of diluted gDNA. Mixed PCR reaction was dispensed into a QuantStudio 3D Digital PCR 20 K Chip (Thermo) according to manufacturer's instruction using a QuantStudio 3D Digital PCR Chip Loader (Thermo). After dispensing, the chip was filled with QuantStudio 3D Digital PCR Immersion Fluid (Thermo) and sealed with a QuantStudio 3D Digital PCR Chip Lid. PCR was performed by flat block thermal cycler GeneAmp PCR System 9700 (Thermo) equipped with a QuantStudio™ 3D Digital PCR Chip Adapter Kit (Thermo) and QuantStudio 3D Tilt Base (Thermo) according to the following program: 96 °C for 10 min, 56 °C for 2 min, 98 °C for 30 s, repeat the last two steps for 44 cycles, and 56 °C for 2 min for final extension. The image of the chip was read using a QuantStudio 3D Digital PCR instrument (Thermo) and analyzed by QuantStudio 3D AnalysisSuite Cloud Software (Thermo, https://apps.thermofisher.com/quantstudio3d/). Raw data were extracted and visualized in Microsoft Excel 2010 (Microsoft) and Adobe Illustrator CS6 (Adobe) without modifying data integrity.

**Chromatin immunoprecipitation sequencing**. *Escherichia coli* strains MG1655 and eMS57 were grown in 50 ml of M9 minimal medium at 37 °C with agitation. Cells were sampled at mid-log phase (OD$_{600 nm}$ was approximately 0.55 and 0.50 MG1655 and eMS57, respectively). Then, 1.4 ml of 37% formaldehyde solution was added to 50 ml of culture and incubated for 25 min at room temperature with gentle shaking. Next, 2 ml of 2.5 M glycine solution was added to fixed cell culture and incubated for 5 min at room temperature with gentle shaking. The culture was washed three times with 50 ml of ice-cold TBS. After centrifugation at 4 °C, 3000 × *g* for 15 min, all the supernatants were removed and the cell pellet was resuspended with 1.5 ml of lysis buffer (10 mM Tris-HCl, pH 7.5, 100 mM NaCl, and 1 mM EDTA). Forty microliters of Protease Inhibitor Cocktail (freshly prepared as per the manufacturer's manual, Sigma) and 0.3 μl (10,500U) of Ready-Lyse Lysozyme (Epicentre) were added and incubated at 37 °C for 2 h with gentle rotation. Subsequently, 1.65 ml of IP buffer (100 mM Tris-HCl, pH 7.5, 200 mM NaCl, 2% Triton X-100, and 1 mM EDTA) was added and the sample was incubated at 4 °C with gentle rotation for 1 h. Genomic DNA was sheared by sonicating for a total of 2 min (amplitude was 13%, on sonic for 20 s, and resting for 40 s, 6 cycles) using a Sonic Dismembrator Model 500 (Fisher Scientific) equipped with micro-tip (diameter of 3 mm). A total of 700 μl homogenized sample was used for immunoprecipitation. To this, 12 μg of anti-RpoD mouse IgG (NeoClone, Cat. #663202, Lot. #B193929), anti-c-Myc mouse IgG (Santa Cruz Biotechnology; Cat. #sc-40; Lot. #B1313), or normal mouse IgG (Millipore, Cat. #12-371, Lot. #DAM1774722) was added individually and incubated overnight at 4 °C with gentle rotation. Antibody-coupled chromatin solution was added to 50 μl of Dynabeads Protein A (Life Technologies) for anti-RpoD mouse IgG sample or Dynabeads Pan Mouse IgG (Life Technologies) for anti-c-Myc mouse IgG or normal mouse IgG sample pre-washed with 1 ml of ice-cold bead washing solution (0.5% bovine serum albumin in phosphate-buffered saline (PBS)), and incubated at 4 °C for 6 h with gentle rotation. Beads were pooled down with sequential steps on a magnetic stand with washing buffer I (50 mM Tris-HCl (pH 7.5), 140 mM NaCl, 1% Triton X-100, and 1 mM EDTA) twice, washing buffer II (50 mM Tris-HCl (pH 7.5), 500 mM NaCl, 1% Triton X-100, and 1 mM EDTA), washing buffer III (10 mM Tris-HCl, pH 8.0, 250 mM LiCl, 1% Triton X-100, and 1 mM EDTA), and TE buffer. Two hundred microliters of elution buffer (50 mM Tris-HCl, pH 8.0, 1% SDS, and 1 mM EDTA) was added to washed beads and incubated overnight at 65 °C for elution and reverse crosslinking. Beads were pooled down by magnetic stand and

cleared eluate was isolated. Eluate was purified using the MinElute PCR Purification Kit (Qiagen) after treatment with 100 μg of RNase A (Qiagen) and 80 μg of Protease K (Invitrogen). Purified DNA was subjected to qPCR or ChIP-Seq library preparation. A sequencing library was prepared using a NEXTflex Illumina ChIP-Seq Library Prep Kit (Bioo Scientific) according to the manufacturer's instruction. Constructed sequencing libraries were quantified using a Qubit dsDNA HS Assay Kit (Thermo) with a Qubit 2.0 fluorometer (Thermo). Quality of libraries was analyzed using TapeStation 2200 (Agilent) equipped with High Sensitivity D1000 Screen Tape (Agilent). The sequencing libraries were sequenced using a MiSeq Reagent Kit v2 (Illumina) with 50 cycle single-ended reaction in the MiSeq instrument (Illumina).

**Heterologous expression of native and mutant σ$^{70}$**. Native or mutant *rpoD* was PCR amplified from MG1655 or eMS57 genomic DNA, respectively, using rpoD_F and rpoD_R primers. Plasmid backbone was also PCR amplified from pTrcHis2A plasmid (Invitrogen) using pTrc_inv_F and pTrc_inv_R primers. Two PCR products were combined using In-Fusion HD Cloning Kit (Takara) as per the manufacturer's instructions. Two tandem c-Myc epitope was fused at the N terminus of *rpoD* by rpoD_R and pTrc_inv_F primers.

**Transcriptome sequencing (RNA-Seq)**. Cell culture samples (10 ml) were harvested at the mid-log growth phase (OD$_{600 nm}$ ~0.55 for MG1655, ~0.50 for eMS57). Total RNA was isolated using the RNASnap™ method[47]. Briefly, the harvested cell pellet was resuspended with 100 μl of RNASnap solution (18 mM of EDTA, 0.025% of SDS, 1% of β-mercaptoethanol, and 95% of formamide). The resuspension was incubated at 95 °C for 7 min and centrifuged at $16,000 \times g$ for 5 min. A clear supernatant was transferred to a new microcentrifuge tube and RNA was extracted by ethanol precipitation. Total RNA (5 μg) was treated with 2 U of DNase I (NEB) for 30 min at 37 °C to remove residual DNA contaminants. DNase I-treated RNA samples were purified with phenol–chloroform–isoamyl alcohol extraction followed by ethanol precipitation. Ribosomal RNA was removed from 2 μg of purified total RNA samples using a RiboZero rRNA (ribosomal RNA) Removal Kit (for bacteria, Illumina) according to the manufacturer's instruction. RNA-Seq libraries were constructed from rRNA-subtracted RNA using a TruSeq Stranded mRNA LT Sample Prep Kit (Illumina) according to the manufacturer's protocol. Constructed sequencing libraries were quantified using a Qubit dsDNA HS Assay Kit (Thermo) with a Qubit 2.0 fluorometer (Thermo). Quality of libraries was analyzed using a TapeStation 2200 (Agilent) equipped with High Sensitivity D1000 Screen Tape (Agilent). The sequencing libraries were sequenced using a MiSeq Reagent Kit v2 (Illumina) with 50 cycles single-ended reaction in the MiSeq instrument (Illumina).

**qPCR and qRT-PCR**. RT was performed from 1 μg of total RNA in 20 μl reaction using the SuperScript III First-Strand Synthesis System (Thermo) according to the manufacturer's instruction. Briefly, 1 μg of DNA-subtracted RNA, 50 ng of Random Hexamer, 1 μl of 10 mM dNTP mix, and diethyl pyrocarbonate-treated water to bring reaction volume to 10 μl were mixed in RNase-free PCR tube. The mixture was incubated at 65 °C for 5 min and placed on ice immediately after incubation. Ten microliters of cDNA Synthesis Mix (2 μl of 10× RT buffer, 4 μl of 25 mM MgCl$_2$, 2 μl of 0.1 M dithiothreitol, 1 μl of RNaseOUT, and 200 U of SuperScript III RT) was added and incubated at 25 °C for 10 min. Then, the mixture was incubated 50 °C for 50 min followed by 85 °C for 5 min. One microliter (2U) of *E. coli* RNase H was treated at 37 °C for 20 min to remove RNA. qPCR was performed in 20 μl reaction (10 μl of iQ SYBR Green Supermix (Bio-Rad), 10pmol of forward and reverse primers, 1 μl of cDNA or immunoprecipitated DNA) with the following conditions: 40 cycles of 95 °C for 10 s, 58 °C for 30 s, and 72 °C for 30 s. Reactions were monitored on a C1000 Thermal Cycler (Bio-Rad) equipped with a CFX96 Real-Time PCR Detection System (Bio-Rad). All the primers were designed by Primer-BLAST[48] and there was no non-specific binding found in *E. coli* K-12 MG1655 genome sequence (Acc. NC_000913.3). Sequence of primers and size of amplicons are summarized in Supplementary Table 2. In ChIP-qPCR experiment, promoter region (peak region) was targeted for amplification and ΔΔCq method was used for quantification using geometric mean of four reference peaks (*hpt*, *nrdR*, *yebS*, and *yecD*) as an internal reference point[49].

**Differential RNA-Seq and KEGG pathway enrichment analysis**. Genes with a P value lower than 0.01 (two-sided Welch's *t* test) were subjected to enrichment analysis of differential expression[44]. The analysis was performed using ClueGO plug-in (v2.2.4) of Cytoscape (v3.3.0).

**Cellular NADH/NAD$^+$ and NADPH/NADP$^+$ assay**. NADH/NAD$^+$ and NADPH /NADP$^+$ ratios were determined using a NAD/NADH Quantification Kit (Sigma-Aldrich) and NADP/NADPH Quantification Kit (Sigma-Aldrich), respectively, according to the manufacturer's instruction. Briefly, $2 \times 10^9$ and $5 \times 10^9$ cells were collected for NADH/NAD$^+$ and NADPH/NADP$^+$ assays, respectively. Samples were washed twice with ice-cold PBS buffer and resuspended with NAD/NADH or NADP/NADPH extraction buffer. The samples were flash frozen in liquid nitrogen and ground by a pestle and mortar. The lysates were collected and divided into two conical tubes. One was incubated at 60 °C for 30 min to deplete

NAD(P)$^+$. Then, Cycling Enzyme Mix was applied at room temperature for 5 min to convert NAD(P)$^+$ to NAD(P)H. Absorbance at 450 nm was measured after adding NAD(P)H Developer using a Synergy H1 microplate reader (BioTek).

**Cellular ATP assay**. Intracellular ATP concentration was determined using an ATP Bioluminescence Assay Kit HS II (Roche) according to the manufacturer's instruction. First, $5 \times 10^8$ cells were collected and washed twice with ice-cold PBS buffer. Then, the pellet was resuspended in 500 μl of Dilution Buffer and lysed by Cell Lysis Reagent. The samples were transferred to a Nunc F96 MicroWell Black Polystyrene Plate (Nunclon). Luminescence was measured using a Synergy H1 microplate reader immediately after adding Luciferase Reagent.

**Ribosome profiling**. Ribosome profiling was conducted using a method described in the previous report[50] without tRNA removal step. Briefly, 50 ml of E. coli culture was collected after 5 min treatment of chloramphenicol (34 mg ml$^{-1}$) at an exponential growth phase. Cells were flash frozen with 0.5 ml of lysis buffer (1% Triton X-100, 34μgml$^{-1}$ chloramphenicol, 133 mM of NaCl, 4.75 mM of MgCl$_2$, and 19 mM of Tris-HCl, pH 7.5) and lysed by pestle and mortar. Then, the supernatant containing 10μg of RNA treated with 2000 gel units of Micrococcal Nuclease (NEB). Polysomes were recovered from MNase-digested sample using Illustra MicroSpin S-400 HR Columns (GE Healthcare) followed by phenol:chloroform:isoamyl alcohol extraction. Ribosomal RNA was removed from 5μg of polysome-protected RNA using a RiboZero rRNA Removal Kit (Illumina) according to the manufacturer's instruction. rRNA-subtracted RNA samples were phosphorylated by treating 10U of T4 Polynucleotide Kinase (NEB) at 37 °C for an hour and purified with RNeasy MinElute columns (Qiagen). Sequencing libraries were prepared from phosphorylated RNA samples using the NEBNext Small RNA Library Prep Set for Illumina (NEB) according to the manufacturer's protocol. Next-generation sequencing was performed by the ChunLab (Seoul, South Korea) with high output mode using V4 sequencing-by-synthesis reagent on a HiSeq 2500 instrument.

**Measurement of protein production via flow cytometry**. Cells harboring high copy mRFP1 expression plasmid, BBa_J04450-pSB1C3, were grown aerobically in M9 glucose medium for 12 h at 37 °C, with 0.5 mM of isopropyl β-D-1-thioga-lactopyranoside induction. Then, 1 ml of the cell culture was diluted in 9 ml of PBS and cells were dissociated using a round bottom polystyrene test tube with cell strainer snap cap (Corning). Then, samples were analyzed on an S3e Cell Sorter (Bio-Rad). A total of 100,000 events were collected and analyzed by the FlowJo software (FlowJo). Contour plots showing gating strategy are provided in Supplementary Figure 19.

**Data processing**. Sequencing data was processed on a CLC Genomics Workbench (CLC Bio). Raw reads were trimmed using a Trim Sequence Tool in NGS Core Tools with a quality limit of 0.05. Reads with more than two ambiguous nucleotides were discarded and the quality trimmed reads were mapped on the MG1655 reference genome (NC_000913.3) or MS56 genome sequence (http://cholab.or.kr/data/) with the following parameters: mismatch cost: 2, indel cost: 3, length and similarity fractions: 0.9. For RNA-Seq and ribosome profiling, reads were mapped strand specifically (backward). Reads mapped to the multiple genomic positions were mapped randomly for resequencing and ChIP-Seq and discarded in RNA-Seq and ribosome profiling analysis. Sequence variants were detected using a Quality-based Variant Detection Tool with the following parameters: neighborhood radius: 5; maximum gap and mismatch counts: 5; minimum neighborhood and central qualities: 30; minimum coverage: 10; minimum variant frequency: 10%, and maximum expected alleles: 4. Non-specific matches were ignored and bacterial genetic code was used. Variants in repeat region (i.e., rRNAs and transposases) were discarded. Mutation rate (mutations per genome per generation) was calculated as the number of mutation (including SNVs, MNVs, and indels) divided by cumulative generation. Binding peaks of RpoD were detected from a mapping file using the Model-based analysis of ChIP-Seq (MACS) software with shiftsize of 50[51]. Detected peaks were compared with RpoD binding regions in the RegulonDB database[30]. A promoter motif was found using the MEME Suite software (v4.11.4)[52]. Expression and RPF levels were normalized by reads per kilobase per million mapped reads (RPKM) metric. The mapping file was exported as BAM file format and converted to GFF file format and visualized by the SignalMap software (v2.0.0.5, Roche). TE was obtained by dividing RPF by RNA expression. In the meta-analysis of a ribosome profile, either 5′ or 3′ ends were tested to determine a position of ribosome (Supplementary Figure 17)[53,54]. 5′ Assignment method was used for the meta-analysis, because the method provides clearer 3-nt codon periodicity of translation than 3′ assignment.

**Determination of σ$^{70}$ binding peaks from ChIP-Seq**. ChIP-Seq detects DNA fragments that crosslinked to the target protein. The MACS software was used to determine the binding sites of σ$^{70}$[51]. After analysis with MACS, we obtained 1062 and 1089 raw binding sites from MG1655 and eMS57 samples. The summit of each binding signal was defined as a peak. The raw reads were further adjusted with the RegulonDB database for maximum accuracy and precision[30]. In adjustment, all of the raw binding sites were compared with 2055 RegulonDB binding sites. However, we had to determine a certain length threshold that determined whether two binding sites are the same. Thus, various lengths of sliding windows ranges from 10 to 150nt were examined (Supplementary Figure 8). As the length of the sliding window increases, the number of binding sites retrieved increased. The number of retrieved binding sites divided by the number of raw binding sites detected by MACS (termed retrieval rate hereafter) reached a plateau at a sliding window of 80nt or longer. Simultaneously, the number of peaks that were bound by both wild-type and mutant σ$^{70}$ were compared (Supplementary Figure 8). At a sliding window of 80nt, the ratio of shared peaks to total peaks was 0.381 and reached a plateau. The ratio remained unchanged (0.407 at a sliding window of 10,000nt). A starting point of the plateau (80nt) was used as a threshold. Using the threshold, we determined a total of 421 and 418 binding sites from MG1655 and eMS57, respectively (Fig. 3a). Among them, 320 were overlapped.

**Normalization of RNA expression level and RPFs**. RNA expression and RPFs were normalized by the RPKM metric[55]. Although there remains some controversy related to RPKM transformed expression[56], recent RNA-Seq analysis pipelines could not be applied on two significantly different genomes. The pipelines, such as DESeq2, assume that most of the genes are not differentially expressed[57]. With thorough inspection of the data, we decided to transform transcriptomic data into RPKM and genes with a wide range of expression were validated with qRT-PCR (Supplementary Figure 10). Furthermore, we believe this method is more reliable when calculating translational efficiency from RNA expression and RPFs that are expressed in the same unit.

**Statistical analyses**. All bacterial growth measurements were biologically triplicated and their differences were examined by two-sided t test of unequal variance (Welch's t test). Transcriptional change according to promoter category was tested by Wilcoxon's rank-sum test (Fig. 3). Welch's t test was used for DEG determination across biologically duplicated sequencing results. All statistical analyses were performed using the statistical analysis of SciPy package[44].

**Reporting summary**. Further information on experimental design is available in the Nature Research Reporting Summary linked to this article.

## Data availability
Next-generation sequencing data generated during the current study are available in the EMBL Nucleotide Sequence Database (ENA) with primary accession number PRJEB21199. The source data underlying Figs. 1a–d, 1f–h, 2b–c, 2f–h, 3d–e are provided as a Source Data file. All other data are available from the authors upon reasonable request.

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

## Acknowledgements

This work was supported by the Intelligent Synthetic Biology Center of the Global Frontier Project (2011-0031957 to B.-K.C. and 2011-0031955 to S.C.K.), the Basic Core Technology Development Program for the Oceans and the Polar Regions (2016M1A5A1027458 to B.-K.C), and the Basic Science Research Program (2018R1A1A3A04079196 to S.C. and 2018M3A9H3024759 to B.-K.C.) through the National Research Foundation of Korea (NRF) funded by the Ministry of Science and ICT. This work was funded by a grant from the Novo Nordisk Foundation (to B.-K.C.). We thank Marc Abrams for editing the manuscript.

## Author contributions

B.-K.C. designed and supervised the project; D.C., J.H.L., M.Y., S.H., B.H.S. and S.C. performed experiments; D.C., S.C., B.P. and B.-K.C. analyzed the data; D.C., S.C., B.P., S.C.K. and B.-K.C. wrote the manuscript. All authors read and approved the final manuscript.

## Additional information

**Competing interests:** The authors declare no competing interests.

