## [Peer Review File · Nature Communications]

Reviewers' Comments:

Reviewer #1:

Remarks to the Author:

Manuscript ID: Nature Communications manuscript NCOMMS-18-09484-T

Authors: Donghui Choe, Jun Hyoung Lee¹, Minseob Yoo, Soonkyu Hwang, Bong Hyun Sung, Suhjung Cho, Bernhard Palsson, Sun Chang Kim, Byung-Kwan Cho

This manuscript describes about the systems analysis of about 1.1 Mbp reduced size of *Escherichia coli* genome, which showed impaired growth under M9 minimal medium condition, though kept all of essential genes for survival in the medium condition used. To understand this unpredicted phenotypes by removal of non-essential genes, the authors developed adaptive laboratory evolution (ALE) method to re-optimize growth performance of a genome-reduced strain during over 800 generation time period. They comprehensively performed analyses of phenotypic changes, such as nutrient utilization for carbon, nitrogen, phosphorus and sulfur sources by BIOLOG phenotype microarray technology, identification of causal mutation by re-sequencing of genomes, monitoring the alteration of transcriptome and translome by RNA seq and Ribosomal profiling of strains under adaption at every five days, respectively. From their analyses, their interpretation of growth recovery during adaptation is caused by metabolic remodeling of the transcriptome and translome in adapted strain, eMS57.

The authors showed comprehensively prepared solid experimental data. I am sure that their analysis will give great benefits to the readers of the journal.

I think it is clearly beneficial to be published.

Before publication, I would like to ask the authors to consider comments below;

Comments;

1) How many and which family of TFs were deleted in MS56 genome? No consideration or discussions about their deletion is required?

2) Only MS56 was chosen as the starting strain for ALE analysis. Why the authors did not try to compare the ALE analysis between different reduced genome strains? All of growth impaired reduced genome strains have the similar route to adapt the environmental condition like *sufD*, *rpoA*, *rpoD*, etc.? I think the initial introduction of mutation might be a kind of trigger with very stochastic event, is this wrong?

During adapted evolution, series of introduction of mutation might also be stochastic events. Even though those mutation happen randomly, the adaptation by introducing mutation have a direction reaching to a certain optimal metabolic network point, which shows good balance to survive in the environmental condition used for the analysis. I think the route to get the optimal point might also be random, though the final goal share the same or similar point. So, my estimation is that, different starting strains may show different routes for adaptation but all of the adapted strains may share the similar metabolic feature. Is this idea totally wrong?

One possibility might be that the initial introduction of mutation may stochastic but during adaptation, though lineage of adaptation may be varied, the final optimal metabolic balance might be very narrow range to adapt the environmental condition. To prove this, I think one option is to compare the ALE analysis between the different starting mutant strains.

In other word, alteration of deoxynucleotide biosynthesis leading to the new balance of cellular concentration of ATP, NADH, NADPH and Pyruvate, which showed in this manuscript, might be a goal for adaptation as an optimal point to the environmental condition?

3) Translational buffering might be caused by the balance between the cellular amount of functionally active transcriptional and translational machineries. The functionally active cellular concentration of these factors may have an important information.

4) In *E. coli* during stationary growing phase, ribosomes form 100S inactive complex. For translational buffering, is there any mutation during adaptation involved in the regulation of translational activities such as *rmf* gene, which function to regulate the formation of inactive 100S ribosome?

5) Figure 1c, grey bar might mean supplementation percentage of LB. Description in the legend is not sufficient for easy reading. It is required to check whole manuscript to give minimum but sufficient description, especially for figure legends.

6) The similar problem as 5) in Figure 2e – j, it is not easy to distinguish colors of spots. And more explanation in the legend is required. What is FAM and VIC?

7) I felt too many sub-figures in each of figures in the main text. It makes this manuscript not easy to read. I think it is one option to focus on the essential sub-figures and the rest of those might be moved into the supplementary information. Reconsideration may improve this manuscript easy to understand.

Reviewer #2:

Remarks to the Author:

General:

In the manuscript of Choe and coworkers an adaptive laboratory evolution (ALE) study with a previously constructed genome-reduced E. coli strain is described. During ALE the growth performance of this strain on M9 minimal medium could be restored to wild-type level at low cell densities. Subsequent multi-omics analysis revealed many changes on the genomic and metabolic level and the authors did a great deal on speculating about their origin. In my opinion, the paper is mostly descriptive and does not provide any new fundamental insight into the core metabolism of E. coli. Hence, it does not deliver a blueprint for an optimal minimal genome of this model organism.

Major remarks:

- L62, L38: "...we implement ALE to allow for self-optimization of the unknown processes encoded on a genome." The reduced genome still contains unknown genes those functions can be self-optimized, but no clear functional annotation is provided after ALE. To be considered as an optimal minimal genome, at least, one functional annotation for each underlying gene should be provided.

- L73: "Optimal minimal genome" - For what biotechnological purpose?

- L136: What else is missing to recovery the growth of eMS57?

- L39: The high number of mutations is not surprising for such a long ALE experiment. In my opinion, however, the whole approach has a major flaw. As a reference case, wild-type E. coli should be evolved under the same conditions. Then differential genome, transcriptome and metabolome analysis can be applied to unravel the specific responses of MS56 in its genome-reduced background.

- p7ff: "It is unlikely ..."; "Rather, it is more plausible..."; "However, it is unlikely ..."; "Instead, it is more likely..."; "Thus, we hypothesized that..."; "It is unclear how ..."

These are a few examples on how the authors speculate about the meaning of their results throughout the whole text. In the end, the reader is left with many open questions.

L211, L281: This application for eMS57 remains to be shown! All experiments were performed at very low cell densities (cmp. Fig. 1), which are not relevant for industrial production.

L320: This kind of study can be expect for this journal!

L347: "The clearest overall lesson is that growth retardation in genome-reduced strains results from a metabolic imbalance; a systemic function." This sentence sums up the weakness of this study.

Specific comments:

- Please provide a table with all plasmids and major strains used and generated in this study including relevant characteristics and references. This would help a lot to follow your experiments and understand the ancestry of the corresponding strains.

- In some cases very few explanations in the figure legends makes it difficult to understand figures without further information.

- Discussion is in major parts a repetition of the results. Please compare your results e.g. with the outcome of other ALE projects.

- L43: To my knowledge, the genome size of native *M. mycoides* is about 1.08 Mbp. Please check.
- L86: What were the criteria for selecting these 55 regions for deletion?
- L95: Was the ALE performed in independent replicates? Can the results be reproduced in another ALE experiment? Please provide more details for the ALE experiment in the methods section, e.g. culture volume, passage automatically or manually,...
- L98: How was eMS57 isolated?
- L103: Please provide experimental details for TEM and SM pictures.
- L105-114: Are these Biolog-experiments? If the data is based on one replicate only, it might not be feasible to draw the given conclusions without further experimental proof. Moreover, with the Biolog-system metabolic activity is measured, not growth!
- L114ff: What was the reason to measure pyruvate in the supernatant of cultures?
- L149-164: Have mutations in *rpoA* or *rpoD* been identified in other ALE experiments?
- L196-197: How was the mutation rate determined?
- L103-205: How was the performance of the two further evolved strains? Was the strain with more mutations better than the strain with less mutations?
- L237-238: Why should a generally increased transcription of sigma70-dependent genes enable the adaptation to M9 medium?
- How good is the match between ChIP_Seq and transcriptome data?
- L253: ...eMS57 utilizes glucose via the ED-pathway in part,... ◊ Can you draw this conclusion solely based on the transcriptomic-data?
- L298: MG1655? I understood in the sentence before you measure in eMS57?
- L308-309: How was this measured?
- L406: The Biolog does not measure growth!
- L592-599: Were these experiments performed according to the MIQE-Guidelines?

Reviewer #3:

Remarks to the Author:

This is a nice piece of work regarding a very exciting topic that is the engineering of minimal cells. They are working with an *E. coli* missing 1.1 Mb of non-essential genes which grows poorly in minimal media. The authors have done adaptive evolution to improve its growth rate and they have obtained a new variant that grows efficiently in minimal medium. Then they have proceeded to determine what the genetic changes that improve the growth rate are. They have found a deletion of 21 Kbases plus different mutations in other genes and they have looked at the accompanying changes in transcription and translation.

My main concern is that the authors have found that a single deletion of a gene in that region explains 80% of the improvement in growth rate. This per se is interesting but also raises the question of how many of the changes in other processes or genes explain the remaining 20%. I think it is very important that the authors should do the same 21 Kbase deletion in the MS56 to see if the growth rate increases even further. This is important since if growth rate recovers above 80% with this deletion then many of the conclusions obtained after by the authors could be just mutational noise.

Regarding the mutation in the sigma 70, and the changes in specificity I have my doubts that the differences found are significant looking at the large overlap and at the sequence fingerprints. Ser 253 is not found at any of the Sigma 70 domains 1-4 and therefore it is difficult to see how it will affect specificity of the promoter. To see if this is the case they should do the same mutation in the WT *E. coli* and see if they get the same results.

Finally the part regarding translation is not very clear.

As a conclusion I think it is a nice piece of work but I would like to see more controls about the role of the different mutations they have found. This is one of the problems of adaptive evolution, we can obtain better strains but it is difficult to pinpoint which changes are the responsible.

Point-by-point Response to the Reviewer's Comments

Reviewer #1 (Remarks to the Author):

This manuscript describes about the systems analysis of about 1.1 Mbp reduced size of *Escherichia coli* genome, which showed impaired growth under M9 minimal medium condition, though kept all of essential genes for survival in the medium condition used. To understand this unpredicted phenotypes by removal of non-essential genes, the authors developed adaptive laboratory evolution (ALE) method to re-optimize growth performance of a genome-reduced strain during over 800 generation time period. They comprehensively performed analyses of phenotypic changes, such as nutrient utilization for carbon, nitrogen, phosphorus and sulfur sources by BIOLOG phenotype microarray technology, identification of causal mutation by re-sequencing of genomes, monitoring the alteration of transcriptome and translome by RNA seq and Ribosomal profiling of strains under adaption at every five days, respectively. From their analyses, their interpretation of growth recovery during adaptation is caused by metabolic remodeling of the transcriptome and translome in adapted strain, eMS57. The authors showed comprehensively prepared solid experimental data. I am sure that their analysis will give great benefits to the readers of the journal. I think it is clearly beneficial to be published. Before publication, I would like to ask the authors to consider comments below;

Comments;

1. How many and which family of TFs were deleted in MS56 genome? No consideration or discussions about their deletion is required?

Response: We would like to thank the reviewer for reading our manuscript and for providing insightful comments which helped to strengthen our manuscript. We also thank you for raising an important point regarding the transcription factors deleted in MS56. Among the 304 transcription factors (184 experimentally characterized) in the *E. coli* K-12 strain¹, 63 were deleted in MS56. The 63 TFs are members of 28 transcriptional regulator families (**Supporting Table 1**) and there was no bias in any specific family of TFs (**Supporting Figure 1**).

Supporting Figure 1. Number of transcription factor (TF) families in MG1655 genome and deleted regions. Genomic composition of transcription factor families is presented as blue bars, while that of 63 deleted transcription factors is shown as yellow bars.

Supporting Table 1. Transcription factors deleted in MS56. U: regulon unknown. D: regulon deleted. P: part of regulon deleted. R: regulon remained.

Gene	Family	Description	Type
perR	LysR	putative transcriptional regulator	U
yagI ¹	IcIR	CP4-6 prophage; DNA-binding transcriptional repressor	D
ecpR	LuxR/UhpA	DNA-binding transcriptional dual regulator	D
rclR	AraC/XylS	DNA-binding transcriptional activator	D
abgR	LysR	putative LysR-type DNA-binding transcriptional regulator	D
ydaS ¹	HTH_3	Rac prophage; toxin YdaS; putative DNA-binding transcriptional regulator	R
feaR	AraC/XylS	DNA-binding transcriptional activator	D
paaX	PaaX	DNA-binding transcriptional repressor	D
alpA ¹	AlpA	CP4-57 prophage; DNA-binding transcriptional activator	D
ydfH	GntR	DNA-binding transcriptional repressor	R
cspI ¹	Cold	Qin prophage; cold shock protein	U
cspB ¹	Cold	Qin prophage; cold shock-like protein	U
cspF ¹	Cold	Qin prophage; cold shock protein	U
relE ¹	RelE	Qin prophage; mRNA interferase toxin	D
relB ¹	RelB	Qin prophage; mRNA interferase toxin	D
dicC ¹	DicC	Qin prophage; DNA-binding transcriptional regulator for	D
dicA	HTH_3	DNA-binding transcriptional dual regulator	D
yjhU ¹	Sugar binding	KpLE2 phage-like element; putative DNA-binding transcriptional regulator	U
yjhI ¹	IcIR	KpLE2 phage-like element; putative DNA-binding transcriptional regulator	U
sgcR ¹	DeoR	KpLE2 phage-like element; putative DNA-binding transcriptional regulator	U
glcC	GntR	DNA-binding transcriptional dual regulator	D
ymfL ¹	YmfL	e14 prophage; uncharacterized protein	U
bluR	MerR	DNA-binding transcriptional repressor	D
ybcM ¹	AraC/XylS	DLP12 prophage; putative DNA-binding transcriptional regulator	U
appY	AraC/XylS	DNA-binding transcriptional activator	P
nhaR	LysR	DNA-binding transcriptional activator	P
pgrR	LysR	DNA-binding transcriptional repressor	P
flhC	FlhC_like	DNA-binding transcriptional dual regulator	P
flhD	FlhD_like	DNA-binding transcriptional dual regulator	P
hypT	LysR	HypT-[Met]reduced	P
yjiR	GntR	fused putative DNA-binding transcriptional regulator/putative aminotransferase	U
lgoR	GntR	putative DNA-binding transcriptional regulator	U
ogrK ¹	OgrK	prophage P2 late control protein	U
cspH	Cold	stress protein, member of the CspA family	U
cspG	Cold	cold shock protein	U
torR	OmpR	DNA-binding transcriptional dual regulator	P
rutR	TetR/AcrR	DNA-binding transcriptional dual regulator	P
putA	PutA	fused DNA-binding transcriptional repressor/proline dehydrogenase/1-pyrroline-5-carboxylate dehydrogenase	D
csgD	LuxR/UhpA	DNA-binding transcriptional dual regulator	P
yahA	LuxR/UhpA	DNA-binding transcriptional activator/c-di-GMP phosphodiesterase	D
yahB	LysR	putative LysR-type DNA-binding transcriptional regulator	U
prpR	EBP	DNA-binding transcriptional dual regulator	D
cynR	LysR	DNA-binding transcriptional dual regulator	D
lacI	GalR/LacI	DNA-binding transcriptional repressor	D
mhpR	IcIR	DNA-binding transcriptional activator	D
hyfR	EBP	DNA-binding transcriptional activator	D
atoC	EBP	DNA-binding transcriptional activator/ornithine decarboxylase inhibitor	D

yidZ	LysR	putative LysR-type transcriptional regulator	U
alsR	RpiR	DNA-binding transcriptional repressor	D
phnF	GntR	putative transcriptional regulator	U
mngR	GntR	DNA-binding transcriptional repressor	D
yehT	LytTR	DNA-binding transcriptional activator	D
mlrA	MerR	DNA-binding transcriptional activator	P
ybiH	TetR/AcrR	DNA-binding transcriptional dual regulator	D
ybiI	DksA	zinc finger domain-containing protein	U
mntR	DtxR	DNA-binding transcriptional dual regulator	P
cueR	MerR	DNA-binding transcriptional dual regulator	P
allS	LysR	DNA-binding transcriptional activator	D
allR	IclR	DNA-binding transcriptional activator	D
hicB	HTH_3	antitoxin of the HicA-HicB toxin-antitoxin system	U
ydcR	GntR	fused putative DNA-binding transcriptional regulator/putative aminotransferase	U
mcbR	GntR	DNA-binding transcriptional dual regulator	P
yddM	HTH_3	putative DNA-binding transcriptional regulator	U

¹Transcription factors in prophage.

Except for the *dicA* single knock-out strain, we have confirmed the growth of *E. coli* Keio strains (K-12 strain; close relative of MG1655²) with one of the 62 TFs deleted (**Supporting Figure 2**). The *dicA* single knock out strain is not part of the Keio strain collection, as only the $\Delta dicA \Delta dicB$ double knockout strain is viable³. No strains showed significant growth retardation in M9 glucose medium, although the growth rates of the $\Delta ydaS$ and $\Delta abgR$ strains were decreased to 72.4% and 69.8% of that of wild-type *E. coli*, respectively. Growth retardation of $\Delta ydaS$ did not result from *ydaS* deletion⁴. Instead, slower growth occurred because of transcriptional interference caused by the terminator-less kanamycin cassette inserted at the *ydaS* locus, which activated downstream *ydaT*. Overall, no transcription factors were found to induce significant growth defects.

Supporting Figure 2. Growth curve of *E. coli* lacking one of the 62 transcription factors. Growth of 62 knock-out strains in M9 glucose medium was monitored in a 96-well plate on Synergy H1 microplate reader (BioTek). The plate was incubated at 37°C with constant double orbital shaking (5 mm amplitude). WT: *E. coli* K-12 strain BW25113.

Next, we examined the detailed function of each transcription factor. Twenty-two of the 63 TFs were uncharacterized, and thus their effects on transcriptional regulation are unknown (**Supporting Figure 3**). Among the remaining 41 TFs, none were global regulators and 27 were completely deleted along with their regulons (**Supporting Figure 3**); thus, the direct effect of the 27 TFs on the transcriptome may be negligible. Finally, 14 TFs were

deleted while part of their regulons remained (103 genes) de-regulated (**Supporting Figure 3**).

Supporting Figure 3. Transcription factors deleted in MS56 and categorization by regulon.

The 103 regulons were no longer regulated by 14 TFs and only 16 genes were differentially expressed (**Supporting Table 2**). Based on the mode of regulation of the 14 TFs (**Supporting Table 2**), we can predict the behavior of the regulon after deletion of its regulator. The expression levels of 6 genes, *rspB*, *mdh*, *yccT*, *moaA*, *moaB*, and *yciG*, were changed as predicted. RspB is a putative zinc-binding dehydrogenase. Although its function has not been demonstrated, considering that *rspA* (contained in *rspAB* operon) blocks stress-mediated induction of RpoS, RspB was predicted to be related to AHL (acyl homoserine lactone)-mediated quorum sensing⁵. Additionally, considering that the well-characterized lactonase in *Bacillus* is a zinc-binding metalloenzyme and dehydrogenate lactone ring, RspB appears to be a lactonase. Because *rpoS* was deleted in eMS57, its target was no longer present in eMS57. *mdh* encodes a malate dehydrogenase involved in the TCA cycle. Perturbation of central carbon metabolism can cause serious growth retardation. However, FlhC/D deletion did not induce growth retardation, possibly because the expression level of *mdh* is regulated by multiple global regulators such as Crp, DpiA, and ArcA in *E. coli*. *moaAB* are related to molybdenum cofactor (molybdopterin) biosynthesis. Overexpression of *moaA* appears to induce no defects, as reported in a previous study⁶. Thus, de-inactivation of *moaAB* likely causes no difference in MS56. The functions of *yccT* and *yciG* remain unknown.

Supporting Table 2. Expression level of differentially expressed regulons of deleted TFs. Mode of regulation is either activation or repression. Considering transcription factor function as both an activator and repressor (dual regulator), the mode of regulation is defined as previously described (provided above).

TF	Regulon	RNA expression level		p-value	Mode of regulation
		MG1655	eMS57		
YdfH	rspB	53.40	768.26	0.0077	Repression
PgrR	tyrR	219.02	108.74	0.0019	Repression
FlhC/D	napF	2.10	11.90	0.0025	Activation
FlhC/D	ccmF	2.01	9.20	0.0038	Activation
FlhC/D	glpA	0.84	26.25	0.0000	Activation
FlhC/D	glpB	2.64	27.62	0.0077	Activation
FlhC/D	mdh	1066.15	1937.04	0.0002	Repression
HypT	cydA	149.36	357.39	0.0046	Activation
HypT	cydB	129.77	419.88	0.0052	Activation
CsgD	yccT	39.46	9.31	0.0058	Activation
CsgD	nlpA	216.54	925.80	0.0010	Activation
MlrA	rplU	323.04	924.55	0.0045	Activation
MlrA	rpmA	678.43	1367.59	0.0033	Activation
CueR	moaA	105.46	736.61	0.0001	Repression

CueR	moaB	76.89	486.25	0.0080	Repression
McbR	yciG	297.40	6.82	0.0077	Activation

Finally, the expression changes of 10 genes (*tyrR*, *napF*, *ccmF*, *glpAB*, *cydAB*, *nlpA*, *rplU*, and *rpmA*) showed opposite results from the prediction. eMS57 appeared to establish a new transcriptional balance that canceled the effect of TF deletion, but this transcriptional change between MS56 and eMS57 could not be evaluated because MS56 did not grow on M9 glucose medium. We have included a new figure (**Supplementary Fig. 5**) to further illustrate the effects of the 63 transcription factors deleted in MS56 and have discussed these points in the Results and Discussion as the reviewer suggested.

(Page 10, Lines 217 – 219) Additionally, 63 transcription factors were deleted in MS56. Although none of the deletions induced severe growth defects (**Supplementary Fig. 5**), they may have caused transcriptional interference in MS56.

(Page 16, Lines 380 – 385) In addition, we would like to acknowledge that 63 transcription factors, comprised of 28 families, were deleted in MS56. Except for the *dicA* single knock-out strain (which can only be deleted with *dicB*³⁷), we have confirmed no significant growth defect of *E. coli* Keio strains with one of the 62 TFs deleted (**Supplementary Fig. 5**). Even though the growth was unchanged by transcription factor deletions, there must be transcriptional perturbation that interferes with new transcriptomic balance back and forth in eMS57.

(Supplementary Fig. 5)

Supplementary Fig. 5. Growth curve of *E. coli* lacking one of the 62 transcription factors. Growth of 62 knock-out strains in M9 glucose medium was monitored in a 96-well plate on a Synergy H1 microplate reader (Bio-Tek). The plate was incubated at 37°C with constant double orbital shaking (5 mm amplitude). WT: *E. coli* K-12 strain BW25113. Deletion strains were obtained from single gene knockout collection (the Keio collection). *dicA* single knockout strain was not tested, because it is not contained in the Keio strain collection as only the $\Delta dicA \Delta dicB$ double knockout strain is viable. No strain showed significant growth retardation in M9 glucose medium, although the growth rates of the $\Delta ydaS$ and $\Delta abgR$ strains were decreased to 72.4% and 69.8% of that of wild-type *E. coli*, respectively. Growth retardation of $\Delta ydaS$ did not result from *ydaS* deletion (Bindal, G., Krishnamurthi, R., Seshasayee A. S. N., & Rath, D. CRISPR-Cas-mediated gene silencing reveals RacR to be a negative regulator of YdaS and YdaT toxins in *Escherichia coli* K-12. *mSphere* 2, e00483-17, doi: 10.1128/mSphere.00483-17 (2017)).

2. Only MS56 was chosen as the starting strain for ALE analysis. a) Why the authors did not try to compare the ALE analysis between different reduced genome strains?

Response: The main purpose of this study was to recover the growth rate of MS56 in defined medium to the level of MG1655 and evaluate the changes responsible for recovery. Among the six reduced genome *E. coli* strains constructed previously, only two strains ($\Delta 16$ and MS56) have been reported to exhibit growth impairment^{7,8}. The other reduced-genome *E. coli* strains showed growth rates similar to that of the wild-type. It is difficult to infer genetic changes responsible for growth reduction from the normal strains. $\Delta 16$ was reported to show growth impairment and abnormal nucleoid organization even in rich medium, making manipulation and engineering difficult. In contrast, MS56 was ideal for these purposes because it showed impaired growth in defined medium while maintaining its growth rate in complex medium.

b) All of growth impaired reduced genome strains have the similar route to adapt the environmental condition like *sufD*, *rpoA*, *rpoD*, etc.? I think the initial introduction of mutation might be a kind of trigger with very stochastic event, is this wrong?

Response: We agree that the occurrence of mutation is a very random event.

c) During adapted evolution, series of introduction of mutation might also be stochastic events. Even though those mutation happen randomly, the adaptation by introducing mutation have a direction reaching to a certain optimal metabolic network point, which shows good balance to survive in the environmental condition used for the analysis. I think the route to get the optimal point might also be random, though the final goal share the same or similar point. So, my estimation is that, different starting strains may show different routes for adaptation but all of the adapted strains may share the similar metabolic feature. Is this idea totally wrong? One possibility might be that the initial introduction of mutation may stochastic but during adaptation, though lineage of adaptation may be varied, the final optimal metabolic balance might be very narrow range to adapt the environmental condition. To prove this, I think one option is to compare the ALE analysis between the different starting mutant strains. In other word, alteration of deoxynucleotide biosynthesis leading to the new balance of cellular concentration of ATP, NADH, NADPH and pyruvate, which showed in this manuscript, might be a goal for adaptation as an optimal point to the environmental condition?

Response: The reviewer has raised an important point. We agree that different lineages derived from the same starting strain share many common features. However, the presence of multiple adaptive routes to reach an optimal point remains controversial. Previous reports suggested that not only result of adaptation, but also the process of adaptation show large convergence to some degree. For example, glycerol-adapted *E. coli* strains contain mutated glycerol kinase (encoded by *glpK*⁹⁻¹¹). Additionally, regardless of the experimental condition, adapted *E. coli* acquires mutations in RNAP when long-term adaptation is conducted¹²⁻¹⁴. Additionally, mutations are introduced in a specific order; that is, mutations specific to a given stress (i.e. carbon source) emerges prior to the general mutations (i.e. RNAP)¹⁵. However, the exact position and effect of mutations differ. For example, glycerol-adaptation occurred on *glpK*, but the exact position and functional changes differed, showing an activity increase and resistance to FBP inhibition. The difference may result from the different routes of adaptation involving nucleotide-level genotypes. *In silico* assessment of the evolutionary pathway predicted that the evolutionary pathway and outcome can be convergent, divergent, or recurrent depending on the environmental conditions and microbial characteristics¹⁶. Comparing the ALE of a few different starting strains with eMS57 would not reveal whether

the final optimal metabolic balance is very narrow. Thus, this type of rule governing bacterial adaptation is largely unclear and requires careful examination in well-designed experiments involving massively parallel ALE with multiple starting strains. However, this examination is beyond the scope of the current study.

3. Translational buffering might be caused by the balance between the cellular amount of functionally active transcriptional and translational machineries. The functionally active cellular concentration of these factors may have an important information.

Response: We appreciate the reviewer’s insightful comment on translational buffering. Since Ingolia *et al.* evaluated translational dynamics in various organisms using ribosome profiling¹⁷, complex translational processes in microbial cells, such as ribosome buffering and translational pausing, have been observed. However, the molecular basis of these processes has not been widely examined and remains controversial¹⁸⁻²². In addition to ribosome profiling, recent advances in single-molecule imaging enabled evaluation of *in vivo* translational dynamics²³, but is limited to exploring genome-wide dynamics. Although ribosome buffering is unclear, we have included the reviewer’s insightful points in our manuscript as follows:

(Page 13, Lines 301 – 304) This suggests the presence of complex regulation at the post-transcriptional level, functional dynamics of transcription-translation (i.e. ribosome hibernation), or spatiotemporal resource limitation for translation, although the underlying mechanism remains largely unknown.

4. In *E. coli* during stationary growing phase, ribosomes form 100S inactive complex. For translational buffering, is there any mutation during adaptation involved in the regulation of translational activities such as *rmf* gene, which function to regulate the formation of inactive 100S ribosome?

Response: We agree with reviewer’s insightful comments regarding translational hibernation that is generally induced by stationary and stress responses. To identify the factors that induce translational buffering, we examined the mutations, transcriptional level, and translational level of genes related to translation. First, we evaluated 30 mutations remaining in the final ALE population located on 27 genes (**Supporting Table 3**). Except for *yaeJ*, no gene was related to translational regulation (including *rmf*). YaeJ (or ArfB) is known to rescue stalled ribosomes by hydrolyzing peptides and recruiting release factors^{24,25}. However, *yaeJ* is silent in both MG1655 and eMS57 (no detectable expression by both RNA- and Ribo-Seq; **Supplementary Table 5 and 6**). Thus, the mutations did not appear to affect translation in eMS57.

Supporting Table 3. Mutations remaining in the final ALE population as a subset of those shown in Supplementary Table 2. Ref: reference allele. AA: amino acid. freq: mutant allele frequency in percent (%). SNV: single-nucleotide variation. Ins: insertion. Del: deletion. MNV: multiple nucleotide variation. fs: frame-shift mutation. *: premature termination mutation.

Gene	Type	Ref	Allele	AA change	Allelic freq. at day 0	Allelic freq. at day 62	Description
yadG	SNV	G	A	E109K	0.0	57.5	putative ABC transporter
yaeJ	SNV	C	T	T18I	0.0	100.0	peptidyl-tRNA hydrolase, ribosome rescue factor

nfrA	SNV	C	T	G845D	0.0	11.5	bacteriophage N4 receptor
yceG	SNV	C	T	A84V	0.0	98.1	endolytic murein transglycosylase
ddpA	SNV	C	T	A133T	0.0	100.0	putative D,D-dipeptide ABC transporter
rspB	SNV	T	C	-	0.0	80.6	putative zinc-binding dehydrogenase
pdxY	SNV	C	T	-	0.0	71.8	pyridoxal kinase 2
pykF	SNV	C	T	-	0.0	14.0	pyruvate kinase I
yniA	SNV	G	A	G163R	0.0	10.9	putative kinase
ydjN	SNV	T	C	V92A	0.0	100.0	cystine/cysteine/sulfocysteine:cation symporter
cspC	SNV	C	T	E13K	0.0	97.5	stress protein
rcsB	SNV	G	A	A213T	0.0	70.1	transcriptional activator (capsule)
yfaL	SNV	G	A	-	0.0	62.5	putative autotransporter adhesin
lrhA	SNV	A	G	-	0.0	100.0	transcriptional dual regulator (fimbriae)
iscR	SNV	T	C	E43G	0.0	96.8	transcriptional dual regulator (iron-sulfur)
eamB	Ins.	-	G	A78fs	0.0	61.3	cysteine/O-acetylserine exporter
gabT	SNV	C	A	A378D	0.0	100.0	4-aminobutyrate aminotransferase
stpA^a	SNV	T	C	-	0.0	11.1	H-NS-like transcriptional repressor with RNA chaperone activity
stpA^b	Del.	TTC	-	E45del	0.0	58.4	
stpA^c	MNV	TTC	CTG	E45Q	0.0	11.9	
stpA	SNV	C	G	E45Q	0.0	11.9	
rpoD	SNV	T	C	S253P	0.0	100.0	RNA polymerase, sigma 70 factor
yrfF	SNV	A	G	N431S	0.0	98.6	inner membrane protein
xanP	SNV	C	A	L97M	0.0	100.0	xanthine:H ⁺ symporter
ilvN	SNV	G	A	Q68*	0.0	100.0	acetoxy acid synthase I subunit
asnC	SNV	A	G	Y82H	0.0	74.5	transcriptional dual regulator (Asp synthesis)
yifB	Ins.	-	G	G391fs	0.0	81.3	putative magnesium chelatase
yifK	Ins.	-	G	L22fs	0.0	75.2	putative transporter
yihN	SNV	T	C	-	0.0	78.4	putative transporter
leuQ	SNV	A	G	-	0.0	74.7	tRNA-Leu (CAG)

Furthermore, the transcription and translation levels of genes related to translation (ribosomal proteins, initiation factors, elongation factors, termination factors, and translation modulators) were unchanged in eMS77 (**Supporting Figure 4**).

Supporting Figure 4. Transcription and translation levels of genes related to translational machinery. Box and whisker plot showing the distribution of 54 ribosomal proteins. RNA: RNA expression level by RNA-Seq. RPF: ribosome protected RNA fragment by Ribo-Seq. IF: initiation factors, EF: elongation factors, RF: release factors.

In summary, there was no evidence that mutations or perturbations in translation machinery expression relieved translational buffering in eMS57. We supplemented the discussion of the mutation and expression levels of the translation machinery in the manuscript as follows:

(Page 16 – 17, Lines 388 – 395) A mutation on *yaeJ*, known to rescue stalled ribosome by hydrolyzing peptide and recruiting release factors³⁹, was the only mutation related to translation, however it seems not related to translational buffering because *yaeJ* was silent in both MG1655 and eMS57. In addition, expression level of ribosomal proteins and auxiliary factors (such as initiation, elongation, termination, and modulation) was unchanged. Thus, it remains unclear whether diminished translational buffering occurs because of reduced numbers of genes transcribed in a cell or the conservation of resources for translation otherwise used to produce unnecessary proteins and metabolites.

5. Figure 1c, grey bar might mean supplementation percentage of LB. Description in the legend is not sufficient for easy reading. It is required to check whole manuscript to give minimum but sufficient description, especially for figure legends.

Response: We thank the reviewer for the suggestion to improve the readability of the figures and legends. We revised all figures and their legends to be interpreted more easily. Specific revision in Figure 1c is shown below:

(Fig. 1c and legend)

Figure 1. Adaptive laboratory evolution (ALE) of a genome-reduced strain (MS56) and phenotypic examination of its evolved descendent, eMS57. (c) Cell growth trajectory showing changes in fitness during the ALE of MS56 in M9 minimal medium with supplementation of LB medium. Cell density was measured after 12 h of three individual batch cultivation and error bars indicate the s.d. LB supplementation was step-wise reduced from 0.1% to 0% over time. At the end of the ALE experiment, the evolved population exhibited restored growth rate in M9 minimal medium without any nutrient supplementation.

6. The similar problem as 5. in Figure 2e – j, it is not easy to distinguish colors of spots. And more explanation in the legend is required. What is FAM and VIC?

Response: Again, we appreciate the reviewer's inspection of the figures and legends. In the original submission, we decreased the size of the dots in the figures showing dPCR results, which made it easier to estimate the number of dots (infer allele frequency of mutant). As a tradeoff, the color of the spots became difficult to distinguish, as the reviewer commented. To increase the dot size, we created an additional plot showing the density of dots by contour as

a kernel density plot (**Supporting Figure 3E**). Furthermore, we have removed the green dots that were excluded from further analysis. With the increased dot size, adjusted color scheme, reduction of unnecessary dots, and kernel density plot, we hope that the revised figure can be interpreted more easily. Additionally, FAM and VIC are the most commonly used fluorescent dyes that conjugate with oligonucleotides. FAM- and VIC-derivatized primers were designed to specifically bind the mutant and wild-type allele DNA sequences, respectively (**Supplementary Table 7**). The dyes show different fluorescence emission wavelengths, and thus we can differentiate their signals in a mixture. When a mutant gene is amplified, only FAM fluorescence will be observed and *vice versa*²⁶. When coupled with digital PCR, we counted the exact number of mutant and wild-type alleles in a bacterial population. As the reviewer suggested, we revised the figure and its legend to include more detail and an explanation as follows:

(Supplementary Fig. 3 and legend)

Supplementary Fig. 3. Confirmation of variants detected by whole genome sequencing using dPCR-coupled TaqMan assay. (E) Plots show wild-type and mutant allele of *ilvN* in during the ALE. Each dot indicates an individual PCR reaction of dPCR. FAM and VIC fluorescence dyes were coupled with probe for mutant and reference DNA sequence, respectively. FAM-high and VIC-low dots (orange) indicate mutant allele, while FAM-low and VIC-high dots (blue) indicate amplification of wildtype allele.

7. I felt too many sub-figures in each of figures in the main text. It makes this manuscript not easy to read. I think it is one option to focus on the essential sub-figures and the rest of those might be moved into the supplementary information. Reconsideration may improve this manuscript easy to understand.

Response: Thank you for pointing out the organization of the figures. As suggested, we reorganized the sub-figures in the main figures. In **Fig. 2e–j**, the dPCR result was removed because it was redundant with **Supplementary Fig. 3E**. Promoter consensus motifs in **Fig.**

3b–d were transferred to **Supplementary Fig. 7**. Heatmaps showing the expression level of nucleotide metabolism in **Fig. 3h** and **i** were moved to **Supplementary Fig. 12**. Please find the reorganized figures as follows:

(**Fig. 2** and **Supplementary Fig. 3**)

Figure 2. Whole genome resequencing analysis of ALE experiment. (a) Spontaneous large deletion in eMS57 spanning 21 kb including 21 genes. No sequencing read was mapped onto this region. From *hycE* to *rpoS*, 21 genes were deleted. (b) The occurrence of large deletions was tracked by PCR amplification of the region. *hfq* was used as a positive control in PCR. (c) Growth rates of MS56 with *rpoS* deletion or a large deletion compared to MG1655 or eMS57. Error bars indicate the s.d. of three individual cultures shown in red circles. (d) Heatmap indicates frequencies of mutations in a given population. Shown are 31 mutations with allele frequency higher than 0.5 at least once during ALE. Dendrogram shows three clonal lineages determined by hierarchical clustering. SNV; single nucleotide variation, *; stop codon, fs; frame-shift. ED; Euclidean distance. (e) Three lineages and their sub-lineages were inferred from hierarchical clustering. *yciH* sub-lineage was emerged within the *sufD* lineage. *ddpA*, *ispU*, and *iscR* are sub-lineages of the *rpoD* lineage. *ydjN* is a sub-lineage of the *rpoA* lineage. Sub-lineages are presented as dotted lines. (f) Construction of the *cspC* point mutation on MS56 had a beneficial effect on fitness, whereas insertion of the *ilvN* mutation showed no effect. The *yifB* point mutation on MS56 decreased the growth rate. Error bars indicate s.d. of three biological replicates shown as red circles. *, *P*-value < 0.05, **, *P*-value < 0.01 (two-sided *t*-test of unequal variance, difference between growth rate of wild-type and mutant were tested, *n* = 3). (g) Growth rate of MG1655 and eMS57 in response to valine supplementation which inhibits cell growth by blocking isoleucine biosynthesis (white bar: MG1655, blue bar: eMS57, error bars indicate s.d. of three biological replicates shown as red circles). eMS57 was completely resistant to valine toxicity. Addition of isoleucine compensated for the inhibitory effect of valine in MG1655. (h) Additional 300 ALE generations of eMS57 and eMS57mutS⁺ strains revealed that inactivation of *mutS* increased the mutation rate of eMS57. Error bars indicate s.d. of two independent ALE populations (shown as red circles).

Supplementary Fig. 3. Confirmation of variants detected by whole genome sequencing using dPCR-coupled TaqMan assay. (E) Plots show wild-type and mutant allele of *ilvN* in during the ALE. Each dot indicates an individual PCR reaction of dPCR. FAM and VIC fluorescence dyes were coupled with probe for mutant and reference DNA sequence, respectively. FAM-high and VIC-low dots (orange) indicate mutant allele, while FAM-low and VIC-high dots (blue) indicate amplification of wildtype allele.

(Fig. 3, Supplementary Fig. 8 and 13)Figure 3. Transcriptome analysis of eMS57. (a) A total of 421 and 418 binding sites of σ^{70} (MG1655) and mutant σ^{70} (eMS57), respectively, were determined by ChIP-Seq; 320 sites are shared (“S”). Except for eMS57 deleted regions (“D”), wild-type (“M”) and mutant σ^{70} (“E”) specifically binds to 56 and 98 promoters, respectively (**b**) Box and whisker plots show changes of gene expression between MG1655 and eMS57 according to differential binding of σ^{70} . T: total promoters examined, S: shared promoters, M: MG1655-specific promoters, E: eMS57-specific promoters. *; *P*-value < 0.001 (Wilcoxon rank sum test). Box limits, whiskers, center lines indicate 1st and 3rd quartiles, 10 and 90 percentiles, and median of a distribution, respectively. White lines indicate median. (**c**) Glycolysis and TCA cycle expression levels are shown with indication of the required cofactors. EMP; Embden-Meyerhof-Parnas pathway, ED; Entner-Doudoroff pathway, GAP; glycerol-3-phosphate, Pyr; pyruvate. (**d**) Intracellular

NADH/NAD⁺ ratio was decreased and NADPH/NADP⁺ ratio was increased in eMS57. ATP intracellular level was decreased in eMS57. Red circles indicate three independent assays from biological replicates. Error bars indicate the s.d.

Supplementary Fig. 8. Consensus sequence of promoters used specifically in MG1655 (M), specifically in eMS57 (E), or in both strains (S). (A) MG1655-specific promoters, n = 56. (B) Shared promoters, n = 98. (C) eMS57-specific promoters, n = 320. (D) Native or mutant RpoD was heterologously expressed in MG1655 and bound promoter was immunoprecipitated by c-Myc epitope tagged to RpoD. Binding strength (DNA abundance in immune-precipitated DNA) on “M” and “E” promoters are presented. Ser253Pro mutation was sufficient for increasing the specificity to “E” promoters. However, mutation in RpoD did not change the binding to “M” promoters. (E) Promoter specificity of native RpoD tested by ChIP-qPCR showed high reproducibility with ChIP-Seq for MG1655. (F) Promoter specificity of mutant RpoD measured by ChIP-qPCR did not correlate with eMS57 ChIP-Seq, although specificity on the “E” promoters was increased. Binding of “M” and “S” promoters in eMS57 appeared to result from the collective interaction between mutant RpoD and other *trans*-acting elements, such as transcription factors. Error bars indicate the s.d. of two biological replicates, each consisting of three technical replicate reactions.

Supplementary Fig. 13. Expression of genes responsible for deoxynucleoside degradation and synthetic pathway. (A) Expression level of deoxynucleoside degradation pathway was increased. (B) Genes related with dNDP/dNTP synthesis from NDP/NTP were down-regulated.

Reviewer #2 (Remarks to the Author):

General:

In the manuscript of Choe and coworkers an adaptive laboratory evolution (ALE) study with a previously constructed genome-reduced *E. coli* strain is described. During ALE the growth performance of this strain on M9 minimal medium could be restored to wild-type level at low cell densities. Subsequent multi-omics analysis revealed many changes on the genomic and metabolic level and the authors did a great deal on speculating about their origin. In my opinion, the paper is mostly descriptive and does not provide any new fundamental insight into the core metabolism of *E. coli*. Hence, it does not deliver a blueprint for an optimal minimal genome of this model organism.

Major remarks:

1. L62, L38: “...we implement ALE to allow for self-optimization of the unknown processes encoded on a genome.” The reduced genome still contains unknown genes those functions can be self-optimized, but no clear functional annotation is provided after ALE. To be considered as an optimal minimal genome, at least, one functional annotation for each underlying gene should be provided.

Response: First, we are grateful to the reviewer for taking the time to evaluate this manuscript thoroughly. We also appreciate the reviewer’s comments regarding functional annotation of unknown processes and genes. As described in the main text, rather than revisiting the functions of all deleted genes, we adopted ALE to MS56 for self-optimization using unknown processes encoded in the genome. The unknown processes may include biological functions of the unknown genes, as the reviewer mentioned; however, these functions are not necessarily confined on unknown genes. In fact, growth restoration was mediated by the various biological processes described in the manuscript. First, deletion of *rpoS* immediately restored the growth rate of MS56 to 80% of that of MG1655 and we found that some mutations were responsible for growth recovery (*cspC*, for example), while some were mutational noise (*yifB*, for example). A full functional study of these functions was beyond the scope of this manuscript. Second, the metabolic and transcriptomic imbalance/perturbation induced by genome reduction was relieved. Transcriptomic re-balancing was, in part, carried out by the new promoter specificity of mutant RpoD. In the revised manuscript, we evaluated the promoter specificity of mutant RpoD (**Supplementary Fig. 8**). This transcriptomic shift mediated by mutation in RpoD is an unknown process that was self-optimized.

2. L73: “Optimal minimal genome” - For what biotechnological purpose?

Response: We failed to explain the term “optimal minimal genome” properly. The growth rate is an important factor in biological production. Particularly, for growth-coupled products such as amino acids and versatile carbon metabolites, the growth rate is directly related to productivity. In the minimal genome of MS56 and its close ascendant MDS42, 303 g of threonine was produced, which was 15.5-fold higher than that produced by the parental strain MG1655 and stable expression of foreign proteins was observed^{8,27}. However, this high productivity cannot be achieved under conditions in which the strain shows growth retardation. Additionally, fermentation time is an important factor in industrial production. A shorter fermentation time is beneficial for reducing costs, thus requiring rapid growth rate. Thus, the first challenge was to restore the growth rate. We adopted the ALE technique to restore the growth rate of MS56. To clarify this point, we revised the sentence as follows:

(Page 4, Lines 72 – 75) Thus, we exploit this robust method to recover innate potential to grow fast on a given medium and construct an optimal growth-recovered minimal genome that contains a minimal set of genes for enabling rapid growth.

3. L136: What else is missing to recovery the growth of eMS57?

Response: Based on the *rpoS* deletion study, deletion of *rpoS* was sufficient to recover the growth rate of MS56 in M9 medium at 80% of the level of MG1655. In addition to *rpoS* deletion, numerous factors contributed to the growth recovery of eMS57. Mutations, transcriptomic/translation remodeling, and metabolic rewiring affect recovery, as discussed in the manuscript. Briefly, MS56 accumulated 117 mutations during ALE and we demonstrated that point mutations in the stress protein CspC recovered the growth rate by 14% (Fig. 2f). Next, eMS57 showed distinct glycolysis and nucleotide metabolism from MG1655, possibly because of the different promoter specificity of the mutant RpoD (Fig. 3a–c, Supplementary Fig. 8, 11, and 12). This altered metabolic strategy in eMS57 established the cellular concentration of the versatile energy sources NAD(P)H and ATP. Finally, C-terminus truncation of the regulatory subunit of acetohydroxy acid synthase I (AHAS I; IlvN) rewired the branched chain amino acid biosynthesis that had been disrupted in the *E. coli* K-12 strain. Reconstitution of the *ilvN* mutation in MS56 resulted in a marginal growth increase (103% of MS56); however, this approach appears to have selective advantages because adaptation is not solely dependent on the growth rate.

4. L39: The high number of mutations is not surprising for such a long ALE experiment. In my opinion, however, the whole approach has a major flaw. As a reference case, wild-type *E. coli* should be evolved under the same conditions. Then differential genome, transcriptome and metabolome analysis can be applied to unravel the specific responses of MS56 in its genome-reduced background.

Response: We understand the reviewer’s criticism. The purpose of this study was to recover the growth rate of MS56 using ALE and determine the relevant systematic changes. We agree that wild-type *E. coli* evolves in glucose minimal medium; however, wild-type *E. coli* showed no severe growth retardation in glucose medium. Thus, mutations occurring because of ALE in wild-type *E. coli* are not strongly related to growth recovery and are mostly mutational noise. Comparing mutations of evolved wild-type *E. coli* and MS56 would reveal the specific mutational characteristics of MS56; however, this was not the main purpose of the study. Finally, as the reviewer pointed out in Comments #11 and #19, we referred to adaptation of wild-type *E. coli* as a reference case, as there was some convergence between wild-type and eMS57.

5. p7ff: “It is unlikely ...”; “Rather, it is more plausible...”; “However, it is unlikely ...”; “Instead, it is more likely...”; “Thus, we hypothesized that...”; “It is unclear how ...” These are a few examples on how the authors speculate about the meaning of their results throughout the whole text. In the end, the reader is left with many open questions.

Response: High-throughput systematic studies, including that conducted in this manuscript, have revealed that it is impossible to explain all factors causing a specific phenotype. We developed many hypotheses and speculations, as the reviewer mentioned. We apologize that we could not support or provide detailed explanations of all predictions. However, some predictions and hypotheses were obvious and self-explanatory, such as the mutational

dynamics described on Page 7. We focused on key hypotheses by focusing on important factors, such as RpoD and carbon metabolism, and described the changes in these factors. *In vivo* binding assays (chromatin immunoprecipitation), transcriptome assessment, and biochemical measurements were conducted. We acknowledge that some discussions are missing and thus have supplemented this information with more detailed discussions for the mutant RpoD and comparisons with previous ALE in the revised manuscript. We hope that the revised manuscript provides sufficient descriptions.

6. L211, L281: This application for eMS57 remains to be shown! All experiments were performed at very low cell densities (cmp. Fig. 1), which are not relevant for industrial production.

Response: We agree with the reviewer’s comment. High-cell density culture is important for industrial application, particularly for producing biomass-coupled primary metabolites. To test this, we conducted mini-scale fed-batch fermentation of eMS57 to evaluate the cell density culture of the strain (**Supporting Table 4**). Fermentation was conducted in a 2-L stirred-tank reactor containing 1 L of LB medium. The temperature was maintained at 37°C with a silicon heat jacket. The culture was aerated with 0.5 Mbar compressed air at a rate of 200 mL/min and agitated by pitched-blade impellers with the speed controlled from 500 to 1500 rpm to ensure that pO₂ did not decrease to below 50% saturation. Feeding solution (50% glucose (w/v), 23.65 mM MgSO₄, and 8.16 mM CaCl₂) was added at an initial rate of 20 mL/h and the rate was increased stepwise to support exponential growth. Antifoam 204 (Sigma) and 2 M NaOH were added to remove excess foam and maintain the pH of the medium. As a result, two strains reached a plateau of cell density after 12 h fermentation and eMS57 grew to a high cell density (4.73 g/L dry cell weight; DCW) which was comparable to the level of MG1655 (4.92 g/L DCW; **Supporting Table 4**).

Supporting Table 4. Fed-batch fermentation of MG1655 and eMS57. Fermentation was repeated three times on different days.

Strain	Biomass (g DCW/L); mean ± s.d.	Specific growth rate (h ⁻¹); mean ± s.d.
MG1655	4.917 ± 0.300	1.792 ± 0.017
eMS57	4.734 ± 0.077	1.240 ± 0.044

7. L320: This kind of study can be expected for this journal!

Response: Thank you for your comment. Since 2009, the translational dynamics of various organisms have been studied by ribosome profiling¹⁷⁻²². Because of these studies, complex translational processes in microbial cells, such as ribosome buffering and translational pausing, have been observed¹⁸⁻²². The molecular basis of these processes is completely unknown. Thus, a study explaining translational buffering would be suited for this journal, as the reviewer commented. However, in-depth analysis of translational buffering is beyond the scope of this manuscript. eMS57 is a suitable starting material for investigating translational buffering by comparing translationally unbuffered eMS57 and buffered relatives.

8. L347: “The clearest overall lesson is that growth retardation in genome-reduced strains results from a metabolic imbalance; a systemic function.” This sentence sums up the weakness of this study.

Response: Systems biology aims to understand a cell based on comprehensive evaluation of its collection of genes, transcripts, peptides, and metabolites. In this manuscript, we evaluated

the growth recovery of MS56 in glucose-limited medium using a systematic approach, as growth retardation could not be explained by the genetic background of the strain. For this, we used the ALE technique to induce self-optimization of MS56 growth. Through comprehensive systematic analysis after recovery, we found that growth retardation of MS56 was due to a metabolic imbalance, which was rewired during ALE. Furthermore, the metabolic rewiring was globally orchestrated by mutations in *rpoD* that altered the promoter binding of RNA polymerase. The metabolic pathway rewired was the glycolytic pathway, which plays a central role in utilizing glucose as a carbon source. The new balance between the two distinct glycolytic pathways occurred in the evolved strain to alter the cellular levels of ATP, NADH, and NADPH. Additionally, rewiring of nucleotide and BCAA metabolism resulted in optimal growth of the strain. This study reflects how little was known regarding how cells systems biology is involved in genome reduction and revealed the optimal metabolic framework required for producing genome-reduced *E. coli*.

Specific comments:

9. Please provide a table with all plasmids and major strains used and generated in this study including relevant characteristics and references. This would help a lot to follow your experiments and understand the ancestry of the corresponding strains.

Response: We thank the reviewer's for providing these suggestions. To help readers understand the history of the strains and source of plasmids used in this study, we supplemented the information on the plasmids and strains used in **Supplementary Table 8** as the reviewer suggested:

Supplementary Table 8. Strains and plasmids used in this study.

Strain	Description	Note
MG1655	Laboratory E. coli , train K-12, substr. MG1655	[52]
MS56	E. coli MG1655 with large deletions MD1 to MD56	[4]
eMS57	E. coli MS56 adaptively evolved in M9 glucose medium	This study
eMS57mutS ⁺	eMS57, puuP::mutS-kan	This study
MS56 Δ rpoS	MS56, rpoS::kan	This study
cspC ^{WT}	MS56, cspC::cspC-kan^R	This study
cspC ^{mut}	MS56, cspC::cspC(G37A)-kan^R	This study
ilvN ^{WT}	MS56, ilvN::ilvN-kan^R	This study
ilvN ^{mut}	MS56, ilvN::ilvN(C202T)-kan^R	This study
yifB ^{WT}	MS56, yifB::yifB-kan^R	This study
yifB ^{mut}	MS56, yifB::yifB(1169C insertion)-kan^R	This study
Plasmid	Description	Note
pKD46	lambda Red (exo , bet , gam), amp^R , repA101ts ori	[42]
pKD13	FRT- kanR -FRT, amp^R , R6K ori	[42]
BBa_J04450-pSB1C3	BBa_R0010(P _{LacI})-BBa_B0034 (RBS)-BBa_E1010 (mrfp1)-BBa_B0015 (terminator), cm^R , pMB1 ori	[31]

10. In some cases very few explanations in the figure legends makes it difficult to understand figures without further information.

Response: We thank reviewer's suggestion for improving the clarity and readability of our manuscript. We have revised all figure legends as follows:

(Figure legends)

Figure 1. Adaptive laboratory evolution (ALE) of a genome-reduced strain (MS56) and phenotypic examination of its evolved descendent, eMS57. (a) Growth profiles of

genome-reduced strain MS56 (red) and wild-type *E. coli* MG1655 (black) in LB medium. **(b)** Growth profiles of genome-reduced strain MS56 (red) and wild-type *E. coli* MG1655 (black) in M9 minimal medium. **(c)** Cell growth trajectory showing changes in fitness during the ALE of MS56 in M9 minimal medium with supplementation of LB medium. Cell density was measured after 12 h of three individual batch cultivation and error bars indicate the s.d. LB supplementation was step-wise reduced from 0.1% to 0% over time. At the end of the ALE experiment, the evolved population exhibited restored growth rate in M9 minimal medium without any nutrient supplementation. **(d)** Growth profiles of a clone eMS57 (red) isolated from the ALE population and wild-type *E. coli* MG1655 (black) in M9 minimal medium. **(e)** Morphological changes between MG1655, MS56, and eMS57. Upper panel, TEM images. Lower panel, SEM images. **(f)** Phenotype microarray characterization of MG1655 and eMS57 showing different nutrient utilization capability. **(g)** Intracellular and extracellular pyruvate concentrations for MG1655 and eMS57 at 4, 6, and 8 h after inoculation. Int; intracellular pyruvate concentration. Ext; pyruvate concentration in medium. Black (MG1655) and red (eMS57) lines show cell density at 4, 6, and 8 h after inoculation. Intracellular pyruvate level is presented as mole pyruvate per 10^9 cells and extracellular pyruvate level was measured in molar concentration. Individual data points are shown as blue circles and error bars indicate the s.d. **(h)** Pyruvate uptake function in MG1655 and eMS57 was examined by growth inhibition induced by a toxic pyruvate analogue (3-fluoropyruvate, FP). FP interferes with the function of pyruvate dehydrogenase (PDH encoded by *aceE*). Error bars throughout the figure indicate s.d. of three biological replicates.

Figure 2. Whole genome resequencing analysis of ALE experiment. **(a)** Spontaneous large deletion in eMS57 spanning 21 kb including 21 genes. No sequencing read was mapped onto this region. From *hycE* to *rpoS*, 21 genes were deleted. **(b)** The occurrence of large deletions was tracked by PCR amplification of the region. *hfq* was used as a positive control in PCR. **(c)** Growth rates of MS56 with *rpoS* deletion or a large deletion compared to MG1655 or eMS57. Error bars indicate the s.d. of three individual cultures shown in red circles. **(d)** Heatmap indicates frequencies of mutations in a given population. Shown are 31 mutations with allele frequency higher than 0.5 at least once during ALE. Dendrogram shows three clonal lineages determined by hierarchical clustering. SNV; single nucleotide variation, *; stop codon, fs; frame-shift. ED; Euclidean distance. **(e)** Three lineages and their sub-lineages were inferred from hierarchical clustering. *yciH* sub-lineage was emerged within the *sufD* lineage. *ddpA*, *ispU*, and *iscR* are sub-lineages of the *rpoD* lineage. *ydjN* is a sub-lineage of the *rpoA* lineage. Sub-lineages are presented as dotted lines. **(f)** Construction of the *cspC* point mutation on MS56 had a beneficial effect on fitness, whereas insertion of the *ilvN* mutation showed no effect. The *yifB* point mutation on MS56 decreased the growth rate. Error bars indicate s.d. of three biological replicates shown as red circles. *, P -value < 0.05, **, P -value < 0.01 (two-sided t -test of unequal variance, difference between growth rate of wild-type and mutant were tested, $n = 3$). **(g)** Growth rate of MG1655 and eMS57 in response to valine supplementation which inhibits cell growth by blocking isoleucine biosynthesis (white bar: MG1655, blue bar: eMS57, error bars indicate s.d. of three biological replicates shown as red circles). eMS57 was completely resistant to valine toxicity. Addition of isoleucine compensated for the inhibitory effect of valine in MG1655. **(h)** Additional 300 ALE generations of eMS57 and eMS57mutS⁺ strains revealed that inactivation of *mutS* increased the mutation rate of eMS57. Error bars indicate s.d. of two independent ALE populations (shown as red circles).

Figure 3. Transcriptome analysis of eMS57. **(a)** A total of 421 and 418 binding sites of σ^{70} (MG1655) and mutant σ^{70} (eMS57), respectively, were determined by ChIP-Seq; 320 sites

are shared (“S”). Except for eMS57 deleted regions (“D”), wild-type (“M”) and mutant σ^{70} (“E”) specifically binds to 56 and 98 promoters, respectively **(b)** Box and whisker plots show changes of gene expression between MG1655 and eMS57 according to differential binding of σ^{70} . T: total promoters examined, S: shared promoters, M: MG1655-specific promoters, E: eMS57-specific promoters. *; *P*-value < 0.001 (Wilcoxon rank sum test). Box limits, whiskers, center lines indicate 1st and 3rd quartiles, 10 and 90 percentiles, and median of a distribution, respectively. White lines indicate median. **(c)** Glycolysis and TCA cycle expression levels are shown with indication of the required cofactors. EMP; Embden-Meyerhof-Parnas pathway, ED; Entner-Doudoroff pathway, GAP; glycerol-3-phosphate, Pyr; pyruvate. **(d)** Intracellular NADH/NAD⁺ ratio was decreased and NADPH/NADP⁺ ratio was increased in eMS57. ATP intracellular level was decreased in eMS57. Red circles indicate three independent assays from biological replicates. Error bars indicate the s.d.

Figure 4. Post-transcriptional changes in eMS57 analyzed by Ribo-Seq. **(a)** Correlation between translation and transcription changes in eMS57 compared to in MG1655. Translational changes in eMS57 generally showed a weak correlation with transcriptional change (Total, R^2 of 0.213). Transcription of ribosomal proteins were upregulated, whereas translation remained relatively unchanged. Each dot indicates a gene. Pearson’s correlation (R^2) constants between transcription and translational changes in total genes (gray) and ribosome (blue) are presented. FC; fold-change (eMS57/MG1655). **(b)** Correlation between transcription and translation changes in DNA synthesis machinery (DNAP), transcription machinery (RNAP), nucleic acid metabolism, central carbon metabolism, and amino acid biosynthetic pathway. No changes were observed in the transcriptional and translational levels of DNA synthesis and transcription machinery. Translation of genes responsible for nucleic acid and central carbon metabolism correlated linearly to transcriptional change. BCAA biosynthetic genes were translated differently than transcription in eMS57. Each dot indicates a gene and dots are colored by their function or pathway. Correlation between transcription and translational changes were examined by Pearson’s correlation (R^2). FC; fold change (eMS57/MG1655). **(c)** Transcription and translation level of BCAA biosynthesis pathway. Translation of valine-resistant AHAS I were markedly upregulated in eMS57. AHAS; acetohydroxy acid synthase, DHMB; 2,3-dihydroxy-3-methylbutanoate, MOB; 3-methyl-2-oxobutanoate, 2-IPM; 2-isopropylmalate, IPOS; 2-isopropyl-3-oxosuccinate, 4-MOP; 4-methyl-2-oxopentanoate, AHB; 2-aceto-2-hydroxybutanoate, DHMP; 2,3-dihydroxy-3-methylpentanoate, 3-MOP; 3-methyl-2-oxopentanoate, IPMHL; isopropylmalate hydrolyase, IPMDH; isopropylmalate dehydrogenase. Metabolites colored green and red are consumed and produced by given enzymatic reaction, respectively.

11. Discussion is in major parts a repetition of the results. Please compare your results e.g. with the outcome of other ALE projects.

Response: We have revised the Discussion section as the reviewer suggested. We compared the mutations in eMS57 with those of previous ALE experiments and discussed the translation machinery and translational buffering:

(Page 15 – 16, Lines 359 – 385) We found that the suboptimal metabolic state of MG56 was fixed during ALE through transcriptome reprogramming by RNAP mutation. In previous reports of long-term ALE, RNAP was the most frequently mutated protein complex. Mutations in subunits constituting the holoenzyme (*rpoA*, *rpoB*, *rpoC*, and *rpoD*), additional factor *nusA*, and *rho* have been reported³⁴. During the ALE of MS56, we detected mutations in *rpoA* and *rpoD*. In RpoA, four mutants (R33H, two R317C, and R317H) were previously observed in ALE under heat stress. In our study, the mutation was located in the alpha C-

terminal domain (G279V), which is involved in the interaction with CRP and upstream elements of the promoter³⁵. The mutation in α -CTD may confer a selective advantage on glucose medium through different CRP regulation or transcriptional changes. However, the mutation in *rpoA* was eliminated because it was dominated by another mutant carrying an *rpoD* mutation. *rpoD*, in contrast, was the fifth most mutated gene among more than 100 cell lines produced by parallel ALE by Tenaillon *et al*³⁴. Harden *et al.* also reported mutations on *rpoD* during acid adaptation³⁶. Considering that most mutations occurred in auto-inhibitory region 1.1, which sequesters the DNA-binding region of free RpoD when heat-adapted³⁴, while acid-adapted *E. coli* and eMS57 accumulated mutations in the non-essential flexible linker region³⁶, there may be a functional context specifying the subunits and domain of RNAP to be mutated depending on the selective stress. Although the functional consequence of *rpoD* mutation was not examined in detail in previous studies, we determined its impact on a specific set of promoters and thus on the transcriptome and concomitant metabolic flux re-optimization that lead to optimal growth of eMS57. In addition, we would like to acknowledge that 63 transcription factors, comprised of 28 families, were deleted in MS56. Except for the *dicA* single knock-out strain (which can only be deleted with *dicB*³⁷), we have confirmed no significant growth defect of *E. coli* Keio strains with one of the 62 TFs deleted (**Supplementary Fig. 5**). Even though the growth was unchanged by transcription factor deletions, there must be transcriptional perturbation that interferes with new transcriptomic balance back and forth in eMS57.

(Page 16 – 17, Lines 388 – 395) A mutation on *yaeJ*, known to rescue stalled ribosome by hydrolyzing peptide and recruiting release factors³⁹, was the only mutation related to translation, however it seems not related to translational buffering because *yaeJ* was silent in both MG1655 and eMS57. In addition, expression level of ribosomal proteins and auxiliary factors (such as initiation, elongation, termination, and modulation) was unchanged. Thus, it remains unclear whether diminished translational buffering occurs because of reduced numbers of genes transcribed in a cell or the conservation of resources for translation otherwise used to produce unnecessary proteins and metabolites.

12. L43: To my knowledge, the genome size of native *M. mycoides* is about 1.08 Mbp. Please check.

Response: The reviewer's comment is correct. We apologize for the confusion. The native *M. mycoides* has a genome size of 1.08 Mbp, while the re-designed JCVI-syn3.0 has a genome size of 531 kb. Please refer to the revised manuscript as follows:

(Page 3, Lines 42 – 43) For example, a native 1.08-Mbp *Mycoplasma mycoides* genome and its re-designed version (JCVI-syn3.0) was generated by *de novo* genome synthesis.

13. L86: What were the criteria for selecting these 55 regions for deletion?

Response: The *E. coli* MDS43 strain, the parental strain of MS56, was constructed by deleting genetic elements responsible for genomic instability such as insertion sequence (IS) elements, transposases, phages, integrases, and recombinases³⁰. Further reduction of K-islands, flagella, fimbriae, part of the LPS synthetic genes, and genes responsible for anaerobic respiration was conducted in MDS43 to produce the MS56 strain⁸.

14. L95: Was the ALE performed in independent replicates? Can the results be reproduced in another ALE experiment? Please provide more details for the ALE

experiment in the methods section, e.g. culture volume, passage automatically or manually, ...

Response: The ALE experiment was conducted in three independent flasks as biological replicates. The mean and s.d. of the final cell density of each batch culture are shown in **Fig 1c**. On day 62, all three populations showed increased growth rates in M9 medium. Thus, the result show that the ALE experiment is reproducible at least at the phenotypic level.

According to previous reports of parallel ALE experiments, even when ALE experiments were conducted under the same conditions, different lineages with numerous different mutations emerged because mutation is a random event^{15,31}. However, at the level of genes, operons, or functional groups, some convergence was observed from independent clones, including mutations at different sites but on the same gene¹⁵. This convergence indicates that ALE experiments are reproducible at the genetic level. Thus, changes in eMS57, such as inactivation of *rpoS* and rewiring of BCAA metabolism, can be reproduced in another ALE experiment. We have supplemented the experimental details of ALE in the **Methods** section as follows:

(Page 18 – 19, Lines 429 – 442)

Adaptive laboratory evolution (ALE) and eMS57 isolation

Escherichia coli MS56 was grown in 50 mL of M9 glucose medium in a 250-mL Erlenmeyer flat-bottom flask at 37°C with agitation. To support the growth of MS56 in M9 glucose medium, 0.1 % LB medium was supplemented initially. The supplementation was reduced in a stepwise manner to eventually achieve supplement-free growth. Batch cultures were manually transferred to fresh medium every 12 h at an initial OD_{600nm} of approximately 0.005. Number of cell divisions during ALE was calculated from final and initial cell densities according to the following equation (**Eq. 1**).

$$\text{Number of generation} = \log_2 \frac{\text{Final cell density}}{\text{Initial cell density}} \quad (\text{Eq. 1})$$

After 807 generations of ALE, three single clones were isolated on M9 glucose agar medium. Because the three clones showed equivalent growth rate, one of the clones (named eMS57) was selected for further analyses and experiments.

15. L98: How was eMS57 isolated?

Response: According to our genome resequencing results (**Fig. 2d**), the final population appears to contain very few lineages, which may be a single dominant clone. Thus, three individual clones were isolated on agar medium and their growth was examined (**Supporting Figure 5**). The three clones showed no significant difference in growth rate, while clone #2 showed a slightly shorter lag time than the other two clones. Clone #2 was chosen for further experiments and named as eMS57.

Supporting Figure 5. Growth profile of MG1655 and three independent clones isolated from the final ALE population (day 62). (A) Growth curves of MG1655 and three clones isolated from the final population. (B) Specific growth rate of three isolates in M9 medium. Clone #2 was selected for analysis and named as eMS57. Error bars indicate the s.d. of three biologically replicated cultures.

We have included a description of eMS57 isolation in the **Methods** section as follows: **(Pages 18 – 19, Lines 429 – 442)**

Adaptive laboratory evolution (ALE) and eMS57 isolation

Escherichia coli MS56 was grown in 50 mL of M9 glucose medium in a 250-mL Erlenmeyer flat-bottom flask at 37°C with agitation. To support the growth of MS56 in M9 glucose medium, 0.1 % LB medium was supplemented initially. The supplementation was reduced in a stepwise manner to eventually achieve supplement-free growth. Batch cultures were manually transferred to fresh medium every 12 h at an initial OD_{600nm} of approximately 0.005. Number of cell divisions during ALE was calculated from final and initial cell densities according to the following equation (Eq. 1).

$$\text{Number of generation} = \log_2 \frac{\text{Final cell density}}{\text{Initial cell density}} \quad (\text{Eq. 1})$$

After 807 generations of ALE, three single clones were isolated on M9 glucose agar medium. Because the three clones showed equivalent growth rate, one of the clones (named eMS57) was selected for further analyses and experiments.

16. L103: Please provide experimental details for TEM and SM pictures.

Response: We supplemented the experimental procedures used for electron microscopy in the **Methods** section as follows:

(Page 19, Lines 445 – 461)

Electron Microscopy

For scanning electron microscopy, 1 mL of exponential phase culture was prefixed in 2.5% paraformaldehyde-glutaraldehyde mixture buffered with 0.1 M phosphate buffer (pH 7.2) at 4°C for 2 h. Next, the prefixed sample was treated with 1% osmium tetroxide solution buffered with 0.1 M phosphate buffer (pH 7.2) for 1 h at room temperature (25°C). The fixed sample was dehydrated in graded ethanol, substituted by isoamyl acetate, and critical point-dried in liquid CO₂. The sample was finally sputter-coated with gold in a Sputter Coater SC502 (Polaron, Quorum Technologies, East Sussex, UK) to 20 nm thickness and SEM images were obtained using the FEI Quanta 250 FEG scanning electron microscope (FEI, Hillsboro, OR, USA) installed at the Korea Research Institute of Bioscience and

Biotechnology at a 10-kV acceleration voltage. For transmission electron microscopy, a sample fixed using the same method as used for SEM imaging was dehydrated in graded ethanol, substituted with propylene oxide, and embedded in Epon-812 Resin for 36 h at 60°C. The embedded sample was ultra-sectioned with an Ultracut E Ultramicrotome (Leica, Wetzlar, Germany) and double-stained with uranyl acetate and lead citrate. The sample was examined under a CM20 transmission electron microscope (Philips, Amsterdam, Netherlands) installed at the Korea Research Institute of Bioscience and Biotechnology at a 100-kV acceleration voltage.

17. L105-114: Are these Biolog-experiments? If the data is based on one replicate only, it might not be feasible to draw the given conclusions without further experimental proof. Moreover, with the Biolog-system metabolic activity is measured, not growth!

Response: Yes, the sentence explains the Biolog experiments. The Biolog experiments were conducted using two independent PM plates. The two biological replicates showed a high correlation with the average Pearson correlation (R^2) of 0.92 (**Supplementary Fig. 2**). We agree with the reviewer’s comment regarding the results of Biolog. The Biolog measures respiratory activity of a cell using redox-dependent dye rather than measuring the growth rate, and thus represents the metabolic capability of a cell utilizing various nutrient sources. We revised the manuscript to precisely describe the Biolog experiment as follows:

(Page 5, Lines 107 – 114) For example, eMS57 did not show respiration capability on glycolate and glyoxylate as the sole carbon source because the genes responsible for glycolate utilization were removed by MD10 deletion⁴. There was no significant change in phosphorus and sulfur source utilization; however, MG1655 and eMS57 exhibited different nitrogen utilization preferences. The respiration rate of MG1655 in cytidine was much higher than that of eMS57, whereas eMS57 preferentially utilized uric acid as the sole nitrogen source; this may have originated from the deletion of nitrate respiration genes.

(Page 20, Lines 470 – 472) Finally, 100 μ l of 85% T cell resuspension was inoculated on PM plates and cellular respiration was measured using an Omnilog instrument (Biolog).

(Fig. 1f and legend)

Figure 1. Adaptive laboratory evolution (ALE) of a genome-reduced strain (MS56) and phenotypic examination of its evolved descendent, eMS57. (f)

Phenotype microarray characterization of MG1655 and eMS57 showing different nutrient utilization capability.

18. L114ff: What was the reason to measure pyruvate in the supernatant of cultures?

Response: We measured common byproducts, such as lactate, pyruvate, and acetate, by high-performance liquid chromatography (**Supporting Figure 6**). Briefly, 500 μ l of the culture was sampled every 2 h and centrifuged at 20,000 $\times g$ for 1 min. The supernatant was filtered using a Minisart RC Syringe Filter with a 0.2- μ m pore size (Sartorius) and analyzed with a Waters 2414 Refractive Index Detector (Waters) equipped with a Waters 1525 Binary HPLC Pump (Waters) and Waters 2707 Autosampler (Waters). Next, 20 μ l of sample was separated

on a MetaCarb 87H Column (Agilent) using 0.007 N sulfuric acid as a solvent and a 0.6 mL/min flow rate. MG1655 and eMS57 produced undetectable byproducts (malate, lactate, ethanol, and methanol) in addition to acetate. The two samples showed no significant difference in acetate production (**Supporting Figure 6**). However, eMS57 produced a small amount of byproduct, where glycerol and pyruvate were observed. Using HPLC analysis, glycerol and pyruvate could not be separated, and thus we conducted a colorimetric assay specific for pyruvate.

Supporting Figure 6. Cell density, glucose consumption, and byproduct production during batch fermentation of MG1655 and eMS57. Cells were grown in 50 mL of M9 glucose medium in a 250-L Erlenmeyer flask with rotary shaking at 37°C and 220 rpm. Measurement was conducted in three individual flasks and error bars indicate the s.d. MG1655 and eMS57 showed identical growth, glucose consumption, and acetate production, while minimal levels of glycerol/pyruvate were produced only by eMS57.

19. L149-164: Have mutations in *rpoA* or *rpoD* been identified in other ALE experiments?

Response: Yes, Tenaillon *et al.* reported mutations in *rpoA* and *rpoD*¹⁵. According to their report, *rpoD* was the fifth most frequently mutated gene following *rpoB*, *ybaL*, *cls*, and *rho*. Harden *et al.* also reported mutations in *rpoD* during acid adaptation²⁹. Additionally, mutations in *cspC* and *rpoS* were repeatedly found in long-term ALE experiments. In the same context as **Comment 11**, we supplemented the discussion comparing previously reported ALE-related mutations with our results as follows:

(Page 15 – 16, Lines 359 – 380) We found that the suboptimal metabolic state of MG56 was fixed during ALE through transcriptome reprogramming by RNAP mutation. In previous reports of long-term ALE, RNAP was the most frequently mutated protein complex. Mutations in subunits constituting the holoenzyme (*rpoA*, *rpoB*, *rpoC*, and *rpoD*), additional factor *nusA*, and *rho* have been reported³⁴. During the ALE of MS56, we detected mutations in *rpoA* and *rpoD*. In *RpoA*, four mutants (R33H, two R317C, and R317H) were previously observed in ALE under heat stress. In our study, the mutation was located in the alpha C-terminal domain (G279V), which is involved in the interaction with CRP and upstream elements of the promoter³⁵. The mutation in α -CTD may confer a selective advantage on glucose medium through different CRP regulation or transcriptional changes. However, the mutation in *rpoA* was eliminated because it was dominated by another mutant carrying an

rpoD mutation. *rpoD*, in contrast, was the fifth most mutated gene among more than 100 cell lines produced by parallel ALE by Tenaillon *et al*³⁴. Harden *et al.* also reported mutations on *rpoD* during acid adaptation³⁶. Considering that most mutations occurred in auto-inhibitory region 1.1, which sequesters the DNA-binding region of free RpoD when heat-adapted³⁴, while acid-adapted *E. coli* and eMS57 accumulated mutations in the non-essential flexible linker region³⁶, there may be a functional context specifying the subunits and domain of RNAP to be mutated depending on the selective stress. Although the functional consequence of *rpoD* mutation was not examined in detail in previous studies, we determined its impact on a specific set of promoters and thus on the transcriptome and concomitant metabolic flux re-optimization that lead to optimal growth of eMS57.

20. L196-197: How was the mutation rate determined?

Response: First, mutations were detected when mutated reads occurred in over 10% of total mapped reads in whole genome resequencing (please see **Methods**, section **Data Processing**). Regardless of the mutant allele frequency (number of mutant reads over total read) and clonal kinetics (occurrence, expansion, and extinction), each mutation was counted as one mutation once detected. The mutation rate was defined as the number of mutations divided by accumulated generations. Previous studies used similar methods³²⁻³⁴. This may not be a precise measure of the mutation rate; however, this method has been used to successfully estimate the overall mutation rate of bacteria. We supplemented the detailed methods used to determine the mutation rate in the **Methods** section as follows:

(Page 29, Line 749 – 751) Mutation rate (mutations per genome per generation) was calculated as number of mutation (including SNVs, MNVs, and indels) divided by cumulative generation.

21. L103-205: How was the performance of the two further evolved strains? Was the strain with more mutations better than the strain with less mutations?

Response: We did not observe a significant difference in the growth rate between eMS57 and eMS57MutS⁺ after 300 generations of additional adaptation (eMS57AA and eMS57MutS⁺AA, respectively) compared to their parental strain (**Supporting Figure 7**).

Supporting Figure 7. Growth rate of eMS57, *mutS* restored strain, and strains after 300 generations of additional adaptation. eMS57 and eMS57MutS⁺ strains were adapted for 300 generations in two replicate populations (Rep1 and Rep2). Error bars indicate the growth rate measured in five biological replicates.

A recent study of the relationship between hypermutators (lacking mutation suppressors) and adaptation showed that fitness increases during adaptation differ according to the mutation rate³⁵. According to the classification described by Sprouffske *et al.*, eMS57 with inactivation of the MutHLS mismatch repair system showed a medium mutation rate (MR^M). The MR^M strain showed increased fitness after 3000 generations. However, there was only a slight difference in fitness between MR^S and MR^M strains after 1000 generations. Thus, the duration of adaptation for eMS57 and eMS57MutS⁺ (~300 generations) were not sufficiently long to induce observable differences in fitness increases, although we observed an increased number of mutations in eMS57 (*mutS* negative) compared to in eMS57MutS⁺.

22. L237-238: Why should a generally increased transcription of sigma70-dependent genes enable the adaptation to M9 medium?

Response: The adaptation of eMS57 in M9 glucose medium was orchestrated by transcriptomic remodeling of carbon metabolism (i.e. glycolytic strategy and nucleotide metabolism) mediated by the new promoter specificity of σ 70 and other metabolic regulations as described on Lines 213 – 284. The sentence did not intend to mean that increased transcription of σ 70-dependent genes was responsible for the adaptation. Instead, the sentence describes the assumption that the transcription levels of many genes were changed in eMS57, which may have been induced by the different promoter specificity of mutant σ 70 (**Supplementary Fig. 8**).

23. How good is the match between ChIP_Seq and transcriptome data?

Response: As described briefly in **Fig. 3a–e**, we observed 421 and 418 RpoD binding events on promoters in MG1655 and eMS57, respectively. The eMS57-specific binding of RpoD at the promoter increased the expression level of genes downstream of the promoter in eMS57 compared to in MG1655 (please see **Fig. 3e**; “E” promoters). Additionally, promoters bound by RpoD only in MG1655 (**Fig. 3e**; “M” promoters) expressed downstream genes in MG1655 at higher levels than in eMS57, although the change was not significant. However, there was no correlation between changes in RpoD binding and the transcription level for promoters bound by RpoD in both MG1655 and eMS57 (**Supporting Figure 8A**). This is interesting but not surprising. Two explanations can describe the poor correlation.

First, transcription is dependent on not only RpoD, but also on numerous *cis*- and *trans*-acting factors such as transcriptional attenuation, transcription factors, and nucleoid associated proteins (NAPs). Next, many researches have debated about the quantitative traits of ChIP-Seq when comparing different samples. Even when ChIP-Seq data from samples were normalized by sequencing depth in this study, comparison between samples was technically complicated because of various factors such as the number of peaks, association/dissociation kinetics, etc. ChIP-Seq is a qualitative and semi-quantitative measure, and thus quantification is reliable only within a sample. In fact, we observed a weak correlation between the ChIP-Seq and RNA-Seq data within each sample (**Supporting Figure 8B and C**).

Supporting Figure 8. Correlation between RpoD binding strength and RNA expression.

(A) Changes in RpoD binding to the promoter and transcription level of bound promoters are presented. Each dot indicates an individual promoter ($n = 320$). Differential RpoD binding to the promoter of eMS57 over MG1655 did not result in differential transcription levels. (B and C) Correlation between RpoD binding strength and transcription level in (B) MG1655 and (C) eMS57. Each dot indicates an individual promoter ($n = 320$). Pearson's correlation constants between RpoD-binding strength and transcription level are also shown (R^2).

24. L253: ...eMS57 utilizes glucose via the ED-pathway in part,... \diamond Can you draw this conclusion solely based on the transcriptomic-data?

Response: To determine the systematic difference in the eMS57 transcriptome compared to MG1655, we conducted functional categorization of DEGs by pathway enrichment analysis (see **Methods** and **Supplementary Fig. 11**). Carbohydrate metabolism was enriched in both up- and down-regulated DEGs, which was interesting because ALE was conducted under carbon-limiting conditions. Thus, we explored the expression change of relevant pathways, such as glycolytic pathways, TCA cycle, and pentose phosphate pathway. Expression level of EMP and ED pathways were significantly changed (**Fig. 3c** and **Supplementary Fig. 12**); we are also aware of a previous report describing proteome costs³⁶. Our conclusions are also supported by the results of biochemical measurements (**Fig. 3d**).

25. L298: MG1655? I understood in the sentence before you measure in eMS57?

Response: We failed to clearly describe the ribosome profiling experiment. We measured and compared the translation of MG1655 and eMS57 by ribosome profiling. We revised the manuscript accordingly to avoid confusion as follows:

(Page 13, Line 305 – 307) Thus, we measured the translational level of MG1655 and eMS57 by ribosome profiling (Ribo-Seq) to examine whether the transcriptional changes were consistent with changes in the translational levels (**Supplementary Fig. 14** and **Supplementary Table 6**)

26. L308-309: How was this measured?

Response: Expression of red fluorescence protein was measured by flow cytometry. As described in the **Methods** section, cells harboring the high-copy mRFP1 expression plasmid, BBa_J04450-pSB1C3, were grown for 12 h in M9 glucose medium (with 0.5 mM of IPTG for induction) at 37°C. Next, 1 mL of the cell culture was diluted in 9 mL of PBS and the cells were dissociated with a cell strainer snap cap (Corning). A total of 100,000 cells was observed using an S3e Cell Sorter (Bio-Rad) and analyzed with FlowJo software (FlowJo)

(**Supplementary Fig. 15D**). The median fluorescence intensities (relative fluorescence unit; RFU) of MG1655 and eMS57 expressing RFP were 146 and 457, respectively, revealing a 3.1-fold higher fluorescence intensity of eMS57 compared to MG1655. We made an error in calculating the fold-difference in fluorescence intensity between the two strains. We sincerely apologize for this oversight. However, the conclusions remained the same and the value has been corrected in the revised manuscript as follows:

(**Page 14, Lines 317 – 319**) Indeed, eMS57 showed 3.1-fold higher fluorescence intensity than MG1655 (**Supplementary Fig. 12**).

27. L406: The Biolog does not measure growth!

Response: The reviewer's comment is correct. We intended to show the difference in metabolic capability between MG1655 and eMS57 upon genome reduction and ALE. We have revised the manuscript accordingly as follows:

(**Page 5, Lines 107 – 114**) For example, eMS57 did not show respiration capability on glycolate and glyoxylate as the sole carbon source because the genes responsible for glycolate utilization were removed by MD10 deletion⁴. There was no significant change in phosphorus and sulfur source utilization; however, MG1655 and eMS57 exhibited different nitrogen utilization preferences. The respiration rate of MG1655 in cytidine was much higher than that of eMS57, whereas eMS57 preferentially utilized uric acid as the sole nitrogen source; this may have originated from the deletion of nitrate respiration genes.

(**Page 20, Lines 470 – 472**) Finally, 100 μ l of 85% T cell resuspension was inoculated on PM plates and cellular respiration was measured using an Omnilog instrument (Biolog).

(**Fig. 1f and legend**)

Figure 1. Adaptive laboratory evolution (ALE) of a genome-reduced strain (MS56) and phenotypic examination of its evolved descendent, eMS57. (f) Phenotype microarray characterization of MG1655 and eMS57 showing different nutrient utilization capability.

28. L592-599: Were these experiments performed according to the MIQE-Guidelines?

Response: Yes, the experiments were performed according to these guidelines. The qRT-PCR analysis was designed to provide essential information regarding MIQE-guidelines, if applicable. The essential information suggested by MIQE-guidelines are described below. Some of the items are not applicable in this study, as qRT-PCR was used for cross-validation of RNA-Seq in a relative manner and not for absolute quantification or clinical detection.

- Samples

MG1655 and eMS57 grown in 50 mL M9 glucose medium in a 250-mL Erlenmeyer flask at 37°C were harvested at mid-log phase ($OD_{600\text{ nm}} \sim 0.55$ for MG1655, ~ 0.50 for eMS57). Next, 10 mL of the culture was harvested by centrifugation at $4000 \times g$ for 10 min at 4°C.

- Nucleic acid extraction

Total RNA was isolated from the cell pellet immediately after centrifugation using the RNASnap™ method³⁷. RNA integrity was assessed by electrophoresis on a 2% agarose gel (**Supporting Figure 9**) and we observed no signs of degradation or contamination. Total RNA was quantified using a NanoDrop 2000 spectrophotometer (Thermo). Next, 5 µg of total RNA was treated with 2 U of RNase-free DNase I (NEB) in a 50-µL reaction mixture at 37°C for 30 min. The sample was then purified by phenol-chloroform-isoamyl alcohol extraction followed by ethanol precipitation.

Supporting Figure 9. Total RNA analyzed on 2% agarose gel. Electrophoresis was conducted at 135 V for 10 min.

- Reverse transcription

cDNA was synthesized from 1 µg of DNA-subtracted RNA in a 20-µL reaction using the SuperScript III First-Strand Synthesis System (Thermo) according to the manufacturer's instructions. Briefly, 1 µg of DNA-subtracted RNA, 50 ng of Random Hexamer, 1 µL of 10 mM dNTP mix, and DEPC-treated water to bring the reaction volume to 10 µL were mixed in an RNase-free PCR tube. The mixture was incubated at 65°C for 5 min and placed on ice immediately after incubation. Next, 10 µL of cDNA Synthesis Mix (2 µL of 10× RT Buffer, 4 µL of 25 mM MgCl₂, 2 µL of 0.1 M DTT, 1 µL of RNaseOUT, 1 µL (200 U) of SuperScript III RT) was added and incubated at 25°C for 10 min. The mixture was incubated 50°C for 50 min followed by 85°C for 5 min; 1 µL (2 U) of *E. coli* RNase H was treated at 37°C for 20 min to remove the RNA. Synthesized cDNA was immediately placed on ice.

- qPCR target information and oligonucleotides

Five genes, *rcaA* (b1951), *ydjN* (b1729), *pepN* (b0932), *zapA* (b2910), *gadX* (b3516), and *rplR* (b3304), were subjected to quantification. The DNA sequences of the genes were obtained from the NCBI *E. coli* K-12 MG1655 reference genome sequence (NCBI Acc. NC_000913.3). Primers were designed by Primer-BLAST³⁸ and no non-specific binding was observed in the *E. coli* K-12 MG1655 genomic DNA sequence (Acc. NC_000913.3). Primer sequences are listed in **Supplementary Table 7**. Amplicon sizes of *rcaA*, *ydjN*, *pepN*, *zapA*, *gadX*, and *rplR* are 111, 101, 105, 108, 104, and 107 bp, respectively.

- qPCR protocol

qPCR was performed using iQ SYBR Green Supermix (Bio-Rad) from 1 µL of cDNA synthesized at RT (see above), 10 pmol of each primer in a 20-µL reaction under the following conditions: 40 cycles of 95°C for 10 s, 58°C for 30 s, and 72°C for 30 s. Reactions were monitored on a C1000 Thermal Cycler (Bio-Rad) equipped with a CFX96 Real-Time PCR Detection System (Bio-Rad). Real-time fluorescence was remotely gathered by Bio-Rad CFX Manager (v3.1) software.

- qPCR validation

After 40 cycles of qPCR, the products were analyzed by electrophoresis on a 2% agarose gel and no non-specific amplification was observed (**Supporting Figure 10**). The mean C_q of NTC was 38.6 (**Supporting Table 5**).

Supporting Figure 10. qPCR product analyzed on a 2% agarose gel by electrophoresis. Lane; L: DNA ladder, 1: *rcsA* from MG1655, 2: *ydjN* from MG1655, 3: *pepN* from MG1655, 4: *zapA* from MG1655, 5: *rplR* from MG1655, 6: *rcsA* from eMS57, 7: *gadX* from eMS57, 8: *pepN* from eMS57, 9: *zapA* from eMS57, 10: *rplR* from eMS57.

Supporting Table 5. Cq values of NTC

	rcsA	ydjN	gadX	pepN	zapA	rplR
Rep1	40.5	37.9	39.1	39.8	39.5	37.4
Rep2	39.0	39.0	39.0	39.6	38.9	37.5
Rep3	39.7	37.9	37.6	37.7	37.2	37.5

- Data analysis

Real-time fluorescence was gathered and the Cq value was determined using a built-in auto-detection method (single threshold) in Bio-Rad CFX Manager (v3.1) software. Further analysis was conducted using Microsoft Excel 2013 (Microsoft) without modifying data integrity. The reproducibility of replicates was high, such that no outlier was disposed. All data points were used to draw **Supplementary Fig. 9**. We added a more detailed explanation of the quantitative (reverse transcription) PCR experiments to provide essential information for readers, as the reviewer suggested.

(Page 28, Lines 666 – 685)

Quantitative and quantitative reverse transcription-PCR (qPCR and qRT-PCR)

Reverse transcription was performed from 1 µg of total RNA in 20 µl reaction using the SuperScript III First-Strand Synthesis System (Thermo) according to manufacturer's instruction. Briefly, 1 µg of DNA-subtracted RNA, 50 ng of Random Hexamer, 1 µl of 10 mM dNTP mix, and DEPC-treated water to bring reaction volume to 10 µl were mixed in RNase-free PCR tube. The mixture was incubated at 65°C for 5 min and placed on ice immediately after incubation. 10 µl of cDNA Synthesis Mix (2 µl of 10× RT Buffer, 4 µl of 25 mM MgCl₂, 2 µl of 0.1 M DTT, 1 µl of RNaseOUT, and 200U of SuperScript III RT) was added and incubated at 25°C for 10 min. Then the mixture was incubated 50°C for 50 min followed by 85°C for 5 min. 1 µl (2 U) of *E. coli* RNase H was treated at 37°C for 20 min to remove RNA. Quantitative PCR was performed in 20 µl reaction (10 µl of iQ SYBR Green Supermix (BioRad), 10 pmol of forward and reverse primers, 1 µl of cDNA or immunoprecipitated DNA) with the following conditions: 40 cycles of 95°C for 10 s, 58°C for 30 s, and 72°C for 30 s. Reactions were monitored on a C1000 Thermal Cycler (BioRad) equipped with a CFX96 Real-Time PCR Detection System (BioRad). All the primers were designed by Primer-BLAST⁴⁴ and there was no non-specific binding found in *E. coli* K-12 MG1655 genome sequence (Acc. NC_000913.3). Sequence of primers and size of amplicons are summarized in **Supplementary Table 7**. In ChIP-qPCR experiment, promoter region (peak region) was targeted for amplification and $\Delta\Delta Cq$ method was used for quantification with four reference peaks (*hpt*, *nrdR*, *yebS*, and *yecD*) as previously described⁴⁵.

Reviewer #3 (Remarks to the Author):

This is a nice piece of work regarding a very exciting topic that is the engineering of minimal cells. They are working with an *E.coli* missing 1.1 Mb of non-essential genes which grows poorly in minimal media. The authors have done adaptive evolution to improve its growth rate and they have obtained a new variant that grows efficiently in minimal medium. Then they have proceeded to determine what the genetic changes that improve the growth rate are. They have found a deletion of 21 Kbases plus different mutations in other genes and they have looked at the accompanying changes in transcription and translation.

1. My main concern is that the authors have found that a single deletion of a gene in that region explains 80% of the improvement in growth rate. This *per se* is interesting but also raises the question of how many of the changes in other processes or genes explain the remaining 20%. I think it is very important that the authors should do the same 21 Kbase deletion in the MS56 to see if the growth rate increases even further. This is important since if growth rate recovers above 80% with this deletion then many of the conclusions obtained after by the authors could be just mutational noise.

Response: We appreciate the reviewer's concern regarding the effect of mutations other than large deletions. We deleted the 21-kb large region from MS56 to mimic that of eMS57. The deletion strain ($\Delta 21$ kb) did not grow better than the *rpoS* deletion strain (Fig. 2c). Thus, we concluded that 80% of the growth improvement originated from *rpoS* deletion. We supplemented the results in the revised manuscript as follows:

(Fig. 2c and legend)

Figure 2. Whole genome resequencing analysis of ALE experiment. (c) Growth rates of MS56 with *rpoS* deletion or a large deletion compared to MG1655 or eMS57. Error bars indicate the s.d. of three individual cultures shown in red circles.

(Page 6, Lines 135 – 137) A single knockout of *rpoS* or deletion of the 21-kb region from MS56 recovered its growth rate to 80% of that of eMS57; however, the deletion did not fully recover to the growth rate of eMS57 (Fig. 2c).

2. Regarding the mutation in the sigma 70, and the changes in specificity I have my doubts that the differences found are significant looking at the large overlap and at the sequence fingerprints. Ser253 is not found at any of the Sigma 70 domains 1-4 and therefore it is difficult to see how it will affect specificity of the promoter. To see if this is the case they should do the same mutation in the WT *E. coli* and see if they get the same results.

Response: We appreciate the reviewer's insightful comment. As the reviewer suggested, we introduced mutant *rpoD* into wild-type *E. coli*. Rather than mutating the genomic copy of

rpoD, which we failed to construct, native or mutant *rpoD* was heterologously expressed under the Trc promoter using a plasmid system. By comparing immunoprecipitated DNA from *E. coli* MG1655 expressing native and mutant sigma 70, differential binding of promoters by mutant sigma 70 could be assessed. To avoid interference of the native sigma 70 from the genomic copy, we tagged recombinant sigma 70 with 2 tandem c-Myc epitopes. Using this method, we selectively immunoprecipitated plasmid-derived sigma 70 by using an anti-Myc antibody, but not an anti-sigma 70 antibody. We selected four “S” promoters showing similar peak intensities in MG1655 and eMS57 as reference genes for quantitative PCR. Additionally, we randomly selected two “M” promoters, two “E” promoters, and six “S” promoters to test the specificity of mutant sigma 70. Based on the qPCR results (Supplementary Fig. 8D), mutant sigma 70 showed no difference in binding to the “M” promoters. However, the specificity to “E” promoters was increased as in eMS57 (Supplementary Fig. 8D). Thus, we concluded that the Ser253Pro mutation in sigma 70 was sufficient to provide new promoter specificity to the “E” promoters. Additionally, the promoter specificity of native sigma 70 measured by ChIP-qPCR showed a high correlation with the ChIP-Seq results of MG1655, indicating that the two ChIP experiments were reproducible (Supplementary Fig. 8E). However, the promoter specificity of mutant sigma 70 measured by ChIP-qPCR did not correlate with that determined by ChIP-Seq in eMS57 (Supplementary Fig. 8F). Thus, we concluded that the new specificity of mutant sigma 70 on “M” and “S” promoters in eMS57 was not induced by Ser253Pro mutation alone but appears to be related to additional *trans*-acting elements, such as transcription factors. However, we demonstrated that mutation of sigma 70 was sufficient to provide new specificity of sigma 70 on “E” promoters, resulting in upregulation of genes controlled by “E” promoters (Fig. 3b). We have updated this information in the revised manuscript as follows:

(Supplementary Fig. 8)

Supplementary Fig. 8. Consensus sequence of promoters used specifically in MG1655 (M), specifically in eMS57 (E), or in both strains (S). (A) MG1655-specific promoters, n =

56. (B) Shared promoters, n = 98. (C) eMS57-specific promoters, n = 320. (D) Native or mutant RpoD was heterologously expressed in MG1655 and bound promoter was immunoprecipitated by c-Myc epitope tagged to RpoD. Binding strength (DNA abundance in immune-precipitated DNA) on “M” and “E” promoters are presented. Ser253Pro mutation was sufficient for increasing the specificity to “E” promoters. However, mutation in RpoD did not change the binding to “M” promoters. (E) Promoter specificity of native RpoD tested by ChIP-qPCR showed high reproducibility with ChIP-Seq for MG1655. (F) Promoter specificity of mutant RpoD measured by ChIP-qPCR did not correlate with eMS57 ChIP-Seq, although specificity on the “E” promoters was increased. Binding of “M” and “S” promoters in eMS57 appeared to result from the collective interaction between mutant RpoD and other trans-acting elements, such as transcription factors. Error bars indicate the s.d. of two biological replicates, each consisting of three technical replicate reactions.

(Page 10, Lines 229 – 236) To examine the effect of Ser253Pro mutation on the specificity of σ^{70} , we compared native and mutant σ^{70} in MG1655 which has no genetic background related to genome reduction and ALE. Under control of the Trc promoter, native or mutant σ^{70} tagged with the c-Myc epitope was expressed in MG1655 and bound DNA fragments were immunoprecipitated and quantified by qPCR. Mutant σ^{70} showed high specificity to “E” promoters (**Supplementary Fig. 8**), while the specificity to “M” promoters remained unchanged. Thus, the mutant σ^{70} bound an additional set of “E” promoters, while loss of the ability to bind “M” promoters was not caused by the mutation.

(Page 27, Lines 642 – 648) Heterologous expression of native and mutant σ^{70} .

Native or mutant *rpoD* was PCR amplified from MG1655 or eMS57 genomic DNA, respectively, using *rpoD_F* and *rpoD_R* primers. Plasmid backbone was also PCR amplified from pTrcHis2A plasmid (Invitrogen) using *pTrc_inv_F* and *pTrc_inv_R* primers. Two PCR products were combined using In-Fusion HD Cloning Kit (Takara) as manufacturer’s instructions. Two tandem c-Myc epitope was fused at the N-terminus of *rpoD* by *rpoD_R* and *pTrc_inv_F* primers.

3. Finally the part regarding translation is not very clear.

Response: We apologize for the unclear description of the translational dynamics of eMS57. First, to summarize translation in the manuscript, we explored the translation of the evolved strain by ribosome profiling. Ribosome profiling captures ribosome-protected mRNA fragments (RPF) that are actively translated. In general, the level of RPF correlates with the transcription rate (**Supporting Figure 11**). In *Streptomyces* and yeast, the translation levels of genes showing high transcription levels were lower than expected and *vice versa* for genes with low transcription levels because of post-transcriptional buffering or translational buffering^{19,20}. MG1655 showed the same phenomenon, such that the ratio of translation to transcription (RPF/RNA level; translational efficiency; TE) decreased as transcription level increased (**Supporting Fig. 15A** and **Supporting Figure 11A**). Unexpectedly, transcription and translation in eMS57 linear correlated with a slope of 0.973 and Pearson’s correlation constant of 0.750 (**Supporting Figure 11B**). However, we found no mutation or difference in the transcription and translation of translational machinery, which may contribute to the reduction of translational buffering.

Thus, unfortunately, sections regarding translational buffering are descriptive and undeterministic. However, the observation of non-buffered translation in eMS57 can be a starting point for investigating translational buffering. In-depth comparison of translation in eMS57 using a translationally buffered strain will provide insight into this phenomenon.

Supporting Figure 11. Correlation between transcription and translation level in (A) MG1655 and (B) eMS57. Transcription and translation in eMS57 correlated linearly with the slope of 0.973 and Pearson's correlation constant of 0.750.

4. As a conclusion I think it is a nice piece of work but I would like to see more controls about the role of the different mutations they have found. This is one of the problems of adaptive evolution, we can obtain better strains but it is difficult to pinpoint which changes are the responsible.

Response: We agree with the reviewer's comment. Adaptive evolution is a powerful method for rapidly obtaining desired phenotypes, but it is difficult to distinguish between causal mutations and other neutral mutations. Using current genetic techniques, including lambda recombination, it is difficult to introduce point mutations into the bacterial genome without scar sequence. Moreover, specifically devised methods, such as MAGE³⁹, are more problematic in this case because this method lacks antibiotic selection. Using MAGE, a successfully recombined strain with mutations that negatively affect the growth rate can be washed out during propagation to end up with a non-successful clone. Finally, mutating essential genes such as *rpoD* is extremely difficult, as the S253P mutation lies in the middle of the gene and recombination can occur through one-step double crossover with a large DNA piece, greatly decreasing recombination efficiency^{39,40}. We reproduced three mutations in the MS56 genome through multiple attempts of lambda recombination and the mutation in *cspC* increased the growth rate. Surprisingly, *yifB* mutation decreased the growth rate, indicating epistatic interactions between mutations⁴¹. This complex nature of mutation indicates that multiple combinations of mutations are required to determine the effect of mutations, but this approach is limited by current technological efficiency. Although we were not able to elucidate the full translation of the 117 mutations, we hope that the reviewer understands the practical limitations and why three SNVs were reconstructed.

- 1 Perez-Rueda, E. & Collado-Vides, J. Common history at the origin of the position-function correlation in transcriptional regulators in archaea and bacteria. *J Mol Evol* **53**, 172-179, doi:10.1007/s002390010207 (2001).
- 2 Baba, T. *et al.* Construction of *Escherichia coli* K-12 in-frame, single-gene knockout mutants: the Keio collection. *Mol Syst Biol* **2**, 2006 0008, doi:10.1038/msb4100050 (2006).
- 3 Kato, J. & Hashimoto, M. Construction of consecutive deletions of the *Escherichia coli* chromosome. *Mol Syst Biol* **3**, 132, doi:10.1038/msb4100174 (2007).
- 4 Bindal, G., Krishnamurthi, R., Seshasayee, A. S. N. & Rath, D. CRISPR-Cas-Mediated Gene Silencing Reveals RacR To Be a Negative Regulator of YdaS and YdaT Toxins in *Escherichia coli* K-12. *mSphere* **2**, doi:10.1128/mSphere.00483-17 (2017).
- 5 Huisman, G. W. & Kolter, R. Sensing starvation: a homoserine lactone--dependent signaling pathway in *Escherichia coli*. *Science* **265**, 537-539 (1994).
- 6 Mehta, A. P. *et al.* Catalysis of a new ribose carbon-insertion reaction by the molybdenum cofactor biosynthetic enzyme MoaA. *Biochemistry* **52**, 1134-1136, doi:10.1021/bi3016026 (2013).
- 7 Hashimoto, M. *et al.* Cell size and nucleoid organization of engineered *Escherichia coli* cells with a reduced genome. *Mol Microbiol* **55**, 137-149, doi:10.1111/j.1365-2958.2004.04386.x (2005).
- 8 Park, M. K. *et al.* Enhancing recombinant protein production with an *Escherichia coli* host strain lacking insertion sequences. *Appl Microbiol Biotechnol* **98**, 6701-6713, doi:10.1007/s00253-014-5739-y (2014).
- 9 Applebee, M. K., Joyce, A. R., Conrad, T. M., Pettigrew, D. W. & Palsson, B. O. Functional and metabolic effects of adaptive glycerol kinase (GLPK) mutants in *Escherichia coli*. *J Biol Chem* **286**, 23150-23159, doi:10.1074/jbc.M110.195305 (2011).
- 10 Herring, C. D. *et al.* Comparative genome sequencing of *Escherichia coli* allows observation of bacterial evolution on a laboratory timescale. *Nat Genet* **38**, 1406-1412, doi:10.1038/ng1906 (2006).
- 11 Honisch, C., Raghunathan, A., Cantor, C. R., Palsson, B. O. & van den Boom, D. High-throughput mutation detection underlying adaptive evolution of *Escherichia coli*-K12. *Genome Res* **14**, 2495-2502, doi:10.1101/gr.2977704 (2004).
- 12 Barrick, J. E., Kauth, M. R., Streliaoff, C. C. & Lenski, R. E. *Escherichia coli rpoB* mutants have increased evolvability in proportion to their fitness defects. *Mol Biol Evol* **27**, 1338-1347, doi:10.1093/molbev/msq024 (2010).
- 13 Conrad, T. M. *et al.* RNA polymerase mutants found through adaptive evolution reprogram *Escherichia coli* for optimal growth in minimal media. *Proc Natl Acad Sci U S A* **107**, 20500-20505, doi:10.1073/pnas.0911253107 (2010).
- 14 Utrilla, J. *et al.* Global rebalancing of cellular resources by pleiotropic point mutations illustrates a multi-scale mechanism of adaptive evolution. *Cell Syst* **2**, 260-271, doi:10.1016/j.cels.2016.04.003 (2016).
- 15 Tenaille, O. *et al.* The molecular diversity of adaptive convergence. *Science* **335**, 457-461, doi:10.1126/science.1212986 (2012).
- 16 Josephides, C. & Swain, P. S. Predicting metabolic adaptation from networks of mutational paths. *Nat Commun* **8**, 685, doi:10.1038/s41467-017-00828-6 (2017).
- 17 Ingolia, N. T., Brar, G. A., Rouskin, S., McGeachy, A. M. & Weissman, J. S. The ribosome profiling strategy for monitoring translation *in vivo* by deep sequencing of

- ribosome-protected mRNA fragments. *Nat Protoc* **7**, 1534-1550, doi:10.1038/nprot.2012.086 (2012).
- 18 Ebrahim, A. *et al.* Multi-omic data integration enables discovery of hidden biological regularities. *Nat Commun* **7**, 13091, doi:10.1038/ncomms13091 (2016).
- 19 Jeong, Y. *et al.* The dynamic transcriptional and translational landscape of the model antibiotic producer *Streptomyces coelicolor* A3(2). *Nat Commun* **7**, 11605, doi:10.1038/ncomms11605 (2016).
- 20 McManus, C. J., May, G. E., Spealman, P. & Shteyman, A. Ribosome profiling reveals post-transcriptional buffering of divergent gene expression in yeast. *Genome Res* **24**, 422-430, doi:10.1101/gr.164996.113 (2014).
- 21 Mohammad, F., Woolstenhulme, C. J., Green, R. & Buskirk, A. R. Clarifying the translational pausing landscape in bacteria by ribosome profiling. *Cell Rep* **14**, 686-694, doi:10.1016/j.celrep.2015.12.073 (2016).
- 22 Wang, S. H., Hsiao, C. J., Khan, Z. & Pritchard, J. K. Post-translational buffering leads to convergent protein expression levels between primates. *Genome Biol* **19**, 83, doi:10.1186/s13059-018-1451-z (2018).
- 23 Ruijtenberg, S., Hoek, T. A., Yan, X. & Tanenbaum, M. E. Imaging translation dynamics of single mRNA molecules in live cells. *Methods Mol Biol* **1649**, 385-404, doi:10.1007/978-1-4939-7213-5_26 (2018).
- 24 Chadani, Y., Ito, K., Kutsukake, K. & Abo, T. ArfA recruits release factor 2 to rescue stalled ribosomes by peptidyl-tRNA hydrolysis in *Escherichia coli*. *Mol Microbiol* **86**, 37-50, doi:10.1111/j.1365-2958.2012.08190.x (2012).
- 25 Handa, Y., Inaho, N. & Nameki, N. YaeJ is a novel ribosome-associated protein in *Escherichia coli* that can hydrolyze peptidyl-tRNA on stalled ribosomes. *Nucleic Acids Res* **39**, 1739-1748, doi:10.1093/nar/gkq1097 (2011).
- 26 Holland, P. M., Abramson, R. D., Watson, R. & Gelfand, D. H. Detection of specific polymerase chain reaction product by utilizing the 5'----3' exonuclease activity of *Thermus aquaticus* DNA polymerase. *Proc Natl Acad Sci U S A* **88**, 7276-7280 (1991).
- 27 Lee, J. H. *et al.* Metabolic engineering of a reduced-genome strain of *Escherichia coli* for L-threonine production. *Microb Cell Fact* **8**, 2, doi:10.1186/1475-2859-8-2 (2009).
- 28 Igarashi, K. & Ishihama, A. Bipartite functional map of the *E. coli* RNA polymerase alpha subunit: involvement of the C-terminal region in transcription activation by cAMP-CRP. *Cell* **65**, 1015-1022 (1991).
- 29 Harden, M. M. *et al.* Acid-adapted strains of *Escherichia coli* K-12 obtained by experimental evolution. *Appl Environ Microbiol* **81**, 1932-1941, doi:10.1128/AEM.03494-14 (2015).
- 30 Posfai, G. *et al.* Emergent properties of reduced-genome *Escherichia coli*. *Science* **312**, 1044-1046, doi:10.1126/science.1126439 (2006).
- 31 Tenailon, O. *et al.* Tempo and mode of genome evolution in a 50,000-generation experiment. *Nature* **536**, 165-170, doi:10.1038/nature18959 (2016).
- 32 Ford, C. B. *et al.* Use of whole genome sequencing to estimate the mutation rate of *Mycobacterium tuberculosis* during latent infection. *Nat Genet* **43**, 482-486, doi:10.1038/ng.811 (2011).
- 33 Kucukyildirim, S. *et al.* The rate and spectrum of spontaneous mutations in *Mycobacterium smegmatis*, a bacterium naturally devoid of the postreplicative mismatch repair pathway. *G3 (Bethesda)* **6**, 2157-2163, doi:10.1534/g3.116.030130 (2016).
- 34 Saxer, G. *et al.* Whole genome sequencing of mutation accumulation lines reveals a

- low mutation rate in the social amoeba *Dictyostelium discoideum*. *PLoS One* **7**, e46759, doi:10.1371/journal.pone.0046759 (2012).
- 35 Sprouffske, K., Aguilar-Rodriguez, J., Sniegowski, P. & Wagner, A. High mutation rates limit evolutionary adaptation in *Escherichia coli*. *PLoS Genet* **14**, e1007324, doi:10.1371/journal.pgen.1007324 (2018).
- 36 Flamholz, A., Noor, E., Bar-Even, A., Liebermeister, W. & Milo, R. Glycolytic strategy as a tradeoff between energy yield and protein cost. *Proc Natl Acad Sci U S A* **110**, 10039-10044, doi:10.1073/pnas.1215283110 (2013).
- 37 Stead, M. B. *et al.* RNAsnap: a rapid, quantitative and inexpensive, method for isolating total RNA from bacteria. *Nucleic Acids Res* **40**, e156, doi:10.1093/nar/gks680 (2012).
- 38 Ye, J. *et al.* Primer-BLAST: a tool to design target-specific primers for polymerase chain reaction. *BMC Bioinformatics* **13**, 134, doi:10.1186/1471-2105-13-134 (2012).
- 39 Wang, H. H. *et al.* Programming cells by multiplex genome engineering and accelerated evolution. *Nature* **460**, 894-898, doi:10.1038/nature08187 (2009).
- 40 Datsenko, K. A. & Wanner, B. L. One-step inactivation of chromosomal genes in *Escherichia coli* K-12 using PCR products. *Proc Natl Acad Sci U S A* **97**, 6640-6645, doi:10.1073/pnas.120163297 (2000).
- 41 Khan, A. I., Dinh, D. M., Schneider, D., Lenski, R. E. & Cooper, T. F. Negative epistasis between beneficial mutations in an evolving bacterial population. *Science* **332**, 1193-1196, doi:10.1126/science.1203801 (2011).

Reviewers' Comments:

Reviewer #1:

Remarks to the Author:

I satisfied the authors' responses to the original review comments. I think it is worth to be published.

Reviewer #2:

Remarks to the Author:

While the authors could satisfactorily answer all my minor questions, they failed to convincingly handle the most critical ones. In particular, additional data/experiments were expected and these were not provided.

L72ff: Strain eMS57 most likely does not contain a "minimal" gene set that enables rapid growth. Many genes are still of unknown function and some of them might also be dispensable for growth. This also relates to the term "optimal minimal genome", which the authors still use in the discussion section. A complete functional annotation is missing to support this claim. Otherwise, wording and significance of this study has to be adapted accordingly.

The authors admit that many hypotheses and speculations were developed and apologize that they could not support or provide detailed explanations of all predictions (answer to major remark 5). Clearly, this was not expected but some of these speculations might be resolved by conducting well-defined differential genome, transcriptome and/or metabolome analyses to unravel the specific changes in eMS57 compared to E. coli wild type (major remark 4). Moreover, an in-depth analysis of translational buffering was not conducted (major remark 7).

With regard to remark 6, the authors conducted new experiments and could show that higher cell densities can be reached with strain eMS57. However, these experiments were performed on complex medium! This is clearly against the original intention of this study, i.e. to derive a genome-reduced strain with fast growth on minimal medium. What is the growth performance (growth rate and cell density) of strain eMS57 applying bioreactor conditions and M9 minimal medium? Moreover, why is the growth rate of eMS57 significantly lower (31%) on LB medium under bioreactor conditions? In summary, feasible biotechnological application of eMS57 remains to be shown.

Reviewer #3:

Remarks to the Author:

I have read the detailed response to the reviewers and I find that they have satisfactorily addressed all the points raised, therefore I recommend acceptance

Point-by-point Response to the Reviewer's Comments

Reviewer #1 (Remarks to the Author):

I satisfied the authors' responses to the original review comments. I think it is worth to be published.

Reviewer #2 (Remarks to the Author):

While the authors could satisfactorily answer all my minor questions, they failed to convincingly handle the most critical ones. In particular, additional data/experiments were expected and these were not provided.

1. L72ff: Strain eMS57 most likely does not contain a “minimal” gene set that enables rapid growth. Many genes are still of unknown function and some of them might also be dispensable for growth. This also relates to the term “optimal minimal genome”, which the authors still use in the discussion section. A complete functional annotation is missing to support this claim. Otherwise, wording and significance of this study has to be adapted accordingly.

Response: We agree with reviewer's comment. In a strict sense, none of the top-down constructed genomes contain a “minimal” number of genes for growth. Furthermore, bottom-up constructed minimal genome JCVI-syn3.0 has never been proven to be contain minimal number of genes. Instead, MS56 and eMS57 have reduced-genome supporting their life. Thus, we have revised the word “optimal minimal genome” as follows:

(Page 4, Lines 71-73) Thus, we exploited this robust method to recover the innate potential for rapid growth on a given medium and constructed a growth-recovered genome containing reduced number of genes enabling rapid growth.

(Page 17, Lines 394-400) This study demonstrates ALE as a way to improve growth phenotypes of a genome-reduced strain in laboratory growth conditions. ALE provided an efficient way to restore genome-reduced *E. coli* fitness without additional genome engineering. Considering the cost and time consumed for *de novo* genome synthesis, integration of ALE with rational genome reduction can reduce the remaining practical challenges in the top-down approach to minimal genome construction. ALE as a learning tool reveals a lack of understanding of the reduced strain's systems biology.

2. The authors admit that many hypotheses and speculations were developed and apologize that they could not support or provide detailed explanations of all predictions (answer to major remark 5). Clearly, this was not expected but some of these speculations might be resolved by conducting well-defined differential genome, transcriptome and/or metabolome analyses to unravel the specific changes in eMS57 compared to *E. coli* wild type (major remark 4). Moreover, an in-depth analysis of translational buffering was not conducted (major remark 7).

Response: According to the reviewer's suggestion, we conducted ALE experiment of wild type *E. coli* K-12 MG1655. After 800 generations, we identified a total of 101 mutations from three individual populations (**Supplementary Fig. 4** and **Supplementary Table 4**). None of the advantageous mutations in eMS57 genome, such as *rpoS/mutS* inactivation and *rpoD* mutation, were found from the ALE of MG1655. Instead, the ALE populations have mutations on *rpoC* (RNA polymerase beta prime subunit) and *rpoB* (RNA polymerase beta

subunit; **Supplementary Table 4**). The two genes are the most frequently mutated genes during the ALE of *E. coli* (Wannier, T. M. *et al.* Adaptive evolution of genomically recoded *Escherichia coli*. *Proc Natl Acad Sci U S A*, doi:10.1073/pnas.1715530115 (2018); Deatherage, D. E., Kepner, J. L., Bennett, A. F., Lenski, R. E. & Barrick, J. E. Specificity of genome evolution in experimental populations of *Escherichia coli* evolved at different temperatures. *Proc Natl Acad Sci U S A* **114**, E1904-E1912, doi:10.1073/pnas.1616132114 (2017); Sandberg, T. E., Lloyd, C. J., Palsson, B. O. & Feist, A. M. Laboratory evolution to alternating substrate environments yields distinct phenotypic and genetic adaptive strategies. *Appl Environ Microbiol* **83**, doi:10.1128/AEM.00410-17 (2017); Tenaillon, O. *et al.* The molecular diversity of adaptive convergence. *Science* **335**, 457-461, doi:10.1126/science.1212986 (2012)), which are well known to induce large-scale transcriptional reprogramming (Conrad, T. M. *et al.* RNA polymerase mutants found through adaptive evolution reprogram *Escherichia coli* for optimal growth in minimal media. *Proc Natl Acad Sci U S A* **107**, 20500-20505, doi:10.1073/pnas.0911253107 (2010); Utrilla, J. *et al.* Global rebalancing of cellular resources by pleiotropic point mutations illustrates a multi-scale mechanism of adaptive evolution. *Cell Syst* **2**, 260-271, doi:10.1016/j.cels.2016.04.003 (2016)). Considering that the *rpoC* and *rpoB* mutations are commonly observed in the adaptively evolved *E. coli* strains, *rpoD* mutation on eMS57 is quite unique feature of genome-reduced bacteria. Additionally, four mutations (*nfrA*, *glpA*, *yfaL*, and *yifB*) are commonly found in the ALE of MG1655 and MS56. NfrA is an outer membrane bacteriophage N4 receptor and YfaL is a putative autotransporter adhesin. Culture condition in this study is not related to functions of the two genes, however, considering they are membrane proteins, there should be a selective advantage by mutating them. Interestingly, deleterious mutation *yifB*, encoding a putative ATP-dependent protease, which function remains unknown, was also observed during the ALE of MG1655 and this is a clear indication of epistasis of mutations and evolutionary convergence between MG1655 and eMS57. The reference ALE experiment has been included in the revised manuscript as follows:

Supplementary Fig. 4. Adaptive laboratory evolution of a wild type *E. coli* K-12 MG1655. (A) Growth rate trajectory shows growth rate increase during the ALE and supplementation of LB medium. Amount of LB supplementation was reduced the same as ALE of MS56. Orange line indicates LB supplementation in MG1655 ALE and gray line shows LB supplementation in ALE of MS56 as a reference. (B) Growth rate of 15 clones isolated from three end point cultures of ALE.

(Page 9, Lines 190-213) We next conducted ALE of MG1655 in M9 glucose medium with LB supplementation. After 800 generations, we identified a total of 101 mutations from three individual populations (**Supplementary Fig. 4** and **Supplementary Table 4**). None of the

advantageous mutations in the eMS57 genome, such as *rpoS/mutS* inactivation and *rpoD* mutation, were found from the ALE of MG1655. Instead, the ALE populations have mutations on *rpoC* (RNA polymerase beta prime subunit) and *rpoB* (RNA polymerase beta subunit; **Supplementary Table 4**). The two genes are the most frequently mutated genes during the ALE of *E. coli*^{15,25-27}, which are well known to induce large-scale transcriptional reprogramming^{20,28}. Considering that the *rpoC* and *rpoB* mutations are common in the adaptively evolved *E. coli*, the *rpoD* mutation on eMS57 is quite distinct feature of genome-reduced bacteria. Additionally, four mutations (*nfrA*, *glpA*, *yfaL*, and *yifB*) occurred during the ALE of both MG1655 and MS56. NfrA is an outer membrane bacteriophage N4 receptor and YfaL is a putative autotransporter adhesin. The culture condition in this study is not related to functions of the two genes, however, considering they are membrane proteins, there should be a selective advantage by mutating them. Interestingly, deleterious mutation *yifB*, encoding a putative ATP-dependent protease, whose function remains unknown, was also observed during the ALE of MG1655 and this is a clear indication of epistasis of mutations and evolutionary convergence between MG1655 and eMS57.

Taken together, clonal lineage analysis and the respective point mutations in MS56 demonstrated how subtle genetic variations orchestrate rapid adaptation of genome-reduced *E. coli* to the minimal media condition. It is notable that the genome-reduced MS56 followed a similar adaptation trajectory, such as mutating RNA polymerase subunits and inactivating redundant or unnecessary proteins, with the limited repertoire of genes. Although the genome-reduced *E. coli* showed similar adaptation mechanism with wild type *E. coli*, molecular players of the functional changes seem to be different due to the fundamental difference in gene composition.

Furthermore, transcriptomic analysis of eMG1655 (a clone isolated from the evolved MG1655 population) reveals transcriptomic changes of the evolved wild-type and the reduced genome compared to un-evolved wild-type *E. coli* (**Supplementary Table 7**). First, expression levels of EMP pathway was over 7-fold higher than ED pathway in MG1655 and eMG1655 (**Fig. 3c** and **Supplementary Fig. 13**). In eMS57, however, expression levels of EMP-specific enzyme were only 3.6-fold higher than that of ED enzymes (**Fig. 3c** and **Supplementary Fig. 13**), indicating that eMS57 utilizes glucose via the ED pathway in part, losing one ATP but gaining NADPH from NADH. Furthermore, it is more evident that eMS57 has an active deoxynucleoside degradation (**Fig 3e** and **Supplementary Fig. 14**). Taken together, it has become clear that the preferential use of ED pathway over EMP pathway and increased deoxynucleotide degradation are the distinct characteristics of eMS57. We thank the reviewer raising up this comment to improve our manuscript. Please refer revised manuscript as follows:

(Pages 12-13, Lines 283-288) Expression levels of EMP genes were 7.20 and 7.25-fold higher than ED pathway genes in MG1655 and eMG1655 (a clone isolated from the evolved MG1655 population) (**Fig. 3c** and **Supplementary Fig. 13**). In eMS57, EMP-specific enzyme expression levels were only 3.61-fold higher than ED enzyme levels (**Fig. 3c** and **Supplementary Fig. 13**), indicating that eMS57 utilizes glucose via the ED pathway in part, losing one ATP but gaining NADPH from NADH.

(Page 13, Lines 301-305) Transcriptomic analysis showed that the expression levels of deoxyribonucleotide biosynthesis were lower in both eMG1655 and eMS57 compared to MG1655, and conversely deoxynucleoside degradation was higher in eMS57 than eMG1655 and MG1655 (**Fig 3e** and **Supplementary Fig. 14**). Salvage of dNTP surplus is considered to

be a distinct metabolic feature of eMS57, whereas synthesis is inhibited in both evolved strains.

Figure 3. Transcriptome analysis of eMS57. (a) A total of 421 and 418 binding sites of σ^{70} (MG1655) and mutant σ^{70} (eMS57), respectively, were determined by ChIP-Seq; 320 sites are shared (“S”). Except for eMS57 deleted regions (“D”), wild-type (“M”) and mutant σ^{70} (“E”) specifically binds to 56 and 98 promoters, respectively (b) Box and whisker plots show changes of gene expression between MG1655 and eMS57 according to differential binding of σ^{70} . T: total promoters examined, S: shared promoters, M: MG1655-specific promoters, E: eMS57-specific promoters. *: P -value < 0.001 (Wilcoxon rank sum test). Box limits, whiskers, center lines indicate 1st and 3rd quartiles, 10 and 90 percentiles, and median of a distribution, respectively. White lines indicate median. (c) Glycolysis and TCA cycle expression levels are shown with indication of the required cofactors. EMP: Embden-Meyerhof-Parnas pathway, ED: Entner-Doudoroff pathway, GAP: glycerol-3-phosphate, Pyr: pyruvate. (d) Intracellular NADH/NAD⁺ ratio was decreased and NADPH/NADP⁺ ratio was increased in eMS57. ATP intracellular level was decreased in eMS57. Red circles indicate three independent assays from biological replicates. Error bars indicate the s.d. (e) Relative gene expression for deoxynucleoside degradation and synthesis between the evolved strains (eMS57 and eMG1655) and the wild type *E. coli* K-12 MG1655.

Supplementary Fig. 14. Expression of genes responsible for deoxynucleoside degradation and synthetic pathway. (A) Expression level of deoxynucleoside degradation pathway was increased. (B) Genes related with dNDP/ dNTP synthesis from NDP/NTP were down-regulated.

Finally, detailed analysis of translation buffering has been conducted. We have done meta-analysis of ribosome profile across all CDSs. According to the previous study (Woolstenhulme, C. J., Guydosh, N. R., Green, R. & Buskirk, A. R. High-precision analysis of translational pausing by ribosome profiling in bacteria lacking EFP. *Cell Rep* **11**, 13-21, doi:10.1016/j.celrep.2015.03.014 (2015)), resolution of ribosome profile depends upon position of sequencing read assignment (5' or 3'). In this study, 5' end assignment method showed clear 3 nt periodicity and used further (**Supplementary Fig. 17A**). Meta-analysis of ribosome profile aligned to start or stop codons revealed that there is no significant difference in translation initiation and termination between MG1655 and eMS57 (**Supplementary Fig. 17B**). In addition, RPF re-calculation excluding first and last 30 bp of the CDS confirmed that translational buffering was not an artifact generated by a temporarily paused ribosome at the Shine-Dalgarno sequence, start codon, or stop codon (**Supplementary Fig. 17C**). Furthermore, changes in elongation speed could be inferred from ribosome profile. If elongation speed changed, ratio of ribosome density at start/stop codon to codons in gene body would be changed, which did not. Thus, we concluded that the translational buffering was not originated from translational kinetics. Next, specific sequence motif on 5' UTR of CDSs had been accessed and there was no particular sequence content that causes translational buffering (**Supplementary Fig. 18A**). Also, computational prediction of translation initiation rate (TIR) based on energetics (Salis, H. M., Mirsky, E. A. & Voigt, C. A. Automated design of synthetic ribosome binding sites to control protein expression. *Nat Biotechnol* **27**, 946-950, doi:10.1038/nbt.1568 (2009)) revealed no correlation between TIR and TE, indicating the translational buffering was not associated with 5' UTR sequence (**Supplementary Fig. 18B**). We concluded that reduced number of genes and increased level of available ribosome per transcripts establish streamlined translation in eMS57. Two evidences support this claim: (1) Number of genes expressed in eMS57 is always few hundred smaller than that of MG1655 (more than 500 genes with RPKM cutoff of 10; **Supporting Fig. 1**) and (2) Lower variance of TE distribution in eMS57, compared to MG1655 (**Supplementary Fig. 16A** and **16B**).

Supporting Figure 1. Number of genes expressed in MG1655 and eMS57 according to RPKM cutoffs.

Supplementary Fig. 16. Translational efficiency of MG1655 and eMS57. (A, B) Division of genes in (A) MG1655 or (B) eMS57 into ten bins (percentile) according to their expression level showed translational buffering of genes with high expression level. Translational efficiency equals translation level (RPF) divided by transcription level. T; total genes. Red lines are linear regression of mean values of each bins.

Supplementary Fig. 17. Meta-analysis of Ribo-Seq profile. (A) Average ribosome profile aligned at the start or stop codon by different read assignment method. either 5` or 3` ends was tested to determine position of ribosome (Woolstenhulme, C. J., Guydosh, N. R., Green, R. & Buskirk, A. R. High-precision analysis of translational pausing by ribosome profiling in bacteria lacking EFP. *Cell Rep* 11, 13-21, doi:10.1016/j.celrep.2015.03.014 (2015)). 5` assignment method was used for meta-analysis, because the method provide more clear 3 nt codon periodicity of translation than 3` assignment. RD: average normalized ribosome density. Ribosome density was normalized with the maximum peak height in 200 nt window considered. (B) Meta-analysis of ribosome profile on CDSs assigned with 5` end of the reads. (C) RPF calculated by exact CDS region, CDS excluding initiation/termination region (30 bp), or CDS including 100 bp upstream/downstream region. Drawings above the graph illustrate the calculated regions.

Supplementary Fig. 18. Meta-analysis of sequence motif in 5' UTR of CDSs. (A) Sequence motif found from 5' UTR of 91 genes which are translationally buffered in MG1655 and un-buffered in eMS57. (B) Correlation between predicted translation initiation rate and RPF or TE. TIR was calculated from -30 to +30 nt mRNA sequence of the start codon using RBS Calculator (Salis, H. M., Mirsky, E. A. & Voigt, C. A. Automated design of synthetic ribosome binding sites to control protein expression. *Nat Biotechnol* **27**, 946-950, doi:10.1038/nbt.1568 (2009)).

(Page 34, Lines 815-818) In meta-analysis of ribosome profile, either 5' or 3' ends was tested to determine position of ribosome (Supplementary Fig. 17)^{53,54}. 5' assignment method was used for meta-analysis, because the method provide more clear 3 nt codon periodicity of translation than 3' assignment.

Taken together, in-depth analysis indicates that there was no particular factor or difference between the two strains that causes the translational buffering. However, we found that eMS57 with reduced number of genes had low variation of TE values all across their CDS that illustrates streamlined translation in eMS57. Please find revised manuscript as follows:

(Pages 15-16, Lines 352-377) Thus, reduced translational buffering is unlikely to be induced by abundance of transcription or translation machinery. Major rate-limiting and energy consuming steps in translation are initiation and termination. The two steps are highly likely to be different between MG1655 and eMS57, if the translational buffering was originated from the kinetics of translation. According to meta-analysis of ribosome density, MG1655 and eMS57 showed no difference near proximity of the start and stop codon (Supplementary Fig. 17). To examine that the translational buffering was an artifact of high ribosome density at start and stop codon, we recalculate RPF level of each CDS excluding 30 bp from both ends (Supplementary Fig. 17). The translational buffering remained unchanged in the recalculated RPF levels, indicating that it was not originated from translational kinetics.

Next, we examined sequence level difference on 5' untranslated region (5' UTR) which might induce the translational buffering. We sought common sequence motif in 5' UTR of 91 coding sequences (CDSs) that are translationally buffered in MG1655 ($TE < 0.8$) and un-buffered in eMS57 ($0.91 < TE < 1.1$). There was no particular sequence motif other than well conserved Shine-Dalgarno sequence (SD sequence; AAGGAG) (Supplementary Fig. 18). Because structure and interaction of the 5' UTR with ribosome play a critical role in

translation, we computationally predicted translation initiation rate (TIR), which is calculated collectively from multiple factors such as RNA structure and interaction with ribosome³⁸. TIR of the 91 CDSs with low TE and random CDSs showed no correlation with TE (**Supplementary Fig. 18**). Conclusively, there was no difference in translation mechanism between the two strains, the specific sequence motif, and the RNA structure that induces the translational buffering. Despite the same ribosome profile and sequence motif, the genes in eMS57 showed low variance in TE distribution (**Supplementary Figs. 16**). Thus, reduced number of genes provides an increased level of available ribosome and establishes the unbuffered translation in the reduced genome *E. coli*.

3. With regard to remark 6, the authors conducted new experiments and could show that higher cell densities can be reached with strain eMS57. However, these experiments were performed on complex medium! This is clearly against the original intention of this study, i.e. to derive a genome-reduced strain with fast growth on minimal medium. What is the growth performance (growth rate and cell density) of strain eMS57 applying bioreactor conditions and M9 minimal medium? Moreover, why is the growth rate of eMS57 significantly lower (31%) on LB medium under bioreactor conditions? In summary, feasible biotechnological application of eMS57 remains to be shown.

Response: We agree with the reviewer’s comment. We conducted fed-batch fermentation of *E. coli* in M9 glucose medium using a 2-l stirred-tank reactor. Temperature was maintained at 37 °C. Culture was aerated with 1 bar compressed air with a rate of 200 ml/min and agitated by pitched-blade impellers with speed controlled from 1000 to 1800 rpm so as pO₂ was not to drop below 90% saturation. Feeding solution (50% glucose (w/v), 23.65 mM MgSO₄, and 8.16 mM CaCl₂) was added by rate of 20 ml/hr to support exponential growth. Antifoam 204 (Sigma) and 2 M NaOH were added to remove excess foam and maintain pH of medium. As a result, eMS57 yielded final biomass of 1.67 gDCW/l. MG1655 and eMS57 showed no apparent difference of growth rate or final biomass yield in M9 glucose medium (Error! Reference source not found.). We included fermentation result of the two strain in revised manuscript as follows:

Supplementary Table 2. Fed-batch fermentation of MG1655 and eMS57. Fermentation was repeated twice in different days. DCW: dried cell weight.

Medium	Strain	Biomass (g DCW/l); mean ± s.d.	Specific growth rate (h ⁻¹); mean ± s.d.
LB	MG1655	4.917 ± 0.300	1.792 ± 0.017
	eMS57	4.734 ± 0.077	1.240 ± 0.044
M9 glucose	MG1655	1.743 ± 0.098	0.381 ± 0.011
	eMS57	1.671 ± 0.116	0.407 ± 0.004

(Page 6, Lines 123-126) Lastly, we obtained the biomass yield of eMS57 equivalent to MG1655 from fed-batch fermentation in LB or M9 minimal medium (**Supplementary Table 2**). This result illustrates comparable capability of eMS57 to its wild-type ancestor for potential applications in industrial scale.

(Page 22, Lines 522-529) **Fed-batch fermentation.** The fermentation was conducted in a 2-l stirred-tank reactor containing 1 l of LB or M9 glucose medium. Temperature was maintained at 37 °C with silicon heat jacket. Culture was aerated with 1 bar compressed air with a rate of 200 ml/min and agitated by pitched-blade impellers with speed controlled from

1000 to 1800 rpm so as pO₂ was not to drop below 90% saturation. Feeding solution (50% glucose (w/v), 23.65 mM MgSO₄, and 8.16 mM CaCl₂) was added by rate of 20 ml/hr to support exponential growth. Antifoam 204 (Sigma) and 2 M NaOH were added to remove excess foam and maintain pH of medium.

Lastly, *E. coli* strain evolved in M9 minimal medium has also been reported to have reduced growth rate in LB medium (Utrilla, J. *et al.* Global rebalancing of cellular resources by pleiotropic point mutations illustrates a multi-scale mechanism of adaptive evolution. *Cell Syst* **2**, 260-271, doi:10.1016/j.cels.2016.04.003 (2016); **Supporting Table 2**). Also, growth rate of eMS57 in LB medium was lower than MG1655 not only under bioreactor, but in flask cultures (**Supporting Fig. 2**). *E. coli* seems to loss its ability to rapidly grow in LB medium as a tradeoff of increase fitness in M9 medium. It may be due to reprogrammed transcriptome and metabolism in eMS57.

Supporting Figure 2. Growth profile of MG1655, MS56, and eMS57 in LB medium. Culture was done in three biologically replicated flask. Error bars indicate s.d. n.s: statistically not significant. *: *P*-value = 0.009 (two-sided *t*-test of unequal variance).

Reviewer #3 (Remarks to the Author):

I have read the detailed response to the reviewers and I find that they have satisfactorily addressed all the points raised, therefore I recommend acceptance.

Reviewers' Comments:

Reviewer #2:

Remarks to the Author:

The authors satisfactorily addressed all my concerns and the paper can be accepted for publication.